# Uncovering a Universal Abstract Algorithm for Modular Addition in Neural Networks

**Gavin McCracken**[1,2*], **Gabriela Moisescu-Pareja** [1,2*], **Vincent Létourneau**[1,3],
**Doina Precup**[1,2,4], **Jonathan Love**[5]

## Abstract

We propose a testable universality hypothesis, asserting that seemingly disparate neural network solutions observed in the simple task of modular addition are unified under a common abstract algorithm. While prior work interpreted variations in neuron-level representations as evidence for distinct algorithms, we demonstrate, through multi-level analyses spanning neurons, neuron clusters, and entire networks, that multilayer perceptrons and transformers universally implement the abstract algorithm we call the approximate Chinese Remainder Theorem. Crucially, we introduce approximate cosets and show that neurons activate exclusively on them. Furthermore, our theory works for deep neural networks (DNNs). It predicts that universally learned solutions in DNNs with trainable embeddings or more than one hidden layer require only $\mathcal{O}(\log(n))$ features, a result we empirically confirm. This work thus provides the first theory-backed interpretation of *multilayer* networks solving modular addition. It advances generalizable interpretability and opens a testable universality hypothesis for group multiplication beyond modular addition.

## 1 Introduction

The *universality hypothesis* posits that neural networks learning related tasks converge to similar internal solutions and that shared principles will underlie their representations regardless of architecture or initialization [1–3]. If true, it could provide a theoretical foundation for generalizing interpretability across diverse neural systems. Yet recent studies on related tasks (modular addition [4–8] and permutations [8, 9]) have cast doubt on this hypothesis by presenting *disjoint* interpretations of what networks learn between the two tasks, and even within the sole task of modular addition.

**We unify prior interpretations on modular addition.** By presenting a generalization of cosets—sets of elements with strict modular equivalence—to *approximate cosets* containing elements that are "behaviorally similar," instead of equivalent, our results abstract away the low-level details of how weights compute activations. This lets us show all previous interpretations [4–7, 10] are consistent under one common abstract algorithm we call the *approximate Chinese Remainder Theorem* (aCRT) (section 4.2). This abstraction reconciles the diversity in previously discovered mechanisms by interpreting them as different realizations of one algorithmic template. The main empirical results validate the breadth of our abstraction's accuracy across hyperparameters, architectures, and depth.

**We open the universality hypothesis as a testable conjecture across all group-theoretic datasets.** On modular addition (cyclic groups) we prove that all ReLU neurons learning sinusoidal functions activate only on approximate cosets or linear combinations of them (Theorem 4.4), giving a direct construction that instantiations of the abstract aCRT are learned. As our approximate cosets generalize cosets, work finding coset circuits in networks trained on permuting lists (permutation groups) [9] aligns with our results. This gives universality between datasets involving incredibly different groups.

---

*Equal contribution. {gavin.mccracken, gabriela.moisescu-pareja}@mail.mcgill.ca

[1]Mila [2]McGill University [3]Université de Montréal [4]Google DeepMind [5]Leiden University

39th Conference on Neural Information Processing Systems (NeurIPS 2025).

**We further the community's understanding of interpretations on modular addition.** Assuming neurons each learn a single frequency, we prove that networks exponentially reduce incorrect logit mass as more distinct frequencies are learned, concentrating the output near a Dirac on the correct answer. This recovers the theoretical result of [7], that 1-layer networks with neurons corresponding to each of the $\lfloor \frac{n}{2} \rfloor$ total frequencies that exist modulo $n$ have learned the maximum margin solution. Furthermore, a corollary predicts that deep neural networks (DNNs) have small margins between correct and incorrect logits unless $\mathcal{O}(\log(n))$ features are learned. These predictions are empirically validated across architectures, training regimes and both prime and composite moduli, whereas past works focused on prime moduli and representative networks from few seeds.

## 2 Related work

The first interpretability work in this domain aimed to understand the phenomenon of *grokking* [11]. Nanda et al. [4] analyzed 1-hidden layer transformers trained on modular addition, finding sinusoidal patterns of three to eight different frequencies in the weights, activations and attention. They showed that for each frequency, the embeddings placed inputs on a circle, and the network performed angle addition on this circle. Since adding angles corresponds to multiplying complex numbers, this nonlinear operation was attributed to the attention mechanism. This was termed the *Fourier Multiplication Algorithm* (later *Clock* [5]), and validated through ablation experiments.

Follow-up work proposed a generalization of the *Fourier Multiplication Algorithm* called *Group Composition via Representations (GCR)* algorithm [10], which extended the idea of angle addition by treating group elements as linear operators and composing them to simulate group multiplication, aiming to unify mechanisms across group tasks. They applied GCR to both modular addition and permutations ($S_n$), as representative group settings. However, later work by Stander et al. [9] reverse-engineered models trained on $S_n$ under identical conditions and found that networks instead learn *coset*-based circuits, refuting the GCR universality claim. Furthermore, Zhong et al. [5] showed that in modular addition, training hyperparameters could induce learning qualitatively different mechanisms. They introduced the *Pizza* circuit, which contrasted with [4]'s Clock. They even showed that both clock and pizza circuits could coexist within the same network simultaneously, suggesting that even with fixed data, networks could converge to non-unique mechanisms.

Meanwhile, theoreticians explored idealized settings for modular addition: Gromov [6], constructed a solution for 1-layer multilayer perceptrons (MLPs) with quadratic activations, showing a local minimum exists where each neuron specializes to a sinusoid of a single frequency, and consequently each of the $\lfloor \frac{n}{2} \rfloor$ frequencies is represented by some neuron in the network. Later, it was proven that this solution maximizes the margin [7], while independently and simultaneously, work connected margin maximization to grokking in similar networks [12].

By this point, the universality hypothesis appeared untenable. No similarities were found between groups and even on *just* modular addition, 1-layer MLPs found $\lfloor \frac{n}{2} \rfloor$ frequencies, 1-layer transformers learned substantially fewer, and changing hyperparameters resulted in learning different circuits [5].

## 3 Background

**Modular addition**, written as $c = (a + b) \mod n$, gives the remainder $c$ when the sum $a + b$ is divided by $n$. For example, $5 + 7 = 12$, and $12 \mod 12 = 0$. We can think of this as wrapping numbers around a circle of size $n$, once we pass $n$, we start over at 0. This arithmetic defines a structure known as the **cyclic group** $C_n = \{0, 1, \ldots, n - 1\}$. In $C_n$, every number is equivalent to its remainder class modulo $n$, denoted $(\mod n)$, *e.g.* $8 \equiv 2 \pmod 6$, since $8 = 6 \cdot 1 + 2$. Modular arithmetic also supports multiplication: for instance, $x \cdot y \equiv 1 \mod n$ when $y$ is the **modular inverse** of $x$, which we denote $x^{-1}$. These inverses exist when $x$ and $n$ are coprime.

Next, consider remainders mod $m$, where $m$ divides $n$. This coarser division groups elements of $C_n$ into **cosets**—sets of values that differ by multiples of $m$. For example, in $C_6$, the elements can be grouped into three cosets mod 3: $\{0, 3\}$, $\{1, 4\}$, and $\{2, 5\}$. Each coset marks out equally spaced points on the circle—it is a cycle. They will play a key role in our work, as neurons (and neuron clusters) will perform coset-like computations. See Appendix A for more discussion on group theory.

**The Chinese Remainder Theorem (CRT)** provides a way to simplify computations modulo $n$ by breaking them into smaller, independent computations. If $n$ factors into coprime integers (meaning they share no factors) $n = q_1 q_2 \cdots q_k$, then computing $(a + b) \bmod n$ is equivalent to computing $(a + b) \bmod q_i$ for each $q_i$, then reconstructing the original result. The CRT guarantees that the system of congruences $(a + b) \equiv m_i \pmod{q_i}$ (for $i = 1, \ldots, k$) has a unique solution modulo $n$. Each equation defines a coset, and the intersection of these cosets gives the final result. For example, suppose we want to find $c = (a + b) \bmod 91$ that satisfies: $c \equiv 3 \pmod 7, c \equiv 10 \pmod{13}$. Each congruence defines a coset $\{3, \mathbf{10}, 17, \ldots\}$ and $\{\mathbf{10}, 23, 36, \ldots\}$ respectively. Their intersection is 10 and thus the unique solution is $c = 10$. *We hypothesize that networks learn structure similar to the CRT decomposition to solve $a + b \pmod{n}$.*

**Cayley graphs** are critical for understanding section 4.1. Recall $C_n = \{0, 1, \ldots, n - 1\}$; take $s \in C_n, s \neq 0$. We generate the following Cayley graph, call it $\Gamma$, by starting at $g \in C_n$ and making an edge to $(g + s) \bmod n$, then an edge to $(g + 2s) \bmod n, \ldots$, until a cycle is made. For example, arrange 6 vertices in a circle. Using $s = 1$, step around the circle, connecting neighbors with an edge, generating a 6-cycle. Using $s = 2$ connects every second vertex, generating two disconnected 3-cycles based on where you start. These 3-cycles are the cosets: $\{0, 2, 4\}$, $\{1, 3, 5\}$. Using $s = 3$ gives three 2-cycles. See $s = 11$, $n = 66$ in panel 1 of Fig. 2. For an exposition including helpful visualizations involving how sine functions fit through cosets and approximate cosets, refer to A.2.1; these visualizations may help readers understand cosets geometrically.

**Clock and Pizza Interpretations.** Both circuits embed inputs $a$ and $b$ on a circle as $\mathbf{E}_a = [\cos(2\pi a/p), \sin(2\pi a/p)], \mathbf{E}_b = [\cos(2\pi b/p), \sin(2\pi b/p)]$. Post-attention, clocks compute the angle sum: $\mathbf{h}(a, b) = [\cos(2\pi(a+b)/p), \sin(2\pi(a+b)/p)]$, and pizzas compute a vector mean of the embeddings on the circle: $\mathbf{h}(a, b) = \frac{1}{2}(\mathbf{E}_a + \mathbf{E}_b) = \frac{1}{2}[\cos(2\pi a/p) + \cos(2\pi b/p), \sin(2\pi a/p) + \sin(2\pi b/p)]$.

**Problem setting and setup.** The task is modular addition: given inputs $(a, b)$, predict $c = a + b \bmod n$. The dataset includes all $n^2$ input pairs. Inputs are either one-hot encoded or embedded via a learned matrix with $n$ rows and 128-dimensional vectors, resulting in concatenated input pairs $(\mathbf{E}_a, \mathbf{E}_b)$. We train 1–4 layer MLPs and 1–4 layer transformers. We use the exact transformer architectures from [5], where attention is modulated by a coefficient $\alpha$. Pizza ($\alpha = 0.0$) uses constant, uniform attention (all-ones matrix), while clock ($\alpha = 1.0$) has learnable sigmoidal attention. Both models share the same structure: an embedding layer, one transformer block, and a 1-hidden layer MLP. We follow prior work in applying L2 regularization, shown to encourage generalization [11]. Also, we apply discrete Fourier transforms (DFT) to analyze the frequencies learned by each neuron.

# 4 Theoretical and empirical results

An intuitive overview of the mathematical details in section 4.1 is: neurons learn sinusoidal functions and we explain how to identify which Cayley graph a neuron understands; the Cayley graphs for the cyclic group are circle graphs; these Cayley graphs are generated by connecting every step size $d^{\text{th}}$ vertex on the circle; approximate cosets are sets of vertices that are close on the graph generated by $d$.

## 4.1 Simple neurons as (approximate) coset detectors

**The simple neuron model.** The mathematics in this paper assumes *the simple neuron model*: neurons approximate or specialize to sinusoidal functions of frequency $f$. The primary empirical results (section 5) validate this assumption across architectures, hyperparameters, random seeds, moduli and depth (Figs. [4-9]). This model is derived by generalizing the sinusoidal model for neurons from [6, 7, 13] as they assume one-hot encoded inputs to the network, to a model fitting both one-hot encoded architectures and those used in practice (having trainable embeddings [4, 5]). For $a + b \bmod n$, the inputs $a$ and $b$ are encoded as embeddings $A = [A_0, \ldots, A_{n-1}]$ and $B = [B_0, \ldots, B_{n-1}]$, where $A_i, B_i \in \mathbb{R}^d$. The output logits are $D = [D_0, \ldots D_{n-1}] \in \mathbb{R}^n$. Let $w(U, V)$ be the dot product of all values from $U$ with edge weights going to $V$. Then a **simple neuron** $N$ has frequency and phase shifts for A and B: $f, s_A, s_B \in C_n$, and positive real number $\alpha$ such that for each $k \in C_n$ we have

$$w(A_i, N) = \cos \frac{2\pi f(i - s_A)}{n}, \quad w(B_j, N) = \cos \frac{2\pi f(j - s_B)}{n}, \quad w(N, D_k) = \alpha \cos \frac{2\pi f(k - s_A - s_B)}{n}.$$

If training makes neurons converge to simple neurons, then their frequencies can be normalized to be 1 by applying an isomorphism (Def. 4.2). Consequently, qualitative comparisons between neurons of different frequencies are now possible (Fig. 1).

**Definition 4.1** (Step size). Let the step size be $d := (\frac{f}{\gcd(f,n)})^{-1} (\mathrm{mod}\, \frac{n}{\gcd(f,n)})$, where the modular inverse is used.

**Definition 4.2** (Remapping: frequency normalization). Consider the function $h(x) = \cos(2\pi f x/n)$ with frequency $f$. We define a new function $g$, allowing us to perform something analogous to a change of variables using the step size $d$: $g(d \cdot x) = h(x) \iff g(x) = h(d^{-1} \cdot x)$.

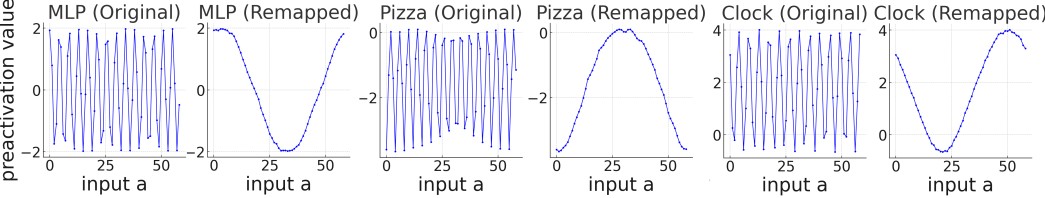

Figure 1: Preactivation values over $a$ fixing $b = 5$ on $c = (a + b) \bmod 59$ of a neuron from an MLP, pizza and clock show qualitative equivalence after remapping (Def. 4.2): they all have frequency 1. To see a neuron that is not qualitatively equivalent (corresponding to poor local minima), please see Appendix G.4, particularly Fig. 28.

**Approximate cosets.** The CRT relies on *cosets*. A neuron with frequency $f$ will only take values on a coset when $\gcd(f, n) > 1$. Since our experiments suggest neural networks approximate the CRT's decomposition *even when the learned frequencies don't factor the modulus*, we instead generalize cosets from requiring a strict equivalence among elements to **approximate cosets**. These require elements to be similar, which means the shortest path distance on their Cayley graph is small. Later, theorem 4.4 gives that all neuron activations (ReLU > 0) in all layers occur on approximate cosets.

Let $f \in C_n$. Recall Def. 4.1: $d$ determines how we step around the circle $C_n$. There are $n' = \frac{n}{\gcd(f,n)}$ distinct positions reachable in this way. These positions form cycle $C_{n'}$, which is a smaller (or equal) copy of $C_n$. $d \in C_n$ is the step size in $C_{n'}$. Generate $\Gamma$, the Cayley graph of $C_{n'}$ using $d$. We now introduce **approximate cosets** (approximate equivalence classes) using the minimum path distance between vertices in $\Gamma$. For distances $1 \le k_1 \le n$ and $1 \le k_2 \le n$, the approximate coset is the set of vertices on the path from $c - (dk_1)$ to $c + (dk_2)$, stepping by $d$. If $\gcd(f, n) > 1$: elements in the same coset as $c$ are distance 0 from each other as they are the same vertex on the Cayley graph, adjacent vertices are distance 1, etc (see panel 1 Fig 2). If $\gcd(f, n) = 1$: $\Gamma$ has one element with distance 0: $c$ (see panel 4 Fig 2). *Thus, approximate cosets are more general than cosets.*

**Definition 4.3** (Approximate cosets). Let $1 \le k_1 \le n$ and $1 \le k_2 \le n$. We call the set $\{c - k_1 d, \ldots, c - 2d, c - d, c, c + d, c + 2d, \ldots, c + k_2 d\}$ an **approximate coset**.

**Theorem 4.4.** *Simple neurons in layer 1 activate (ReLU > 0) on an approximate coset containing the correct answer c, by concentrating their preactivations on approximate cosets that contain a and b; all neurons in later hidden layers activate on linear combinations of approximate cosets.*

See Appendix A.2.1 for examples further illustrating the definitions introduced in this section.

## 4.2   The abstract approximate Chinese Remainder Theorem

By defining approximate cosets, and proving Theorem 4.4, a straightforward construction for the abstract algorithm being instantiated by neural networks can now be given. An abstract algorithm is a general template or high-level strategy for a problem. It outlines the steps to be followed but leaves room for different low-level implementations. For example, the classic breadth-first search algorithm traverses a graph by visiting all neighbors of a node, then all neighbors of those nodes, and so on. While the abstract idea is the same, the implementation details can vary: one version may use a linked list, while another may use a queue. Thus, the compiled machine code can differ significantly, like how the networks features can be computed by very different pizza or clock circuits.

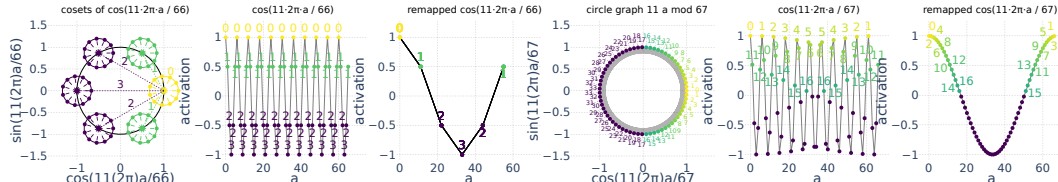

Figure 2: *Visualizing how neurons learn approximate coset structure.* Panel 1 shows the circle graph on 66 elements generated by starting at $a = 0$ and taking 6 steps of $\pm 11$, creating the $\frac{66}{11} = 6$ cosets of points $\{a \pmod 6 \equiv 0\}, \{a \pmod 6 \equiv 1\}, \ldots, \{a \pmod 6 \equiv 5\}$. The graph distance to each coset from coset $\{a \pmod 6 \equiv 0\}$ (in yellow) is given. 2: the neuron learned $\cos(\frac{11(2\pi)a}{66})$; the distances annotated on points follow from 1. This neuron only activates (ReLU $> 0$) on distances 0 and 1. 3: remapping shows all members of each coset collapse into an equivalence class. Panels 4-6 show the circle graph on 67 elements generated by $\pm 11$; since $\gcd(11, 67) = 1$, the neuron can't activate at the same strength on equivalent points (cosets) and instead activates with strengths proportional to distances on the Cayley graph. All elements the neuron takes positive values on are an approximate coset, shown in bright viridis colors decaying with distance. **Note:** each neuron "divides" the Cayley graph it activates on approximately in half.

**Remark 4.5.** *The sinusoidal neuron based CRT.* The CRT decomposes the modular system $(a + b)$ mod $n$ into $\mathcal{O}(\log(n))$ modular subsystems, which follows from a number having at most $\mathcal{O}(\log(n))$ prime factors. The CRT solves the original modular system by intersecting the cosets that the solution of each subsystem belongs to. Suppose the CRT can be used to decompose $(a + b) \mod n$. Then, a sinusoidal neuron based CRT is constructed with $\mathcal{O}(\log(n))$ unique frequencies $f$ with $\gcd(f, n) = f$. Make $\frac{n}{f}$ sinusoids, one for each coset ($\mathcal{O}(n)$ neurons), that are only positive on one of the cosets $\{a + b \pmod{\frac{n}{f}}\}$ using the $y$-intercept (neuron bias) so the neuron only activates if the answer is in the coset. The argmax of the linear combination of these neurons to the output logits selects the correct answer deterministically.

Armed with Theorem 4.4, deriving an abstract algorithm that Remark 4.5, instantiates is simple. Remark 4.5 assumes "the CRT can decompose $(a + b) \mod n$" into coset structure, but cosets are a subset of approximate cosets, making cosets a specific implementation under an abstract template. Furthermore, Theorem 4.7 addresses the approximate cosets case ($f$ does not divide $n$) in section 4.3. It gives that $\mathcal{O}(\log(n))$ randomly selected frequencies are enough to get reasonable margins between correct and incorrect logits, matching the number of frequencies required by the CRT.

**Abstract algorithm 4.6.** *The minimal template: the abstract aCRT.* Take $\mathcal{O}(\log(n))$ random frequencies and for each frequency generate sinusoidal neurons with that frequency and set their phases such that they pick out different approximate cosets that the answer $(a + b) \mod n$ is in.

Note, Alg. 4.6 is more general than the CRT and Remark 4.5, by handling cases where neurons learn frequencies with greatest common divisor (GCD) 1 with the modulus (approximate cosets). Two things are left to show: the $\mathcal{O}(\log(n))$ frequency bound in section 4.3 and that DNNs learn solutions that are realizations of Algorithm 4.6. The latter is exhaustively validated in section 5: finding all architectures are well abstracted by approximate cosets.

### 4.3 How many frequencies are needed to instantiate the aCRT with simple neurons?

We now present Theorem 4.7, which assumes that sinusoids are learned by the neurons, matching the simple neuron model. It predicts that DNNs push incorrect logit values down exponentially as the number of frequencies learned in the network increases. Analogously to Morwani et al. [7], it gives that the maximum margin solution requires all frequencies to be learned, but also yields additional information about the size of margins a network can acquire with fewer frequencies. A corollary gives that $\mathcal{O}(\log(n))$ frequencies are sufficient to get margins larger than $\Omega(\log(n))$.

Let $n$ be the number of output logits (matching the modulus), let $m$ be the number of distinct frequencies learned by the network and $m'$ be the maximum output logit value across the dataset. Fix two parameters: $0 < \delta < 1$, controlling the required margin between the correct and incorrect

outputs; $0 < \rho < 1$, the target probability of success. We model the neural network's output at logit $k$ by $h(k) = \sum_{\ell=1}^{m} \cos\left(\frac{2\pi f_\ell}{n}(k - i - j)\right)$, where each frequency $f_\ell$ is drawn uniformly at random from $\{1, 2, \ldots, \frac{n}{2}\}$. We seek conditions on $m$ so that, with probability at least $\rho$, the value $h(k)$ is well-separated from the maximum $m'$ for all incorrect outputs $k \neq i + j \mod n$.

Under the simple neuron model, the following result holds.

**Theorem 4.7.** *Suppose the integer number of distinct frequencies $m$ and reals $0 < \rho, \delta < 1$ satisfy the inequality*

$$m > \frac{2\log_e n - 2\log_e(2 - 2\rho)}{\log_e(\pi/\delta) - 1}.$$

*Then, with probability at least $\rho$, for all $k \neq i + j \mod n$, we have $m' - h(k) > \delta m'$. See Appendix C for a proof. Note, we want a large $\delta$, i.e. large margins.*

**Corollary 4.8.** *Learning $\mathcal{O}(\log(n))$ distinct frequencies gives a logit margin $\Omega(\log(n))$; after softmax, incorrect classes receive at most $n^{-\Omega(1)}$ probability mass.*

Note: it is still possible for networks to learn solutions utilizing a single frequency! Indeed, these solutions have poor margins, making them poor local minima. Unsurprisingly, they are rarely learned and only show up at the edge of the grid search returned by hyperparameter tuning. We empirically validate corollary 4.8 in Fig. 3 by varying the moduli over multiple orders of magnitude including moduli that are prime, composite numbers, highly composite numbers and powers of only 2 (e.g. 64, 256, etc.). The samples are such that the $R^2$ of fitting them with logarithmic functions is very high, empirically verifying the prediction of Corollary 4.8 that indeed $\mathcal{O}(\log(n))$ frequencies are reasonable. As the data contains both prime and composite moduli, it suggests moduli have little effect on what the network ultimately learns, though if a frequency divides the modulus it can be the case that fewer neurons of that frequency exist (see Appendix. G.7).

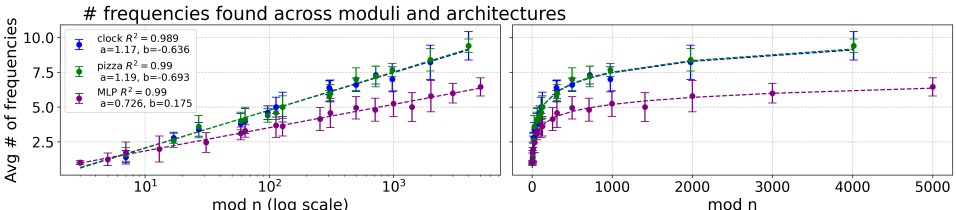

Figure 3: The number of frequencies found in clocks, pizzas, and MLPs as the modulus $n$ increases. We plot the data on logarithmic and linear axes, showing logarithmic fits have very high $R^2$ scores.

## 5 Empirical results supporting the simple neuron model and approximate coset abstraction

The details for the experimental results of this section can be found in Appendix E.

**One-hot encoded MLPs.** Previous work shows $\lfloor \frac{n}{2} \rfloor$ types of neurons are found, each specializing to a sinusoid with one frequency with 1 hidden layer [6, 7]. We show that adding either depth, or a trainable embedding matrix for inputs, causes the network to transition from learning $\lfloor \frac{n}{2} \rfloor$ types of neurons to much fewer types in Fig. 4. The presence of the trainable embeddings is why the models trained in [4, 8, 5] were observed to learn handfuls of frequencies (3-7) instead of $\lfloor \frac{n}{2} \rfloor$ frequencies, despite being one-hidden-layer models.

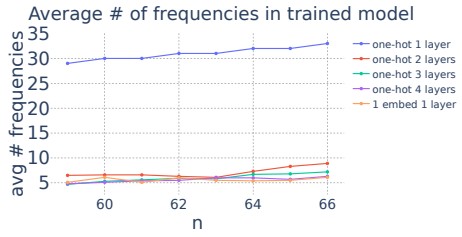

Figure 4: Average number of frequencies found in various MLPs over moduli 59-66.

**The neural pre-activations in 1-layer networks.** In the vast majority of cases, neuron pre-activations in all 1-layer architectures can be approximated well by degree 1 sine functions with frequency $f$. This is because the preactivations of most neurons are

"simple", meaning that they have frequency equal to 1 once remapped (Definition 4.2) (Fig 1). This is despite the presence of secondary frequencies in smaller width architectures. These occur less often as the width of the layers is increased (Fig 5), which shows neurons with secondary frequencies on the left, and the $R^2$ of fitting a single sine, or a sum of two sines with different frequencies, through the preactivations on the right. Thus, as the width is scaled, approximating neurons as simple neurons (1 sine is fit through their preactivations) becomes better. Note: at widths excessive for this task ($\leq 2048$ neurons) it rarely occurs, but it's the case that a few neurons can learn sinusoids with frequency $\frac{f}{2}$ (App. G.4 Fig. 28). While these still satisfy our definition of approximate cosets and Theorem 4.4 covers their existence, they break previous theoretical models assuming integer frequency [6, 7].

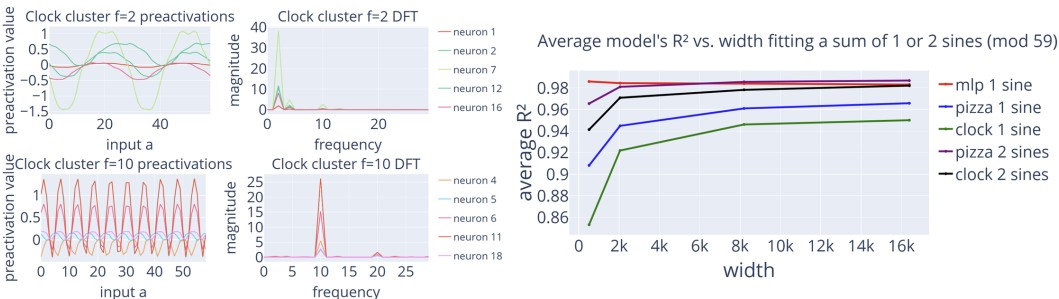

Figure 5: Left: cluster preactivations from a clock with small but present secondary spikes in the DFT. Right: as the width of the models is increased, the presence of the secondary spikes fades, 2 sines is fitting a sum of 2 different $f$ sines, allowing the inclusion of a secondary peak in the fit.

**Depth's effect on neural preactivations.** The first layer is fit with a very high $R^2$ in models of all depths, but adding layers introduces a caveat in the transformer architectures. In MLPs, it is possible to fit every neuron in every layer and maintain 100% test accuracy by assuming the neural preactivations are of the form $f(a, b) = \sin(fa + \phi) + \sin(fb + \phi)$, where $\phi$ is the phase shift (Fig. 6). In transformers however, this only works with a high $R^2$ for the first layer. The reason is that the form of the logits $f(a, b) = \cos(f(a + b - c))$, described by [4] is a second-order (quadratic) sinusoid and starts to appear in layers after the 1st layer, but before the logits. Indeed, we find that in deeper networks, neurons after the first layer can be either simple, $\cos(f(a + b - c))$, or a linear combination in superposition of these two forms. Thus, fitting just $\cos(a + b)$ or just $\cos(a) + \cos(b)$ is not sufficient to maintain 100% accuracy. To see this, see Fig. 8, which shows the percentage of activations that have their best $R^2$ achieved by fitting just order one sinusoids in $a$ and $b$ in a 2-hidden layer transformer. Thus, we fit linear combinations of $(\cos(f(a+b))) + (\cos(fa) + \cos(fb))$ through neurons in layers after 1 in Fig. 7. Note, MLPs can be fit well using only first-order sinusoids.

Furthermore, we could see a preference for learning precise cosets (should they exist) over approximate cosets as this could reduce approximation error in DNNs. We explore this in Fig. 9, showing that for $n = 66$, all architectures present a preference for learning precise cosets. This is strong evidence supporting the abstract aCRT algorithm as it implies DNNs try to learn CRT-like behaviour.

Our results show that in all architectures, layer 1 uses only simple neurons, with other layers still utilizing them, implying Algorithm 4.6 is instantiated. Furthermore, it follows from this that we've shown that all neurons downstream of layer 1 activate on linear combinations of approximate cosets. Combined with the results of [9], that the GCR algorithm [8] is not universal and instead coset circuits are learned in networks learning group multiplication in the permutation group, we open the universality hypothesis on group multiplication datasets as Conjecture 5.1.

**Conjecture 5.1.** *The universality in structures learned by DNNs trained to fit group multiplication will be found as coset circuits, and more generally as approximate coset circuits computing features. DNNs will make use of these circuits in a divide-and-conquer-like manner to achieve logarithmic efficiency.*

At this point, we have shown that our definition of approximate cosets functions as a sufficient abstraction for simplifying the representations learned by networks of various architectures.

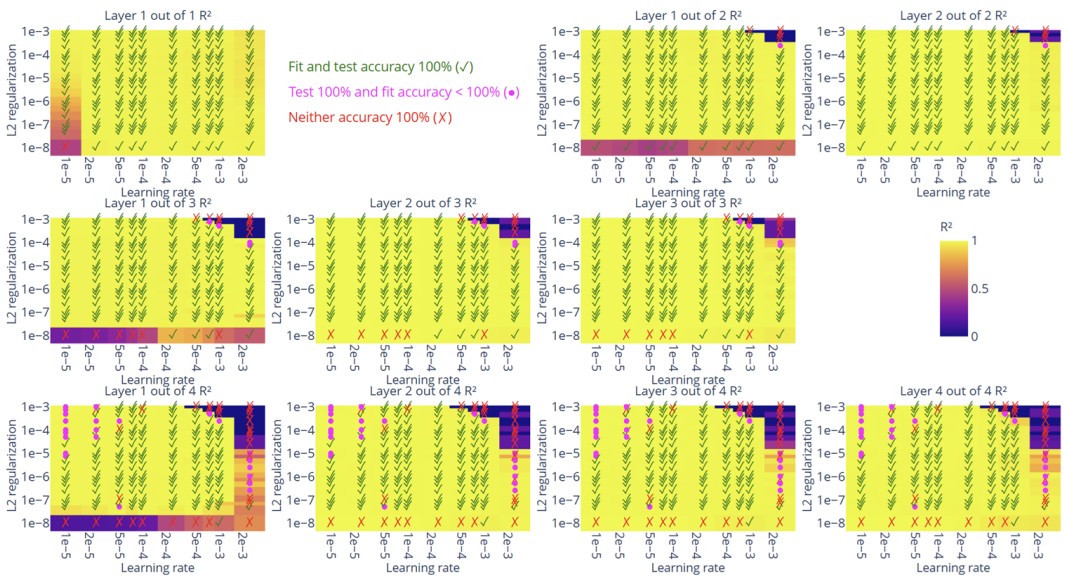

Figure 6: $R^2$ of fitting each neuron in layer 1 as a simple neuron and fitting a sum of sines of each frequency in layer 1 through layers 2-4 for 1,2,3 and 4 layer MLPs. The large volume of green checkmarks implies replacing neurons with simple neurons does not decrease the network's accuracy implying that our abstraction is robust to changes in training conditions and architectures.

# 6   Discussion and Conclusion

We argue that approximate cosets are critical in all architectures because they instantiate the aCRT. We support this both with empirical evidence across a large range of hyperparameters, seeds and different moduli, and theorems. Approximate cosets provide DNNs with structures analogous to the cosets the CRT operates on. The CRT uses $\mathcal{O}(\log(n))$ modular subsystems, and Corollary 4.8 gives that DNNs need $\mathcal{O}(\log(n))$ unique frequencies to analogously induce modular subsystems. Furthermore, approximate cosets shed light on how the network learns second order sine functions with ReLU activations (activating on the coset of $c$ requires understanding coset membership of $a + b$), a previously unexplained result in Nanda et al. [4] and Chughtai et al. [10]. As the proof for theorem 4.4 shows, a neuron learns to fire strongly on the coset that $c$ is in, by understanding which cosets $a$ and $b$ are in. The conclusion is that in abstracting away the small details in how weights in different architectures explicitly compute modular addition, we unify previous interpretations under the abstract aCRT (algorithm 4.6).

Our initial hypothesis was that neural networks trained on modular addition with composite moduli would learn the Chinese Remainder Theorem (CRT), leveraging coset structure where it naturally applies. Early empirical results supported this view, revealing a preference for frequencies that cleanly divide the modulus—suggesting alignment with exact coset structure. However, we observed that networks often learned only a single such frequency, alongside others that did not correspond to precise cosets. This prompted further investigation. Upon examining both qualitative and quantitative behavior, we found no meaningful distinction between neurons associated with coset-aligned frequencies and those that were not. This observation led to a critical insight: networks may be implementing an algorithmic template resembling the CRT even when its mathematical prerequisites are not strictly satisfied. This realization motivated the formulation of the abstract approximate CRT (aCRT), generalizing the role of cosets to approximate cosets as a unifying structure.

Due to modular addition being multiplication in the cyclic group and our approximate cosets generalizing cosets, our interpretation establishes universality between cyclic and permutation groups. This follows from the result that coset circuits are learned in networks trained on the permutation group [9]. Thus, we establish universality between very different tasks, related only by the fact they are both groups. This allows us to open the *testable* universality hypothesis on group multiplication datasets.

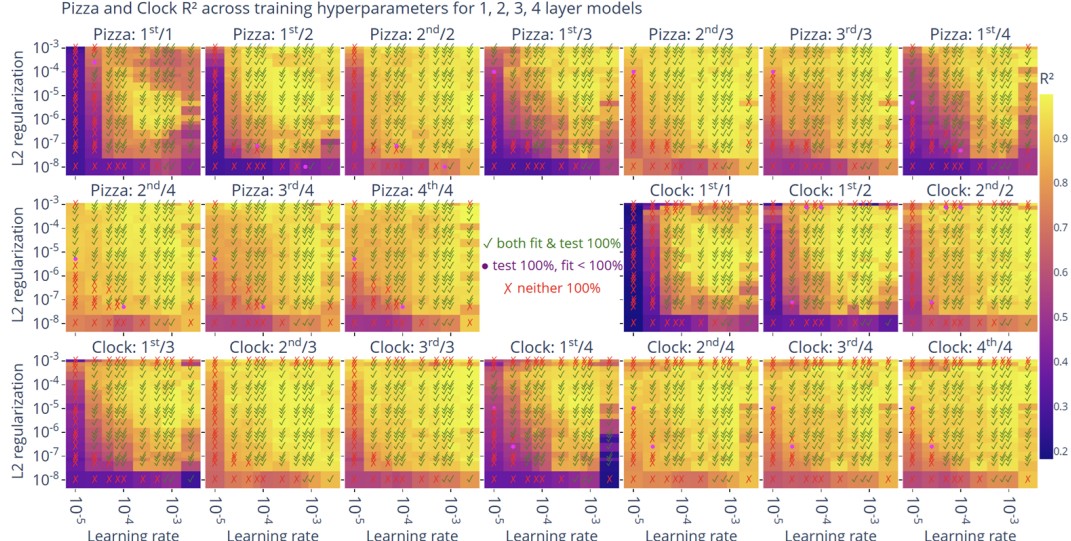

Figure 7: $R^2$ of fitting order 1 sines through neurons in layer 1, then fitting a sum of length equal to the number of unique frequencies in layer 1, of order one or order two sines for layers $2, 3, 4$. The large volume of green checkmarks tells us that our abstraction doesn't affect the model's accuracy and is robust to varying training conditions and architectures.

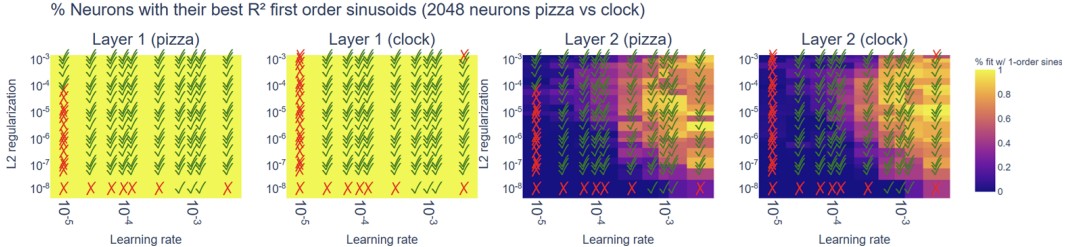

Figure 8: The percent of neurons with their best fit involving only first-order sines. The result can be interpreted as neurons existing that are operating on first-order sinusoids in deep layers. With ideal hyperparameters, almost 100% of neurons in layer 2 can have their best fits coming from order one sinusoids, though this occupies very little volume in the hyperparameter grid. This plot shows that the second-order sinusoidal fits of [4] are more experessiveness than necessary in the first layer, and not optimal.

## 6.1 Toward uncovering the nature of universality

It's important to note that our work doesn't fully illuminate how deep neural networks are *universally* learning modular addition. We propose the following definition, being *abstract universality*, which attempts to capture what is meant by the universality hypothesis which asserts that "models learn similar features and circuits across different models when trained on similar tasks." [8]. Our issue with the quoted statement, is "what does similar" *mean*? The work of Zhong et al. [5] claimed to find two disjoint and disparate circuits, being a clock, or a pizza, but our work shows these two circuits can be unified under one abstraction. It is therefore the case that these two circuits could be different implementations of one abstract divide-and-conquer algorithm. By abstract, we mean an object that can't be directly instantiated (e.g. abstract class in computer programming).

With that said, we propose the idea of *abstract universality*. DNNs trained on similar data learn different implementations of one abstract algorithmic class *i.e.* one algorithmic strategy.

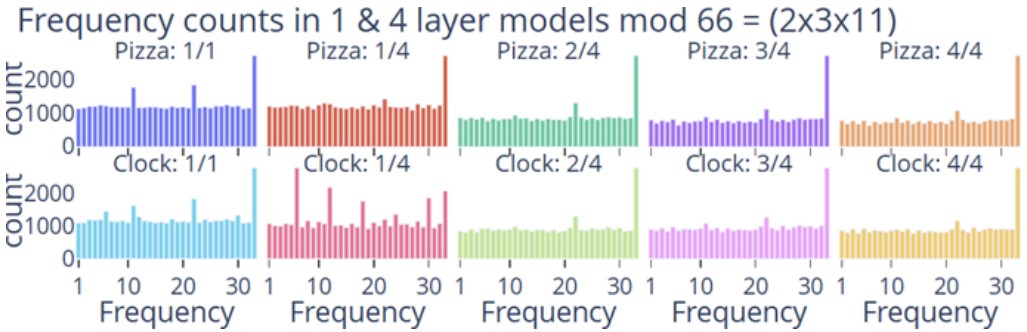

Figure 9: Histograms of the number of times each frequency was learned while training on $(a + b)$ mod 66. Note: attention in the clock models results in learning frequency 6 cosets in layer $1/4$.

## 6.2 Limitations and future work

Finding cosets key to learning modular addition aligns with Stander et al. [9]'s results on permutation groups. A group with the least structure is a cyclic group of prime order, but Cayley's theorem gives all groups are subgroups of the permutation group—the king of group structure. Despite these groups occupying upper and lower extremes of structure, our findings yield universality in the structure DNNs learn on these two disjoint tasks. This is why we believe it's likely that networks learning every group multiplication will utilize structures that can be viewed under a more general definition—*approximate cosets, involving distances on Cayley graphs*. A core limitation of our work is that we do not explore the groups between these two extremes. We believe that this is a promising avenue for future work to explore, with successful testing offering the potential to demonstrate that the universality hypothesis is true in the very diverse space of all possible group multiplications.

While we are the first to point out *how* to cause a phase transition from $\mathcal{O}(n)$ to $\mathcal{O}(\log(n))$ learned frequencies via depth or a trainable embedding (Fig. 4), we don't know *why* it occurs. Theorem 4.7 only says this solution should have great margins, and it does [7]. This gives two directions for future work. *Why* this happens: are the training dynamics wildly different? *What* causes it: does the network learn something significantly different that both we, and prior work, fail to see? The fact these are open questions—on a math task that's become very well understood over 1500 years since inception—*suggests an urgent need for new interpretability tools.* Finally, Theorem 4.7 doesn't answer how many neurons are needed per learned frequency; giving a trivial lower bound of $\Omega(1)$ neurons. Empirical results are in Appendix G.7, but don't give an obvious direction for proving bounds. *We believe answering this is necessary for the interpretability community to gain a full understanding of a task.* This may be an entire paper in itself. It looks non-trivial and requires careful arguments with non-linear ReLU activations.

### 6.2.1 Roadmap to stress testing the universality hypothesis

There's opportunity to learn a great deal more about the nature of the solutions DNNs learn via reverse engineering models trained on group multiplications (Conjecture 5.1). In particular, we believe the logarithmic efficiency to be a potentially remarkable result and hope for its generality. Due to our study being scoped to modular addition (cyclic group multiplication), and the other sample point finding cosets being permutation groups [9], we propose studies on the following tasks to fill in the blanks to compose a more general theory and understanding.

**1.** Approximate cosets were not observed in Stander et al. [9]'s study on the non-commutative permutation group multiplication. Are approximate cosets unique to commutative tasks or just cyclic groups? This can be resolved by studying the non-commutative dihedral group. **2.** Abelian group multiplication is commutative; will networks always learn to divide and conquer via approximate cosets on all Abelian groups? **3.** As cyclic and dihedral groups are both constructed from cyclic groups as the base structure, can approximate cosets be found on DNNs trained on other groups with their underlying simple group being not the cyclic group? We suspect that once these three questions are answered, it will look as though on all tasks DNNs learn a *divide-and-conquer* strategy with logarithmic efficiency (of the order of the group). If this is true (Conjecture 5.1), we believe the result will serve to help explain the efficiency of DNNs from a new perspective.

## Acknowledgements

Sihui Wei and Harley Wiltzer were of great help during the preparation of this manuscript, serving as pseudo-reviewers. We thank Arthur Ayestas-Hilgert, Colin Daniels, Darshil Doshi, and Étienne Pierre-Doray for helpful discussions. This work was supported in part by compute resources from Mila.

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

# Table of Contents for the Appendix

# A   Additional Background

## A.1   Conceptual Background and Additional Related Work

This appendix expands on the conceptual foundations and related work that motivated our approach. We first review key debates in mechanistic interpretability—particularly around universality and abstraction—before turning to mathematical tasks as ideal testbeds for studying learned structure.

**Mechanistic Interpretability, Universality and Levels of Abstraction.**

Mechanistic interpretability seeks to reverse-engineer trained neural networks by identifying the roles of individual components–such as neurons, attention heads, or MLP weights–in the model's learned function [14]. It aims to explain *how* models arrive at their outputs by analyzing the specific components and pathways involved in their internal computations. A central focus of this paradigm is on **circuits** [1, 15, 16]: small groups of components (essentially a subnetwork) that together perform a recognizable subtask such as copying, induction or composition [17]. Over time, researchers

have identified recurring patterns in these circuits–known as **motifs**–such as superposition [18] (where multiple features share the same subspace), equivariance (where computations respect certain transformations) or unioning over cases (where a unit activates for multiple distinct patterns without distinguishing between them). These recurring motifs offer generalizable insight into how networks compute [1].

A central idea in mechanistic interpretability is **universality**–the informal hypothesis that independently trained neural networks tend to develop *similar internal structures* [2, 1]. The appeal is clear: if models trained in different conditions all learn the same solution, then interpreting one model could offer insight into many. However, what "similar" means in this context has often gone unstated. Universality might refer to alignment in learned features, to similarity in neuron roles or circuits, or to shared algorithmic structure—but these possibilities are rarely distinguished. At the same time, empirical findings are mixed: while some studies report that representations in vision and language models become increasingly aligned as model scale increases [3], others show divergences in learned mechanisms even on simple algorithmic/mathematical tasks [5, 9, 19]. Together, these findings highlight a deeper issue: the field lacks a clear definition of *what kind of structure* should be expected to be universal–and *at what level of abstraction* such universality should be evaluated. In essence, a key goal of interpretability is to find the correct level of abstraction for humans to be able to intuitively understand the functionality of what was learned.

A major challenge in this area is the ambiguity of the term circuit, which can refer to anything from small neuron clusters to nearly full-network subnetworks. This flexibility enables compelling case studies, but hinders comparisons across scales. In early interpretability work, [4] introduced the *Fourier Multiplication Algorithm* (later called the Clock), showing that transformers trained on modular addition learned to represent inputs on circles and perform angle addition via attention. Chughtai et al. [8] proposed a broader generalization—the Group Composition via Representations (GCR) algorithm—suggesting a universal algorithm for group tasks based on multiplying group representations. In both cases, the term "algorithm" referred to the local computation implemented by a circuit defined over a specific Fourier frequency or irreducible representation (irrep). However, later work challenged the universality of these circuit-level "algorithms". Zhong et al. [5] found that different training settings could induce qualitatively different frequency-specific circuits (e.g., the Pizza circuit), and showed that multiple distinct frequency-based circuits could coexist within the same model. Stander et al. [9] analyzed models trained on $S_n$ and found coset-based circuits–rather than irrep-based implementations of GCR–further undermining its algorithmic universality claim. These results suggest that even when models solve the same task, they may not implement the same algorithm *at the circuit level*. Our work shifts perspective. We define a model's algorithm as a global computational strategy realized across the full network. To uncover this, we take a multiscale approach–analyzing the behavior of individual neurons, how frequency-aligned clusters of neurons work together, and how these clusters interact to form a coherent global solution. Our simple neuron model helps reconcile prior findings by showing that seemingly different circuit behaviors can be well-approximated within a unified functional form. At the cluster level, we have coset computations. At the full-network level, we identify a consistent solution that emerges across architectures and training runs: a universal abstract algorithm, formalized as an approximate version of the Chinese Remainder Theorem (aCRT). This perspective explains how models can converge to the same high-level structure even when their lower-level mechanisms diverge. Unlike prior analyses focused on isolated subcircuits, we sought to understand how computation emerges across scales–from individual neurons to full-model solutions–and show that models can share the same network-level algorithmic structure despite mechanistic variability.

Much of the confusion around universality stems from comparing models without distinguishing between different **levels of abstraction**. Recent work has proposed frameworks from cognitive science—especially Marr's levels of analysis [20–23]—as useful tools in interpretability, helping clarify what kind of explanations are being offered. Marr distinguishes between (1) the computational level (what problem is solved), (2) the algorithmic level (how it is solved), and (3) the implementational level (how it is physically realized). Although not developed for studying universality and not formally used in our main analysis, we find these ideas helpful as a retrospective lens—both for understanding why prior analyses disagreed—and for identifying common structure where others saw divergence. For example, Vilas et al. [23] propose looking for invariances across levels and enforcing mutual constraints between levels as guiding principles. Our approach reflects both: we uncover a consistent computational-level (in the sense of Marr's levels) solution, the aCRT, that

unifies divergent circuit-level behaviors and reveals a form of universality that holds at a more abstract level than previously recognized—what we refer to as the universal abstract algorithm.

**Mathematical and Algorithmic Tasks as Interpretable Testbeds for Machine Learning.**

Mathematical and algorithmic tasks—such as modular addition, group operations, sorting, and Markov chains—have become valuable testbeds for studying machine learning systems. Their appeal lies in their formal structure: these problems have been studied for centuries, with well-understood properties. Because the task structure is fully known, optimal solutions are analyzable and generalization behavior can be sharply characterized—unlike in typical natural data settings. This makes them ideal environments for probing what neural networks learn and how. While primarily used to study model internals, these tasks also have practical relevance; for example, transformers trained on modular arithmetic have been applied to attack lattice-based cryptographic schemes [24].

These tasks have been central to studying grokking, the phenomenon where models generalize abruptly after a period of overfitting [11]. Using modular addition as a testbed, researchers have linked grokking to structured internal representations [25, 6], margin maximization [12, 7], label corruption [13], and non-neural architectures [26], or how distinct circuits emerge during grokking for different modular arithmetic tasks [27]. Other work has connected grokking to training phase transitions, such as shifts between lazy and rich regimes in modular addition [28] and polynomial regression [29]. Several papers also provide exact analytical constructions of specific network weights: one-layer, one-hot encoded networks with quadratic activations solving modular addition [6, 7], and ReLU networks solving modular multiplication for modeling grokked solutions in modular polynomials [30]. Stepping beyond modular addition to a more general Abelian group, the activations of DNNs trained on elementary $p$ group multiplication have been studied as well [31].

Beyond grokking, mathematical tasks have helped probe generalization, learning dynamics, and in-context learning. Modular addition and Markov chains have served as controlled environments for studying how transformers acquire in-context learning capabilities [32, 33]. Other work has shown that repeating training examples affects generalization in tasks like greatest common divisor (GCD), modular multiplication, and matrix eigenvalue prediction [34], and that arithmetic tasks shed light on how transformers handle length generalization [35, 36]. In the GCD setting specifically, models appear to select from a small, learned set of candidate divisors [37]. Transformers trained in the context of enumerative geometry have also been studied [38].

While most research in this area focuses on supervised learning, some work investigates how reinforcement learning (RL) agents operate in mathematical environments. Agents trained on tasks like matrix multiplication or sorting have been observed to discover novel, interpretable algorithms [39, 40]. Other work leverages group-theoretic structure to enable exact analysis: environments built using the temporal symmetries of affine Weyl groups allow analytical characterization of the policy gradient landscape, yielding closed-form gradient dynamics and local optima and providing insight into how exploration difficulty affects learning [41]. Similarly, interpretable RL environments based on Erdos-Selfridge-Spencer games have been developed, with exact optimal strategies and tunable difficulty controlled by human-interpretable environment parameters [42].

Together, this body of work establishes mathematical tasks as invaluable tools for interpretability. Their known structure enables precise analysis of learned behavior, supports abstraction-driven explanations, and provides testbeds where claims about generalization and universality can be rigorously evaluated.

## A.2 Additional Mathematical Background

This section provides formal definitions and examples of the group-theoretic structures that underlie our analysis of modular addition networks: groups, cosets, Cayley graphs, group representations, and the Chinese Remainder Theorem (CRT). These definitions support the structures described in the main text: simple neurons, approximate cosets and the approximate CRT algorithm we identify in trained networks.

**Groups, Subgroups and Cosets.**

**Definition A.1** (Group). A **group** $(G, \circ)$ consists of a set $G$ equipped with a binary operation $\circ : G \times G \to G$ satisfying:

    1. **Associativity**: $(f \circ g) \circ h = f \circ (g \circ h)$ for all $f, g, h \in G$.

2. **Identity**: There exists an element $e \in G$ such that $e \circ g = g \circ e = g$ for all $g \in G$.

3. **Inverses**: For each $g \in G$, there exists $g^{-1} \in G$ such that $g \circ g^{-1} = e$.

**Definition A.2** (Subgroup). A subset $H \subseteq G$ is a **subgroup** if it is itself a group under the same operation $\circ$. That is, $H$ must contain the identity, be closed under the operation, and contain inverses of its elements.

Subgroups induce a natural partitioning of the group via *cosets*, which are key to understanding modular structure and factorization.

**Definition A.3** (Cosets). Let $H$ be a subgroup of $G$, and let $g \in G$. The **left coset** of $H$ with representative $g$ is:
$$gH = \{g \circ h : h \in H\}.$$
Right cosets are defined similarly: $Hg = \{h \circ g : h \in H\}$. Cosets partition $G$ into disjoint, equally sized subsets.

**Example A.4** (Integers and Even/Odd Cosets). The set $(\mathbb{Z}, +)$ is a group. The identity is $0$; the inverse of $n$ is $-n$. The even integers $2\mathbb{Z}$ form a subgroup. This gives two cosets:
$$2\mathbb{Z} = \{\ldots, -2, 0, 2, \ldots\}, \quad 1 + 2\mathbb{Z} = \{\ldots, -1, 1, 3, \ldots\}.$$
These correspond to the even and odd integers — a familiar example of partitioning via cosets.

**Definition A.5** (Homomorphism). A map $\psi : G \to H$ between groups is a **homomorphism** if it preserves the group operation:
$$\psi(g_1 \circ_G g_2) = \psi(g_1) \circ_H \psi(g_2), \quad \text{for all } g_1, g_2 \in G.$$
If $\psi$ is also bijective, it is called a **group isomorphism**.

Homomorphisms are the natural notion of "structure-preserving" maps between groups.

**Cyclic Groups and Modular Arithmetic.**

In this paper, we focus on modular addition, which forms the cyclic group.

**Definition A.6** (Cyclic Group $C_n$). The **cyclic group of order** $n$, denoted $C_n$ or $\mathbb{Z}_n$, is the set $\{0, 1, \ldots, n-1\}$ equipped with addition modulo $n$. The group operation is defined by
$$a \circ b = (a + b) \mod n,$$
with identity element $0$ and inverses given by $a^{-1} = (n - a) \mod n$.

Subgroups of $C_n$ correspond to evenly spaced subsets, and their cosets partition $C_n$ into congruence classes modulo a divisor of $n$.

**Example A.7** (Cosets mod 4 in $\mathbb{Z}_8$). Let $n = 8$, and consider the subgroup $H = \{0, 4\}$. The left cosets are:
$$0 + H = \{0, 4\}, \quad 1 + H = \{1, 5\}, \quad 2 + H = \{2, 6\}, \quad 3 + H = \{3, 7\}.$$
This partitions $\mathbb{Z}_8$ into four disjoint cosets of size 2.

**Definition A.8** (Modular Inverse). Let $a \in \mathbb{Z}_n$. The **modular inverse** of $a$ is an element $b \in \mathbb{Z}_n$ such that
$$a \cdot b \equiv 1 \mod n.$$
A modular inverse exists if and only if $\gcd(a, n) = 1$, i.e., $a$ and $n$ are coprime.

**The Chinese Remainder Theorem.**

The Chinese Remainder Theorem (CRT) gives a powerful way to decompose modular arithmetic over a large modulus into multiple, independent modular systems over smaller, coprime moduli. This decomposition mirrors the modular structure learned by networks trained on addition tasks, and it is central to our concept of the *approximate CRT*.

**Theorem A.9** (Chinese Remainder Theorem). *Let $n = q_1 q_2 \cdots q_k$ be a product of pairwise coprime integers. Then the map*
$$\phi : \mathbb{Z}_n \to \mathbb{Z}_{q_1} \times \cdots \times \mathbb{Z}_{q_k}, \quad x \mapsto (x \bmod q_1, \ldots, x \bmod q_k)$$
*is a group isomorphism. That is, each element in $\mathbb{Z}_n$ corresponds uniquely to a tuple of residues modulo the $q_i$, and vice versa.*

**Example A.10** (Coset Intersection View). Let $n = 91 = 7 \cdot 13$. Suppose we want to solve:

$$x \equiv 3 \mod 7, \quad x \equiv 10 \mod 13.$$

Each congruence defines a coset:

$$x \equiv 3 \mod 7 \Rightarrow \{3, 10, 17, 24, 31, \dots\},$$
$$x \equiv 10 \mod 13 \Rightarrow \{10, 23, 36, 49, \dots\}.$$

The unique solution mod 91 is the number common to both cosets: $\boxed{10}$.

**Cayley Graphs.**

**Definition A.11** (Generating Set). Let $G$ be a group. A subset $S \subseteq G$ is called a **generating set** of $G$ if every element $g \in G$ can be written as a finite product of elements from $S$ and their inverses. That is, for all $g \in G$, there exist $s_1, \dots, s_k \in S$ and signs $\epsilon_i \in \{-1, 1\}$ such that:

$$g = s_1^{\epsilon_1} s_2^{\epsilon_2} \cdots s_k^{\epsilon_k}.$$

**Example A.12** (Generating Sets in $\mathbb{Z}_6$). The group $\mathbb{Z}_6 = \{0, 1, 2, 3, 4, 5\}$ under addition mod 6 can be:

- Generated by $\{1\}$, since repeated addition gives all elements: $1, 2, \dots, 5, 0$.

- Generated by $\{5\}$, since $5 + 5 = 10 \equiv 4 \mod 6$, and so on.

- Not generated by $\{2\}$, since $2 + 2 = 4, 4 + 2 = 0$, and the subgroup $\{0, 2, 4\}$ is too small.

**Definition A.13** (Cayley Graph). Let $G$ be a group and let $S \subseteq G$ be any subset (not necessarily a generating set). The **Cayley graph** $\Gamma(G, S)$ is a directed graph with one vertex for each element of $G$, and a directed edge from $g$ to $g \cdot s$ for every $s \in S$.

If $S$ is symmetric (i.e., $s \in S \Rightarrow s^{-1} \in S$), then the graph is often treated as undirected. The graph is:

- **Connected** if and only if $S$ generates $G$.

- **Disconnected** if $S$ generates a proper subgroup of $G$.

**Example A.14** (Cayley Graphs in $\mathbb{Z}_6$). Let $G = \mathbb{Z}_6 = \{0, 1, 2, 3, 4, 5\}$ under addition mod 6.

- Using $S = \{1\}$, the graph connects:

$$0 \to 1 \to 2 \to 3 \to 4 \to 5 \to 0$$

forming a connected 6-cycle. Here, $S = \{1\}$ is a generating set.

- Using $S = \{2\}$, we only get:

$$0 \to 2 \to 4 \to 0, \quad 1 \to 3 \to 5 \to 1$$

which are two disconnected 3-cycles. This reflects the subgroup $\{0, 2, 4\}$ and its coset $\{1, 3, 5\}$. The set $S = \{2\}$ does *not* generate $\mathbb{Z}_6$, and the graph is *disconnected*.

**Group representations and the Discrete Fourier Transform (DFT)**

**Definition A.15** (Group representation). A **representation** of a group $G$ on a vector space $V$ is a homomorphism $\rho : G \to \mathrm{GL}(V)$, where $\mathrm{GL}(V)$ is the group of invertible linear maps on $V$.

**Example A.16** (Discrete Fourier Transform). The **discrete Fourier transform (DFT)** comes from the complex representations of the cyclic group $C_n$. Each element $j \in C_n$ is mapped to the complex number $\exp\left(2\pi i \frac{kj}{n}\right)$ for integer $k \in \{0, 1, \dots, n-1\}$, encoding modular structure as rotations in the complex plane. These complex representations correspond to the DFT. They also induce real-valued representations as 2D rotation matrices acting on $(\cos, \sin)$ components.

### A.2.1  Examples: cosets, Cayley graphs, step size $d$

In the background (Section 3) we gave examples of cosets on $C_6$. In this section we will show these examples visually to help the reader better understand approximate cosets, generating Cayley graphs, and labeling the vertices on them with the step size (Definition 4.1). Recall, the step size: $d := (\frac{f}{\gcd(f,n)})^{-1}(\mathrm{mod}\,\frac{n}{\gcd(f,n)})$, where the modular inverse is used. There are $n' = \frac{n}{\gcd(f,n)}$ distinct positions reachable in this way, thus we could write: $d := (\frac{f}{\gcd(f,n)})^{-1}(\mathrm{mod}\,n')$. Note: $d$ is the step size in the graph $C_{n'}$.

We consider the cyclic group on 6 elements, $C_6$. In this example, we will show how cosines of different frequencies encode coset information and how we can construct the corresponding Cayley graphs from this. Since $n = 6$, then $\lfloor \frac{6}{2} \rfloor = 3$, and we have 3 possible frequencies $f = 1, 2, 3$ for $\cos\left(\frac{2\pi \cdot f \cdot a}{6}\right)$ across $a \in C_6$. These three cosines will each step around $C_6$ in different ways, depending on $f$.

If $f = 1$, our cosine is $\cos\left(\frac{2\pi \cdot 1 \cdot a}{6}\right)$. We compute $n' = 6$, thus there are 6 elements we can reach in $C_6$. This gives 6 cosets of size 1:

$$\{0\}, \{1\}, \{2\}, \{3\}, \{4\}, \{5\}.$$

If $f = 2$, our cosine is $\cos\left(\frac{2\pi \cdot 2 \cdot a}{6}\right)$. We compute $n' = 3$, thus we're in $C_3$, which has 3 elements. This gives 3 cosets, each of size 2, corresponding to three different 2-cycles in the original $C_6$ graph, but where each coset is now a point in our new graph for $C_{n'} = C_3$:

$$\{0, 3\}, \{1, 4\}, \{2, 5\}.$$

If $f = 3$, our cosine is $\cos\left(\frac{2\pi \cdot 3 \cdot a}{6}\right)$. We compute $n' = 2$, thus we're in $C_2$, which has 2 elements. This gives 2 cosets, each of size 3, corresponding to two different 3-cycles in the original $C_6$ graph, but where each coset is now a point in our new graph for $C_{n'} = C_2$:

$$\{0, 2, 4\}, \{1, 3, 5\}.$$

See these cosines in Figure 10.

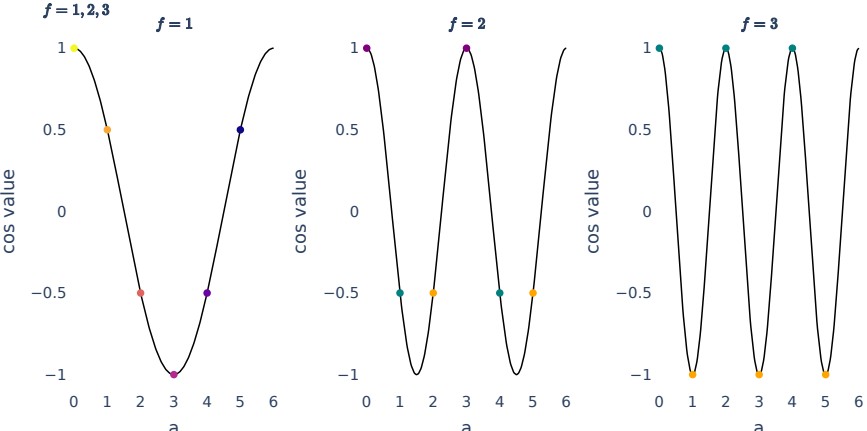

Figure 10: Cosine functions centered at 0, with $f = 1$, $f = 2$, $f = 3$. The points are colored based on their coset membership, *i.e.* equivalence class.

We can calculate the step size, Definition 4.1, and in all three cases we get $d = 1$. We will visualize these sets in a figure coming shortly.

Row 1 shows the original Cayley graph and row 2 shows the new graph after collapsing cosets. Row 1 will be the cyclic group on 6 elements, $C_6$: in other words, we will not collapse the cosets yet into their equivalence classes. This means we aren't making the graphs $C_{n'}$ and resultantly, we plot $C_6$ with distances from the vertices to the first vertex $a = 0$. We also show the cosets, the 3-cycles

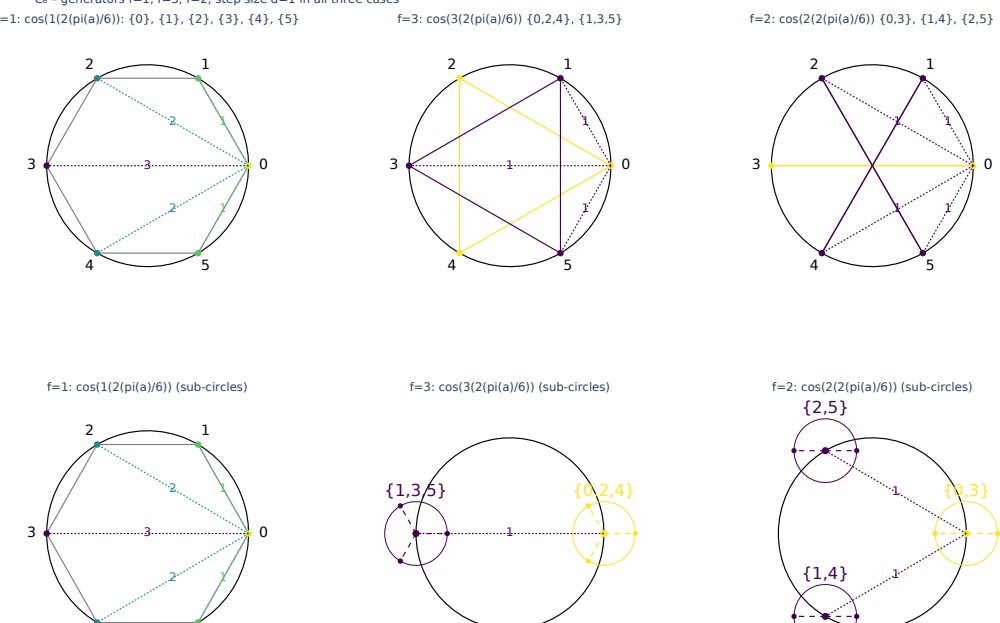

Figure 11: In row 1 we show the Cayley graphs and the distances, which is a bit confusing in panel 3 because we don't collapse the cosets into their equivalence classes. To make it easier to see the distances and where they come from, in row 2 we collapse equivalent points (all points in a coset) into a subcircle. Doing this gives us less points on the graph, giving us $C_{n'}$, which is the graph $d$ is the step size on. It is now easier to see why the distances to approximate cosets are what they are. In row 2 panel 3, we chose to put the two darkly colored cosets on the other side of the circle to emphasize that ReLU is 0 on them.

and 2-cycles. In row 2, we present an equivalent picture that makes things clearer: we plot $C_{n'}$. Furthermore, the step size $d$ is actually the step size in this cyclic group, $C_{n'}$, making the distances easier to see compared to looking at $C_6$. By collapsing vertices into their cosets (equivalence classes), it becomes clear that everything is distance 1 from the yellow coset. We will do this in row 2, giving $C_6$ corresponding to $f = 1$, $C_3$ corresponding to $f = 2$ and $C_2$ corresponding to $f = 3$. See this in Figure 11.

In the case of approximate cosets, nothing changes. Calculate $n'$, calculate $d$, and step around the cyclic group $C_{n'}$ using $d$, labeling the distances of elements. To build an approximate coset, for some multiples $k_1, k_2$, take $k_1$ steps backward and $k_2$ steps forward. The path is the approximate coset—all elements that are "close" on the Cayley graph.

### A.2.2 An example for how the network gets the correct answer

**Simplified Divide and Conquer.** Consider $(a + b) \bmod 6 = c$.

There are two coset types:

mod 2:

$x \equiv 0 \bmod 2 : \{0, 2, 4\}$
$x \equiv 1 \bmod 2 : \{1, 3, 5\}$

mod 3:

$x \equiv 0 \mod 3 : \{0, 3\}$
$x \equiv 1 \mod 3 : \{1, 4\}$
$x \equiv 2 \mod 3 : \{2, 5\}$

Let neuron-1 have frequency 3 and activate when the answer $c \in \{0, 2, 4\}$, *i.e.* it activates on this coset of $c$. It learns:

$f_1(a, b) = \cos(\frac{6\pi a}{n}) + \cos(\frac{6\pi b}{n})$

This neuron activates maximally when both $a, b \in \{0, 2, 4\}$, guaranteeing $c \in \{0, 2, 4\}$.

Let neuron-2 have frequency 2 and activate when $c \in \{0, 3\}$ . It learns:

$f_2(a, b) = \cos(\frac{4\pi a}{n}) + \cos(\frac{4\pi b}{n})$

This function activates maximally when $a, b \in \{0, 3\}$, ensuring $c \in \{0, 3\}$.

Summing the outputs of these two neurons gives the following logits:

$c = 0 \rightarrow 4$
$c = 1 \rightarrow 0$
$c = 1 \rightarrow 0$
$c = 2 \rightarrow 2$
$c = 3 \rightarrow 2$
$c = 4 \rightarrow 2$
$c = 5 \rightarrow 0$

The argmax operation now correctly selects $c = 0$.

Suppose alternatively, that neuron-3 activates for $c \in \{1, 4\}$ (e.g., by firing when $a \in \{0, 3\}$ and $b \in \{1, 4\}$). Then if neuron-1 (firing on $c \in \{0, 2, 4\}$) and neuron-3 activate simultaneously, the argmax would select $c = 4$ due to the higher logit value. This logic continues as you add neurons corresponding to every coset.

This demonstrates a divide-and-conquer strategy: each neuron rules out a large fraction of incorrect outputs, analogously to how binary search eliminates half the search space at each step. This example is with mod 6 and exact cosets. Approximate cosets generalize exact cosets by capturing cases where neurons learn frequencies that are not divisors of the modulus. Theorem 4.7 shows approximate cosets are sufficient for networks to attain strong margins, matching the cosets required by the classical Chinese Remainder Theorem.

As $n$ grows, the number of coset types is logarithmic, *e.g.*

$\mod 6 \rightarrow 2$ coset types
$\mod 3628800 \rightarrow 10$ coset types

## B  Proof of Theorem 4.4

For empirical evidence supporting this theorem, see Figure 2 in the main paper. Every point $> 0$ is in the approximate coset colored by viridis colors, with strength of viridis decaying as the point gets farther from the center element of the approximate coset (an element getting closer to where ReLU won't activate, means it less bright).

In reality, all sinusoidal functions, *i.e.* our simple neuron assumption, will satisfy this theorem. The simplicity of this proof therefore results from the fact that we came up with a very powerful definition for approximate cosets that actually reflects what neurons in the network are learning. This proof requires the assumption that neurons learn periodic functions, and this is why the majority of the paper was dedicated to empirically proving that simple neurons are indeed learned by networks.

*Proof.* **Simple neurons learn approximate cosets.** A neuron satisfying the simple neuron model computes a trigonometric function that has its maxima on the elements of a coset or "approximate coset". If $g := \gcd(f, n) > 1$, the neuron has learned the **coset** of order $g$ containing $s_A + s_B$. More precisely: writing $n = n'g$ and $f = f'g$ for $g = \gcd(f, n)$, we can rewrite $\frac{2\pi f}{n} = \frac{2\pi f'}{n'}$. So if the

input neurons are at positions $a$ and $b$ where $a \equiv s_A \pmod{n'}$ and $b \equiv s_B \pmod{n'}$, then the activation of the neuron has a maximum: $\cos \frac{2\pi f(a-s_A)}{n} = \cos \frac{2\pi f(b-s_B)}{n} = 1$. The neuron points most strongly to every logit satisfying $c \equiv s_A + s_B \pmod{n'}$, because for all such output logits $\cos \frac{2\pi f(c-s_A-s_B)}{n} = 1$. We see that the neuron strongly associates elements of $C_n$ that are congruent modulo $n'$.

Whether $f$ is a divisor of the order $n$ or not, the neuron will activate on what we defined as an approximate coset. More precisely, we can ask the following: for $a \not\equiv s_A \pmod{n'}$, which values of $a$ have the largest activation? We have $\cos \frac{2\pi f'(a-s_A)}{n'}$ very close to 1 if and only if $f'(a - s_A)$ is very close to an integer multiple of $n'$; that is, say, $f'(a - s_A) \equiv m \pmod{n'}$ for some integer $m$ with small absolute value. Letting $d$ denote a modular inverse of $f'$ mod $n$, this is equivalent to $a - s_A \equiv dm \pmod{n'}$. In other words, by taking $a = s_A + dm$ for small integers $m$, the neuron will be activated very strongly. Likewise if $b = s_B + dm'$ for some other small integer $m'$. Now this neuron will point most strongly to $c \equiv s_A + s_B \pmod{n'}$ as discussed above, but if $c$ is a small number of steps of size $d$ away from $s_A + s_B$, it will still have large activation. To summarize: if you can reach each of $a, b, c$ via a small number of steps of size $d$ from $s_A, s_B, s_A + s_B$, respectively, then $N$ fires strongly on inputs $a, b$, and points strongly at $c$.

After ReLU, it follows that since all neurons output (activate) only on approximate cosets, all neurons in the following layers activate on linear combinations of approximate cosets (follows from networks being fully connected, and ReLU only having the potential to make the cosets smaller, i.e. elements below 0 are cut off). $\qquad\square$

## C   Proofs and details for Theorem 4.7 and Corollary 4.8

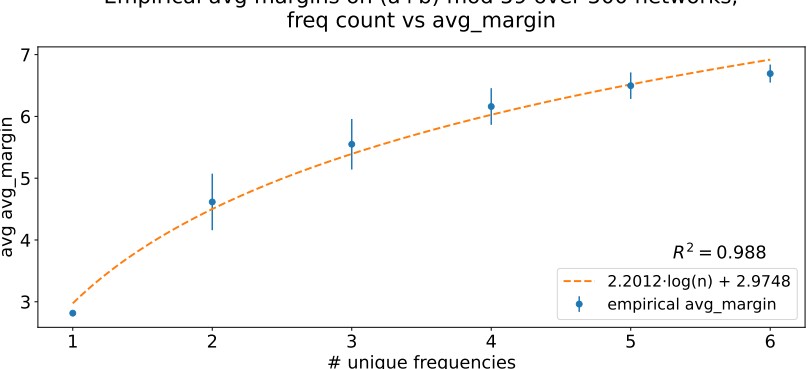

Figure 12: Empirically, 1 hidden layer, 1 trainable embedding matrix, networks have margins grow like $\mathcal{O}(\log(n))$ as more frequencies are learned. In Corollary 4.8 we prove it. Note: 1 std dev error bars are on (x=1, y=2.82); they are just small, ranging from 2.8-2.84. This plot is made by computing the average margin of each network across the full dataset, given the network had learned $x =\#$ unique frequencies. The plot shows, for networks that learned the same number of frequencies, the average average margin and the 1 std dev error bars of this data.

Assume that training results in neurons learning the simple neuron model.

**Theorem C.1.** *Suppose the integer number of distinct frequencies $m$ and reals $0 < \rho, \delta < 1$ satisfy the inequality*

$$ m > \frac{2 \log_e n - 2 \log_e(2 - 2\rho)}{\log_e(\pi/\delta) - 1}. $$

*Then, with probability at least $\rho$, for all $k \neq i + j \mod n$, we have $m' - h_m(k) > \delta m'$.*

*Proof.* Each simple neuron maximally activates a single output, namely $s_A + s_B$, (or possibly maximally activates on a coset of outputs containing $s_A + s_B$). However, if we combine the

contributions from all simple neurons in a single cluster (i.e. all with the same frequency $f$), we observe that the activation level can be maximized at any desired output; more precisely, the activation level at output $k$ given inputs $i, j$ will be of the form $A \cos(2\pi(f(k - i - j)/n))$. Note this has been observed experimentally e.g. see Fig 26(a) and Fig. 44, and has also been previously noticed in the literature: see e.g. the last equation of Section 3 in [10]. In fact, the analysis below still works even if $i + j$ is only somewhat close to the maximal activation of the cluster (see Figure 44), though we assume the maximum is at $i + j$ for simplicity. However, even in this case there will be many output logits that all activate nearly as strongly on the correct answer. To isolate a single answer, we use a superposition of sine waves of multiple frequencies; we observe experimentally in Figure 44 that this process makes the correct answer stand out from the rest.

In light of the above and Section 4.1, if we fix inputs $i, j \in C_n$ then combining the contributions from all clusters (and assuming for the heuristic that the contributions from all clusters have the same amplitude), the sum

$$h_m(k) := \sum_{\ell=1}^{m} \cos\left(\frac{2\pi f_\ell}{n}(k - i - j)\right)$$

gives a model for the activation energy at output logit $k$. If $k = i + j$, then $h_m(k)$ takes on the maximum value $m$. If we want to guarantee the neural net will consistently select $k$, we need to show that $h_m(k)$ is significantly less than $m$ for all other values of $k$. We'll assume a random model where $m$ frequencies $f_1, \ldots, f_m$ are chosen uniformly at random from $1, 2, \ldots, n-1$. Fix a parameter $0 < \delta < 1$; we will compute an approximation for the probability that $m - h_m(k) > \delta m$ for all $k \neq i + j \pmod{m}$.

Let $\{x\} := x - \lfloor x + \frac{1}{2} \rfloor$ be the signed distance to the nearest integer and set $d := k - i - j$. Then using a Taylor expansion,

$$m - h_m(k) = m - \sum_{\ell=1}^{m} \cos\left(2\pi\left\{\frac{f_\ell d}{n}\right\}\right) \approx m - \sum_{\ell=1}^{m}\left(1 - \frac{1}{2}\left(2\pi\left\{\frac{f_\ell d}{n}\right\}\right)^2\right) = 2\pi^2 \sum_{\ell=1}^{m}\left\{\frac{f_\ell d}{n}\right\}^2.$$

(Note that the Taylor approximation is quite bad when $f_\ell d/n$ is far from an integer, and if $k$ is close to an integer then $\{k\}$ is close to 0. It is reasonable to expect that $m - h_m(k)$ will be minimized when the values $f_\ell d/n$ are all close to integers, in which case the approximation is more accurate. (Note, getting a rigorous proof instead of this toy model will require more precise bounds.)

Thus the condition $m - h_m(k) > \delta m$ is related to the following condition: defining the vector $v := \frac{1}{n}(f_1, \ldots, f_m) \in [0, 1]^m$, we need that for all $1 \leq d \leq n - 1$, the point $dv$ has distance at least $\sqrt{\delta m/2\pi^2}$ away from any point in $\mathbb{Z}^m$. Note that $nv$ is an integer point, so $(n - d)v$ is always the same distance from an integer point as $dv$ is. Thus it suffices to require $v$ to be at least $\frac{1}{d}\sqrt{\delta m/2\pi^2}$ away from a point in $\frac{1}{d}\mathbb{Z}^m$ for $d = 1, \ldots, \lfloor n/2 \rfloor$.

We compute an upper bound on the volume of the region to be avoided: that is, the set of all points in $[0, 1]^m$ within $\frac{1}{d}\sqrt{\delta m/2\pi^2}$ of a point of $\frac{1}{d}\mathbb{Z}^m$ for some $d = 1, \ldots, n/2$. For each $d$, there are $d^m$ points in this region, and each has a ball of radius $\frac{1}{d}\sqrt{\delta m/2\pi^2}$ around it; the total volume of the region to be avoided is therefore bounded above by $\frac{n}{2\Gamma(m/2+1)}\left(\frac{\delta m}{2\pi}\right)^{m/2}$. Thus the probability that $m - h_m(k) > \delta m$ is approximately equal to 1 minus this value.

For a given $n$, let's compute the value of $m$ that makes this probability greater than, say, $\rho$.

$$1 - \frac{n}{2\Gamma(m/2 + 1)}\left(\frac{\delta m}{2\pi}\right)^{m/2} > \rho \iff \Gamma(m/2 + 1)\left(\frac{2\pi}{\delta m}\right)^{m/2} > \frac{n}{2 - 2\rho}.$$

Taking a natural logarithm, applying Stirling's approximation $\log_e \Gamma(x + 1) \approx x \log_e(x) - x$, and solving for $m$,

$$\log_e \Gamma(m/2 + 1) + \frac{m}{2}\log_e\left(\frac{2\pi}{\delta m}\right) > \log_e n - \log_e(2 - 2\rho)$$

$$m > \frac{2\log_e n - 2\log_e(2 - 2\rho)}{\log_e(\pi/\delta) - 1}.$$

Thus if the number of neuron clusters $m$ is greater than this expression, then with probability at least $\rho$, the separation $m - h_m(k)$ will be at least $\delta m$. We see that the number grows linearly in $\log_e n$.

Choosing the parameters $\rho$ and $\delta$ can significantly change the precise value of $m$ needed, and it's not clear which values most accurately model the true behavior of the neural net.

As an example, note that if we take $\delta = \pi/e^3 \approx 0.1564$, and $\rho = \frac{1}{2}$, then this whole expression simplifies to just $m > \log_e n$. Thus, if the neural net uses $m = \log_e n$ neuron clusters, then this heuristic predicts that it will guarantee a separation $m - h_m(k) > 0.15m$ for all $k \neq i + j$ with 50% certainty. For $n = 89, 91$ we have $\log_e n \approx 4.5$, which agrees with the number of clusters found in Figures 40 and 3. This process can be interpreted as the approximate CRT; see Remark 4.5 for the analogy. $\qquad\square$

## C.1 Proof of Corollary 4.8

Recall $h$ is the model network evaluated at input $i, j$, $h_m(k)$ is the value of the output distribution at $k$ with maximum value $h_m(i + j) = m$ and $m$ is the number of distinct frequencies simple neurons of the network.

**Corollary C.2** (Logarithmic number of frequencies suffices for a non-trivial margin). *Let $0 < \delta < 1$ and $0 < \rho < 1$ and define*

$$C(\delta, \rho) \;=\; \frac{2}{\log(\pi/\delta) - 1}\left(1 - \tfrac{1}{2}\log_e(2 - 2\rho)\right) \quad (> 0).$$

*If the number $m$ of simple neurons of the network satisfies*

$$m \;\geq\; C(\delta, \rho)\, \log n$$

*then with probability at least $\rho$*

$$h_m(i + j) - h_m(k) \;>\; \delta\, m,$$

*i.e. the logit margin is $\Omega(\log n)$.*

*Proof.* Theorem 4.7 states that the inequality

$$h_m(i + j) - h_m(k) \;>\; \delta m \tag{†}$$

holds for all $k \neq i + j$ with probability at least $\rho$ provided

$$m \;>\; \frac{2\log_e n - 2\log_e(2 - 2\rho)}{\log_e(\pi/\delta) - 1}. \tag{‡}$$

Choose the constant $m$ so $m \geq C(\delta, \rho)\log n$ and inequality (‡) both hold.

Finally, the guaranteed margin $\delta m \geq \delta C(\delta, \rho)\log n$ is $\Omega(\log n)$, which is strictly larger than the $O(1)$ margin attained with only a constant number of frequencies ("minimal margin"). $\qquad\square$

## D  Embeddings contain projections of representations, not representations

Chughtai et al. [8] discover representation values in the embedding matrix. The first step in their GCR algorithm is not true in general. They state, "Translates one-hot $a$, $b$ to representation matrices". We provide evidence against this by training with a mini-batch size equal to the modulus $n$ and training with a full batch size. See the difference in the distribution of the resulting embedding matrices in Fig. 13. Furthermore, neurons in a cluster of frequency $f$ have different phase shifts, and $2 \times 2$ rotation matrices in the embeddings doesn't suffice to explain this behaviour.

Instead, the values found in the embedding matrix may encode scaled projections of a $2 \times 2$ rotation matrix onto a one dimensional subspace. Note that such structure is implied by the hypothesis that neural networks trained on group tasks learn representations, but is more general because of the existence of both amplitude and phase shifts. To get an exact equivalence, we note that this neuron structure can be obtained by an *arbitrary scaled projection* of representations. Suppose

$$\rho(k) = \begin{pmatrix} \cos(2\pi f k/n) & -\sin(2\pi f k/n) \\ \sin(2\pi f k/n) & \cos(2\pi f k/n) \end{pmatrix}$$

is a $2 \times 2$ matrix representation of $C_n$. If we apply $\rho(k)$ to the vector $(1,0)$ and then take the dot product with $(\alpha \cos(2\pi f s_a/n), -\alpha \sin(2\pi f s_a/n))$ (which is the same as projecting onto the subspace spanned by this vector and scaling by $\alpha$) we obtain exactly

$$\alpha \cos \tfrac{2\pi fk}{n} \cos \tfrac{2\pi f s_A}{n} + \alpha \sin \tfrac{2\pi fk}{n} \sin \tfrac{2\pi f s_A}{n} = \alpha \cos \tfrac{2\pi f(k-s_A)}{n} = \alpha w(A_k, N).$$

Thus we have explained the phase shifts of different neurons in a cluster, and shown that it's not just the components of $\rho(k)$ that appear in the embeddings, but rather scaled projections of the representations onto arbitrary 1-d subspaces. In our model of simple neurons we ignore the amplitude to make the analysis simpler, but in general it does need to be included. See Fig 28 for example where the amplitudes are greater than 2.

**Inspecting the distribution of embedding matrix weights.** Contrary to findings by [4, 8], we did not observe the $2{\times}2$ representation matrix values (used to encode rotations) in our embedding matrices outside their reported training conditions. As shown in Fig. 13, the distribution of embedding weights varies significantly between small and full batch size and the tails of the distributions are quite different. In the case of small batch size, numbers can be found in the range (-2, 2), whereas large batch size contains numbers between (-1.5, 1.5). Note that we choose to remove weights that are between (-0.025, 0.025) to make it easier to see the tails of the distribution; this was done due to 2.4million weights occurring within this range when training with the small batch size. Specifically, in the small batch size regime, around 5% of the weights fell outside the interval $[-1, 1]$, including some weights larger than 2. These values are not consistent with rotation matrix entries. Other than this, we could not identify any significant differences in the core structures of what the neural net learns between the batch sizes.

Combining these experimental findings (Fig. 13) with this model (see D) explains that the embedding matrices may contain scaled projections of representations. This explains the different shifts in the periodic functions that can be seen in Figs 26(a), 26(b) and 24, which GCR [8] fails to explain.

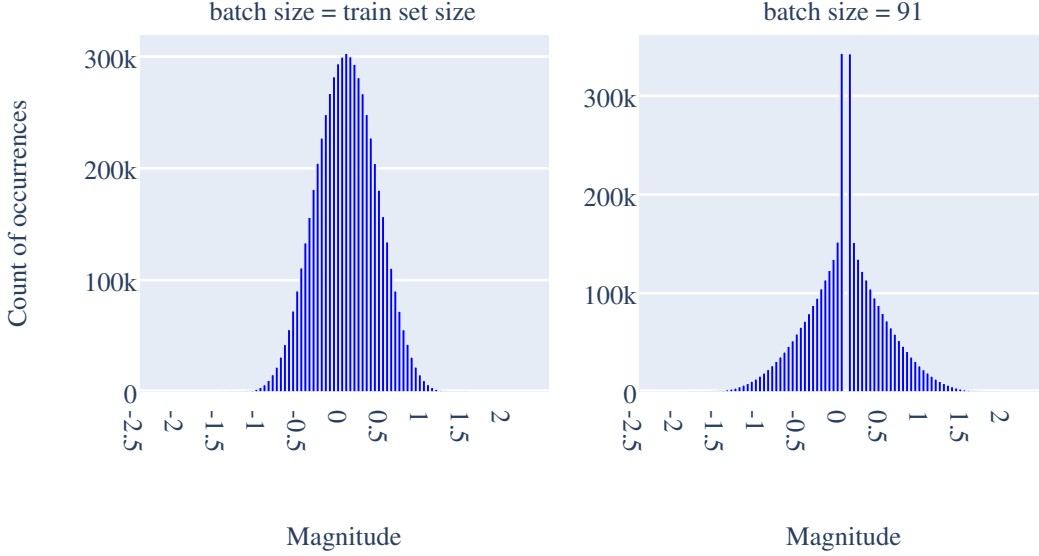

Figure 13: The histograms of embedding weight magnitudes found across 10k random seeds for mod 91 provide evidence against rotation matrices. With batch-size 91 about 5% of the weights are $> 1$ or $< $ -1, whereas when the batch size is the training set size fewer than 0.5% of the weights are $> 1$ or $<$ -1. The bin with 0 was removed for batch size 91 due to so many dead weights obfuscating the plot. The value was 2.4 mil, implying that small batches find sparse embeddings with larger magnitude weights.

# E   Main body experimental details

The train and test splits are 90% and 10% respectively, unless stated otherwise. The optimizer used to update the neural networks weights is always Adam [43].

The only plot with error bars in the main text is Figure 3, which uses 1 standard deviation (std dev) error bars. If a plot has error bars, they are 1 std dev.

## E.1 Figure 1

The point of this figure is to show the reader that qualitatively, neurons are learning sinusoidal functions that are identical after normalization, even when secondary frequencies exist in the Discrete Fourier Transforms. Furthermore, it serves to immediately show the reader that the "remapping: normalizing to frequency 1" definition makes a sinusoidal function have frequency 1. The figure is very easy to generate, just grab arbitrary neurons, plot them, and plot their remapped version. The neuron from the MLP model comes from an MLP with frequency 14. The MLP is one of the MLPs trained on mod 59 for Figure 4. The pizza transformer is model A (`model_p99zdpze5l.pt` checkpoint) and the clock transformer is model B (`model_l8k1hzciux.pt` checkpoint) from Zhong et al. [5]'s Github repository.

## E.2 Figure 2

This plot shows the simple neuron model, approximate cosets, and Cayley graphs that neurons understand distances on. The point of this figure is to familiarize the reader with our definitions as they are essential to understanding how we derive the abstract aCRT. The code to generate this plot is included in the supplementary materials "make_figure_1_toy_approx_cosets.py".

## E.3 Scaling experiment in Figure 3

We report 1 std dev error bars here, as we do on all our plots, though since our arguments are probabilistic in nature, our growth rate is supposed to be in expectation. We just added the 1 std dev to show the std dev if a normal distribution was fit with the same average. Indeed, the standard deviations are low. For the transformer models (clock and pizza), we take the exact model classes from [5]'s Github repository., and translate them into Jax. For clocks, the attention coefficient is set to 1.0 and for pizzas, the attention coefficient is set to 0.0. The d_model is always taken to be the smallest power of 2 that is larger than $n$, because the architecture requires it. The number of heads is always 4, and the d_head is such that 4 times that number is equal to the d_model. For the hyperparameters used when training the transformers, see Tables 1, 2 3.

| Number | Learning Rate | Batch Size | Weight Decay | Training Set Size |
|---|---|---|---|---|
| 7 | 0.001 | 7 | 0.0001 | 49 |
| 17 | 0.001 | 34 | 0.0001 | 289 |
| 27 | 0.001 | 100 | 0.0001 | 729 |
| 59 | 0.001 | 200 | 0.0001 | 1770 |
| 97 | 0.001 | 200 | 0.0001 | 4850 |
| 113 | 0.001 | 500 | 0.0001 | 6780 |
| 303 | 0.0002 | 909 | 0.00002 | 45450 |
| 499 | 0.0002 | 1497 | 0.00002 | 124750 |
| 977 | 0.0001 | 4885 | 0.00001 | 488500 |
| 1977 | 0.000035 | 39540 | 0.0000075 | 2965500 |
| 4013 | 0.00004 | 16052 | 0.000006 | 3691960 |

Table 1: Experimental results with Adam optimizer across varying parameters for both pizza and clock.

| Number | Learning Rate | Batch Size | Weight Decay | Training Set Size |
|---|---|---|---|---|
| 64 | 0.001 | 64 | 0.0001 | 49 |
| 128 | 0.001 | 128 | 0.0001 | 289 |
| 256 | 0.001 | 256 | 0.0001 | 729 |
| 310 | 0.001 | 310 | 0.0001 | 1770 |
| 720 | 0.0001 | 720 | 0.00001 | 309600 |

Table 2: Experimental results with Adam optimizer across varying parameters in pizzas.

| Number | Learning Rate | Batch Size | Weight Decay | Training Set Size |
|---|---|---|---|---|
| 64 | 0.001 | 64 | 0.0001 | 49 |
| 128 | 0.001 | 128 | 0.0001 | 289 |
| 256 | 0.001 | 256 | 0.0001 | 729 |
| 310 | 0.001 | 310 | 0.0001 | 1770 |
| 720 | 0.0001 | 720 | 0.00001 | 309600 |

Table 3: Experimental results with Adam optimizer across varying parameters in clocks.

For the hyperparameters used when training the MLP, see Table 4. The number of neurons is 8 times the moduli $n$.

| Number | Learning Rate | Batch Size | Weight Decay | Training Set Size |
|---|---|---|---|---|
| 3 | 0.01 | 3 | 0.005 | 9 |
| 5 | 0.01 | 5 | 0.005 | 25 |
| 7 | 0.01 | 7 | 0.005 | 49 |
| 13 | 0.009 | 13 | 0.004 | 169 |
| 17 | 0.009 | 1 | 0.004 | 169 |
| 59 | 0.008 | 59 | 0.001 | 1770 |
| 64 | 0.005 | 64 | 0.0005 | 2048 |
| 113 | 0.004 | 113 | 0.0003 | 6780 |
| 128 | 0.002 | 128 | 0.0002 | 13568 |
| 193 | 0.003 | 193 | 0.0001 | 18914 |
| 256 | 0.001 | 256 | 0.0001 | 34560 |
| 310 | 0.0009 | 310 | 0.00007 | 51150 |
| 433 | 0.0006 | 433 | 0.00005 | 86600 |
| 499 | 0.0005 | 499 | 0.00003 | 124750 |
| 720 | 0.0004 | 720 | 0.000015 | 259200 |
| 757 | 0.0003 | 757 | 0.0000085 | 280090 |
| 997 | 0.0003 | 997 | 0.0000015 | 498500 |
| 1409 | 0.00028 | 1409 | 0.0000009 | 986300 |
| 1999 | 0.00024 | 1999 | 0.0000008 | 2398800 |
| 2999 | 0.00018 | 2999 | 0.0000007 | 4798400 |
| 4999 | 0.0001 | 4999 | 0.0000005 | 14997000 |

Table 4: Experimental results with Adam optimizer across varying parameters for the MLP in Figure 3.

### E.4   Figure 4

We trained one hot encoded 1,2,3,4 hidden layer MLPs and also trained a trainable embedding matrix 1 hidden layer MLP over moduli 59-66. The classes for these models are in the "mlp_models_multilayer.py" file.

Table 5: Hyperparameter configurations for one-hot models with varying hidden layers and embedding.

| architecture | hidden layers | # neurons | L2 regularization | learning rate | train size |
|---|---|---|---|---|---|
| One-hot | 1 | 1024 | $1 \times 10^{-5}$ | 0.00075 | 90% |
| One-hot | 2 | 1024 | $1 \times 10^{-5}$ | 0.00075 | 90% |
| One-hot | 3 | 1024 | $1 \times 10^{-5}$ | 0.00075 | 90% |
| One-hot | 4 | 1024 | $1 \times 10^{-5}$ | 0.00075 | 90% |
| Embedding | 1 | 1024 | $1 \times 10^{-5}$ | 0.00075 | 90% |

## E.5   Figure 5

We trained 500 models of each architecture with 1-hidden layer, and each combination of number of neurons in [512, 2048, 8196, 16392] on mod 59. The models are only saved if final accuracy is $\geq$ 99.9999. The neurons with a maximum preactivation below 0.01 across all of the data were deleted and considered "dead" neurons. We find that as the model width is increased, a single sine wave better and better approximates the preactivations of the neurons.

The classes for the architectures are in the "mlp_models_multilayer.py", "transformer_train_get_data_r2_heatmap_attn=0_top-k_layer_all.py" and "transformer_train_get_data_r2_heatmap_attn=1_top-k_layer_all.py". files. See Table E.5 for precise experimental details.

Table 6: Hyperparameter configurations for Figure 5 1-embed, 1-hidden-layer models with varying hidden unit sizes and architectures.

| architecture | hidden layers | # neurons | L2 regularization | learning rate | train size |
|---|---|---|---|---|---|
| **MLP (baseline)** | | | | | |
| Embed=128, MLP | 1 | 512 | $1 \times 10^{-5}$ | 0.00075 | 90% |
| Embed=128, MLP | 1 | 2048 | $1 \times 10^{-5}$ | 0.00075 | 90% |
| Embed=128, MLP | 1 | 8192 | $1 \times 10^{-5}$ | 0.00075 | 90% |
| Embed=128, MLP | 1 | 16384 | $1 \times 10^{-5}$ | 0.00075 | 90% |
| **"Pizza"** | | | | | |
| Embed=128, Pizza | 1 | 512 | $1 \times 10^{-4}$ | 0.00050 | 90% |
| Embed=128, Pizza | 1 | 2048 | $1 \times 10^{-4}$ | 0.00050 | 90% |
| Embed=128, Pizza | 1 | 8192 | $1 \times 10^{-4}$ | 0.00050 | 90% |
| Embed=128, Pizza | 1 | 16384 | $1 \times 10^{-4}$ | 0.00050 | 90% |
| **"Clock"** | | | | | |
| Embed=128, Clock | 1 | 512 | $1 \times 10^{-4}$ | 0.00050 | 90% |
| Embed=128, Clock | 1 | 2048 | $1 \times 10^{-4}$ | 0.00050 | 90% |
| Embed=128, Clock | 1 | 8192 | $1 \times 10^{-4}$ | 0.00050 | 90% |
| Embed=128, Clock | 1 | 16384 | $1 \times 10^{-4}$ | 0.00050 | 90% |

## E.6   Figure 6

There are 10 models trained with 10 different random seeds (different random init and different random train / test splits) for every (learning rate, weight decay) combination. This is the most pessimistic case for our green checkmark vs purple dot scenario because if a single model doesn't have 100% accuracy after ablating the neurons for our fits, then it would receive a purple dot (assuming accuracy of the trained model with no ablations was 100%). The learning rates and weight decays are:
learning_rates = [0.0025, 0.001, 0.00075, 0.0005, 0.00025, 0.0001, 0.000075, 0.00005, 0.000025, 0.00001]
weight_decays = [ 0.001, 0.00075, 0.0005, 0.00025, 0.0001, 0.000075, 0.00005, 0.000025, 0.00001, 0.0000075, 0.000005, 0.0000025, 0.000001, 0.00000075, 0.0000005, 0.00000025, 0.0000001, 0.000000075, 0.00000005, 0.00000001 ]
There are 1024 neurons in every layer of the models. The embedding matrix has 128 features.

## E.7   Figure 7

There are 10 models trained with 10 different random seeds (different random init and different random train / test splits) for every (learning rate, weight decay) combination. This is the most pessimistic case for our green checkmark vs purple dot scenario because if a single model doesn't have 100% accuracy after ablating the neurons for our fits, then it would receive a purple dot (assuming accuracy of the trained model with no ablations was 100%). The learning rates and weight decays are:
learning_rates = [0.0025, 0.001, 0.00075, 0.0005, 0.00025, 0.0001, 0.000075, 0.00005, 0.000025,

0.00001]
weight_decays = [ 0.001, 0.00075, 0.0005, 0.00025, 0.0001, 0.000075, 0.00005, 0.000025, 0.00001, 0.0000075, 0.000005, 0.0000025, 0.000001, 0.00000075, 0.0000005, 0.00000025, 0.0000001, 0.000000075, 0.00000005, 0.00000001 ].
The embedding matrix has 128 features. There are 1024 neurons in every layer.

### E.8  Figure 8

There are 10 models trained with 10 different random seeds (different random init and different random train / test splits) for every (learning rate, weight decay) combination. This is the most pessimistic case for our green checkmark vs purple dot scenario because if a single model doesn't have 100% accuracy after ablating the neurons for our fits, then it would receive a purple dot (assuming accuracy of the trained model with no ablations was 100%). The learning rates and weight decays are:
learning_rates = [0.0025, 0.001, 0.00075, 0.0005, 0.00025, 0.0001, 0.000075, 0.00005, 0.000025, 0.00001]
weight_decays = [ 0.001, 0.00075, 0.0005, 0.00025, 0.0001, 0.000075, 0.00005, 0.000025, 0.00001, 0.0000075, 0.000005, 0.0000025, 0.000001, 0.00000075, 0.0000005, 0.00000025, 0.0000001, 0.000000075, 0.00000005, 0.00000001 ]
2048 neurons in each layer, 2 layers.

### E.9  Figure 9

5000 models were trained to make these figures. 1 and 4 hidden layers on mod 66.
The learning rate = 0.00075.
The weight decay penalty was L2 regularization = 0.0001.

## F  Further experimental results

### F.1  Goodness of fit of the single sinusoidal approximation

These plots ended up below as Figure 36 and Figure 37. This section will be removed in future versions.

## G  Additional experiments we didn't put in the main text

The point of this section is to hopefully provide interested readers with empirical results that weren't necessary for the main text. These experiments are omitted due to either space constraints or low priority.

Note: we call the neurons learning non-integer frequencies over 2, e.g. $f = \frac{1}{2}$ after remapping "fine-tuning" neurons.

### G.1  Principal component analyses (PCA) of embeddings

Below, we answer all the open problems of Zhong et al. [5] involving non-circular embeddings. The idea of doing the Discrete Fourier Transform (DFT) on the embedding PCA's, gives that they are caused by two different frequency sines: one coming from each principle component (PC).

We replicate the results of [5] and add an additional Fourier transform plot next to their PCA plots, which makes it obvious that the principal components come from clusters with the same frequency. It can be seen that all non-circular embeddings and Lissajous embeddings are caused by the two principal components coming from different cosets. This means that we answer all of the open problems of [5], involving non-circular embeddings.

To make this easy to understand, please see Fig. 14, showing this random seed has four clusters, with key frequencies 35, 25, 8, 42. We now know what frequencies to look for in the DFT plots and thus, can figure out which PCs come from which frequency clusters. Doing so reveals that all PCs where

both PCs come from the same frequency cluster are circular. Conversely, all PCs where the PCs come from different clusters are non-circular Lissajous curves.

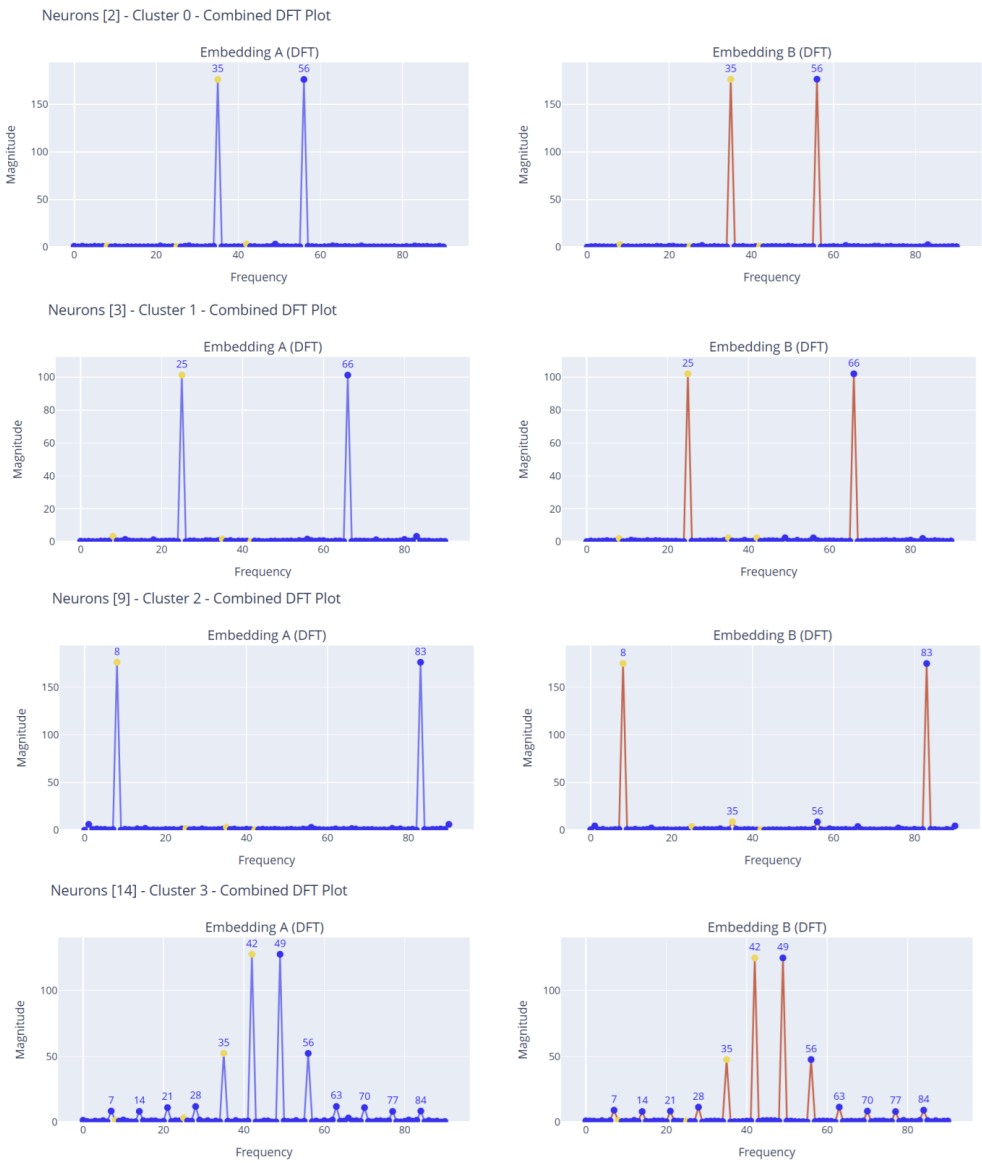

Figure 14: DFT of neurons in each of the four clusters in this random seed. Cluster 0 has frequency 35, cluster 1 has frequency 25, cluster 2 has frequency 8, and cluster 3 is a fine-tuning cluster with frequencies at multiples of 7, 14, 21, 28, 35, and 42.

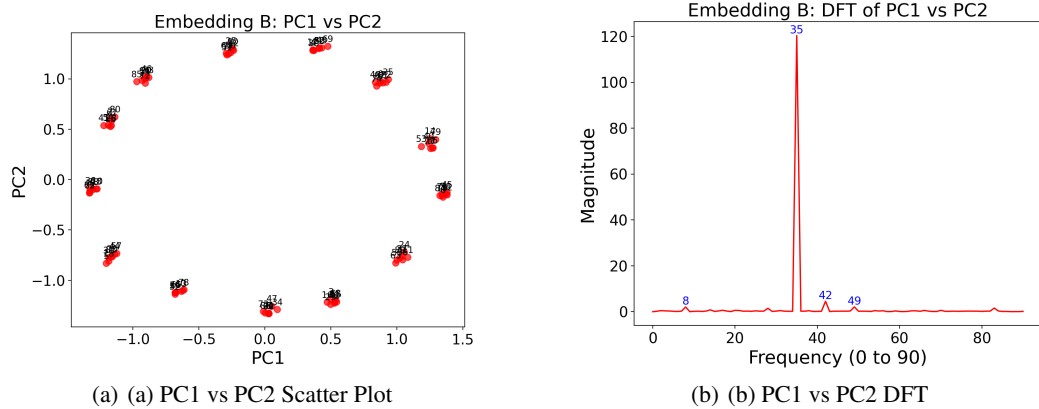

(a) (a) PC1 vs PC2 Scatter Plot

(b) (b) PC1 vs PC2 DFT

Figure 15: PCA and DFT for PC1 vs PC2 showing a circular embedding clustered into cosets. The x and y axis of the left plot are the PC1 and PC2 values for the concatenated embedding matrix for each point $(a, b) \mod 91 \in (0, 0), (1, 1), ..., (90, 90)$. Note that this covers all output classes of the neural network exactly once. Also note that the embedding here is showing 13 cosets with 7 points in them each, *i.e.* all 13 cosets $(a + b) \mod 13 = i, i \in \{0, \ldots, 12\}$ are in the plot. Both PC1 and PC2 have $f = 35$ and since $gcd(35, 91) = 7$, a prime factor, it's possible to learn the exact cosets.

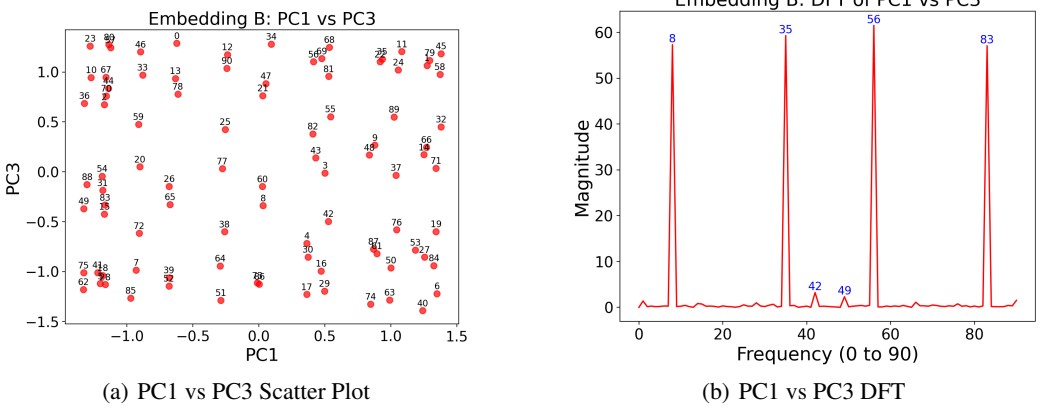

(a) PC1 vs PC3 Scatter Plot

(b) PC1 vs PC3 DFT

Figure 16: PCA and DFT for PC1 ($f = 35$) vs PC3 ($f = 8$), a non-circular embedding.

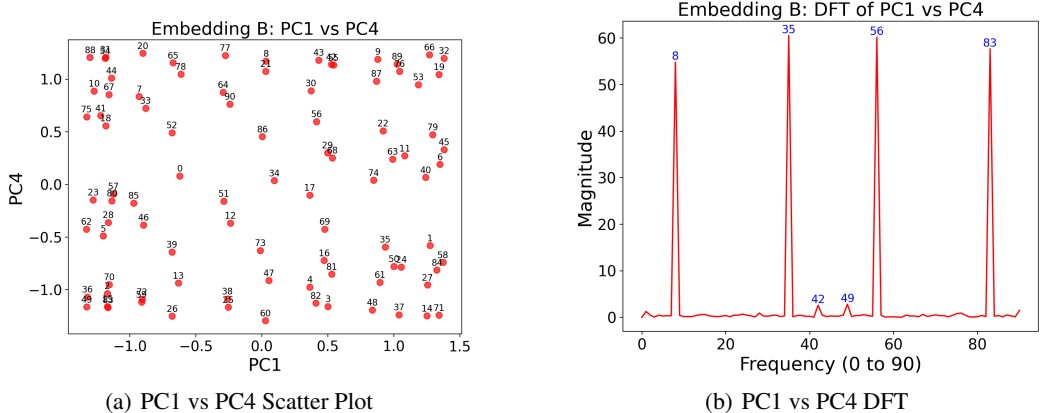

(a) PC1 vs PC4 Scatter Plot

(b) PC1 vs PC4 DFT

Figure 17: PCA and DFT for PC1 ($f = 35$) vs PC4 ($f = 8$), a non-circular embedding.

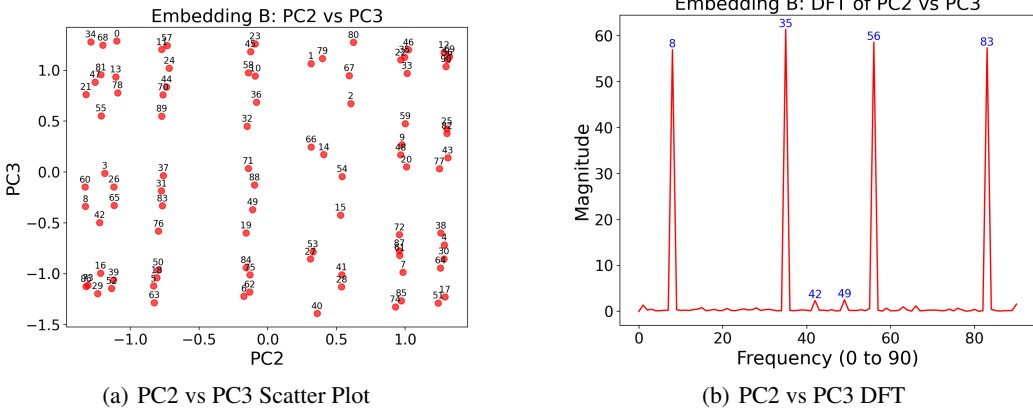

(a) PC2 vs PC3 Scatter Plot

(b) PC2 vs PC3 DFT

Figure 18: PCA and DFT for PC2 ($f = 35$) vs PC3 ($f = 8$), a non-circular embedding.

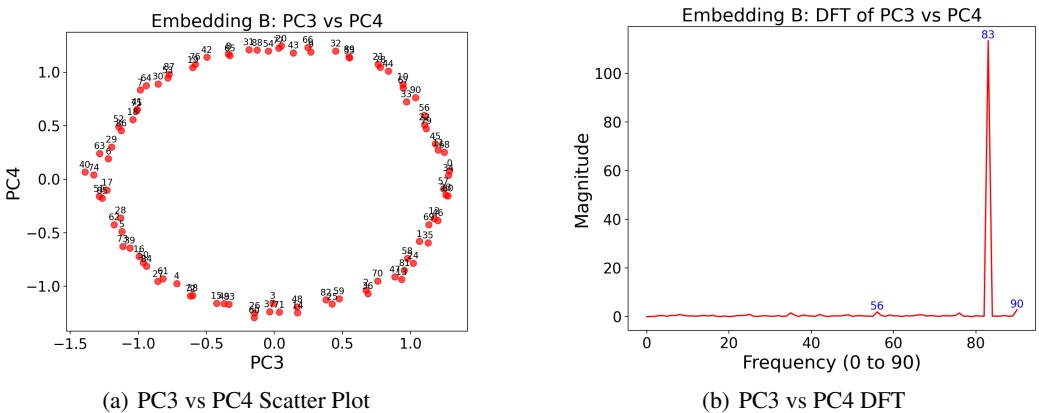

(a) PC3 vs PC4 Scatter Plot

(b) PC3 vs PC4 DFT

Figure 19: PCA and DFT for PC3 ($f = 8$) vs PC4 ($f = 8$), which is a circular embedding because both PC's come from the same frequency cluster.

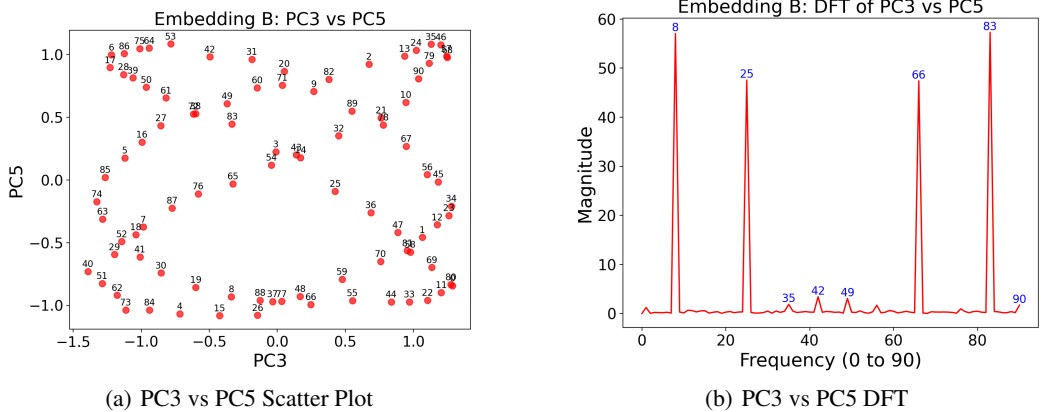

(a) PC3 vs PC5 Scatter Plot

(b) PC3 vs PC5 DFT

Figure 20: PCA and DFT for PC3 ($f = 8$) vs PC5 ($f = 25$), a non-circular embedding.

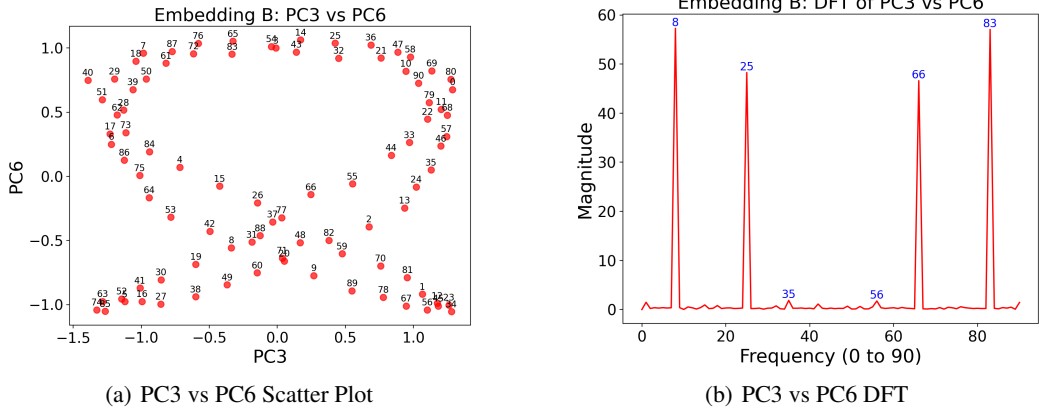

(a) PC3 vs PC6 Scatter Plot

(b) PC3 vs PC6 DFT

Figure 21: PCA and DFT for PC3 ($f = 8$) vs PC6 ($f = 25$), a non-circular embedding.

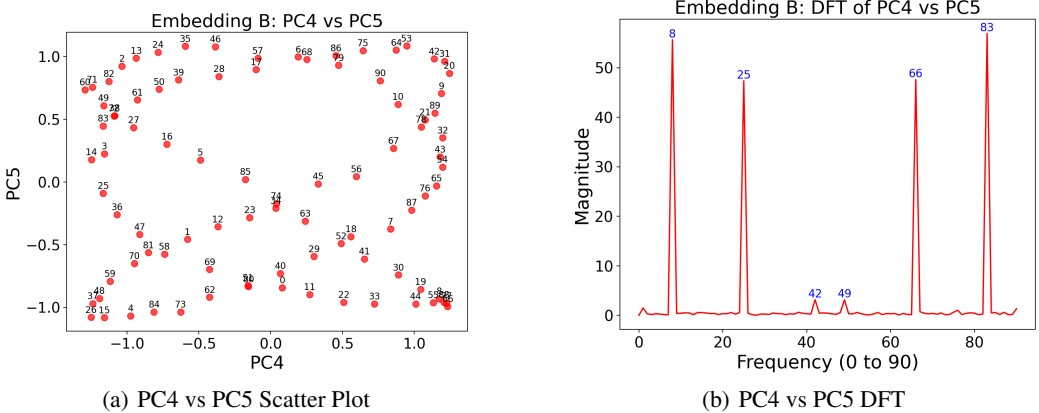

(a) PC4 vs PC5 Scatter Plot

(b) PC4 vs PC5 DFT

Figure 22: PCA and DFT for PC4 ($f = 8$) vs PC5 ($f = 25$), a non-circular embedding.

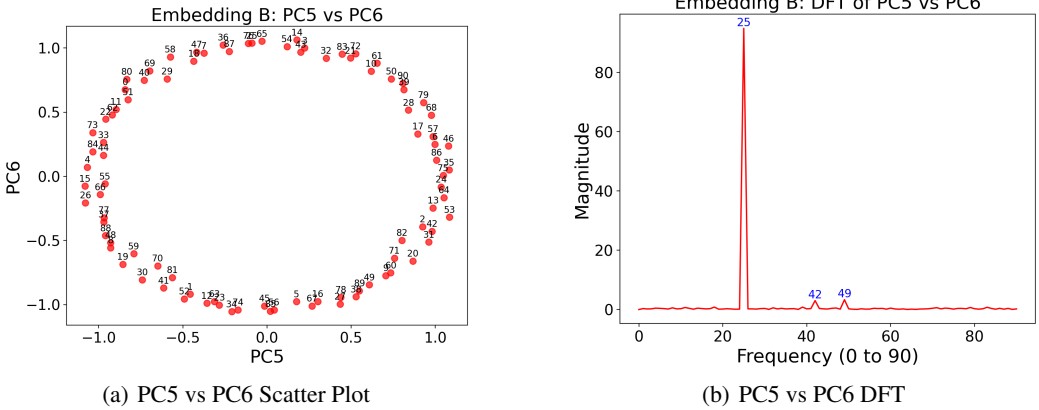

(a) PC5 vs PC6 Scatter Plot

(b) PC5 vs PC6 DFT

Figure 23: PCA and DFT for PC4 ($f = 25$) vs PC5 ($f = 25$), a circular embedding as both PCs come from the same frequency cluster.

## G.2 More examples of simple neurons

See Fig. 24 for a cluster of simple neurons. The neuron remapping (via group isomorphism) can be seen in Fig. 25.

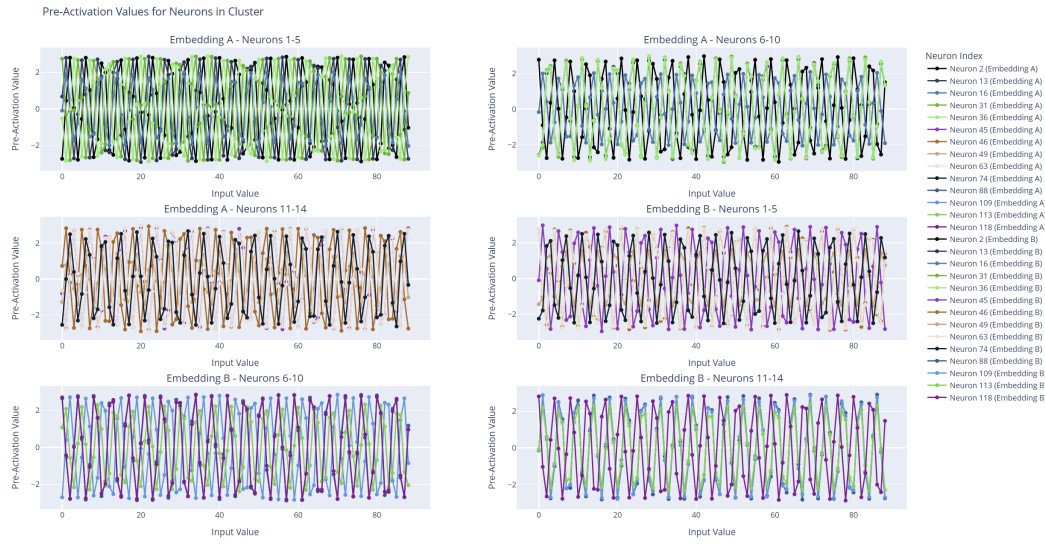

Figure 24: an example cluster of 14 simple neurons of frequency 21.

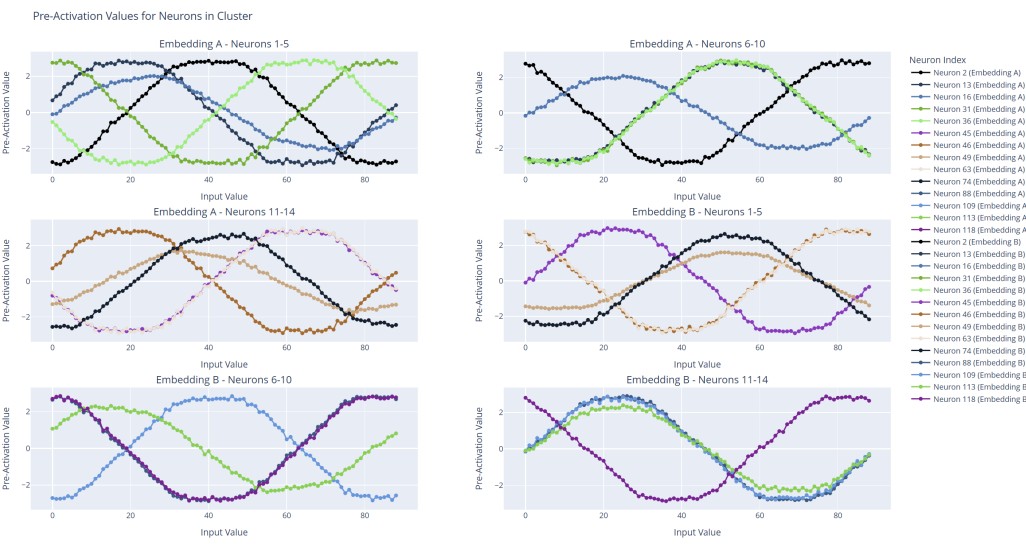

Figure 25: A cluster of simple neurons (from Fig. 24) transformed by group isomorphism so that all neurons have period 1.

## G.3 Studying phase shifts in simple neurons

Here we show how the phases of different neurons in a cluster overlap to give some more information about how clusters of neurons function. See Fig 26(a) for the histograms of the phases of the preactivations of the neurons in a cluster.

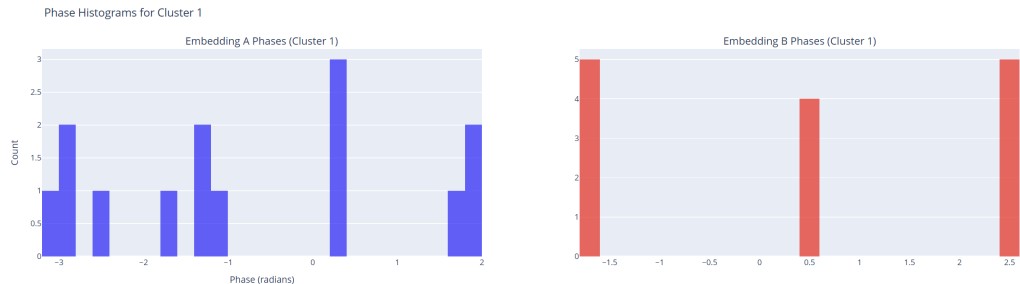

(a) Histogram of preactivations for two neurons in a fine-tuning cluster. The x-axis is the input value into the network for $a$ (left) and $b$ (right).

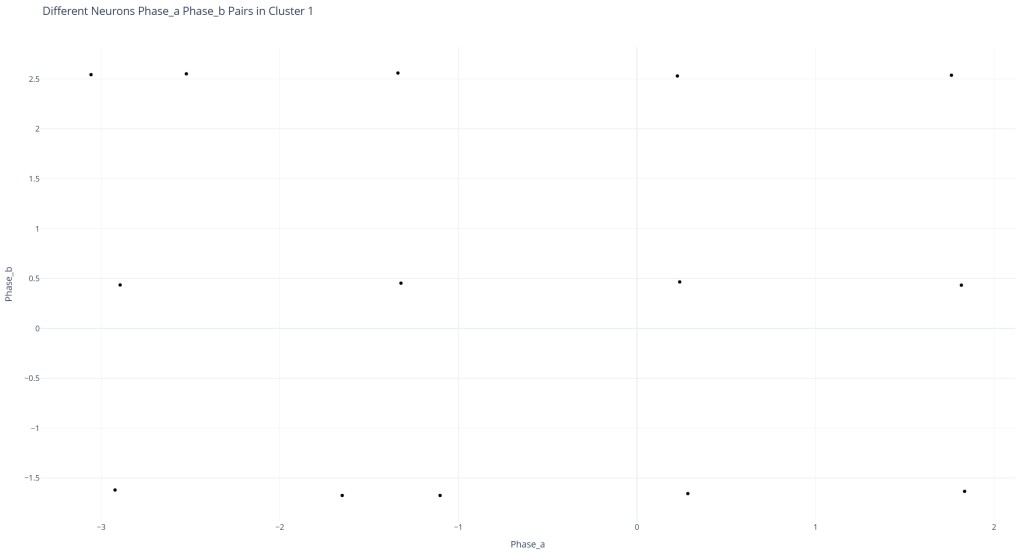

(b) 2D scatter plot created by grouping the phases for each neurons $a$ and $b$ preactivations into a pair (phase-a, phase-b) and plotting the points for all neurons in the cluster in the 2D plane as a black point. In this case, the cluster has 14 neurons of frequency 21.

Figure 26: Histogram of phases (top) and 2D scatter plot of phases (bottom) for a simple neuron cluster with frequency 21.

For a higher resolution view of what's going on, see a 2d scatter plot created by grouping the phases for each neuron's $a$ and $b$ preactivations into a pair (phase-a, phase-b) and plotting the points for all neurons in the cluster in the 2d plane as a black point, see Fig 26(b). It's worth noting that the phases are nice and spread out uniformly like in Fig 26(b) only about half the time. In the other cases

We aren't sure what causes the phases to sometimes align in a nice grid, vs a much more "random" looking configuration as seen in Figure 27. Understanding this is likely essential for proving a realistic bound for the number of neurons needed with the same frequency. We leave this for future work.

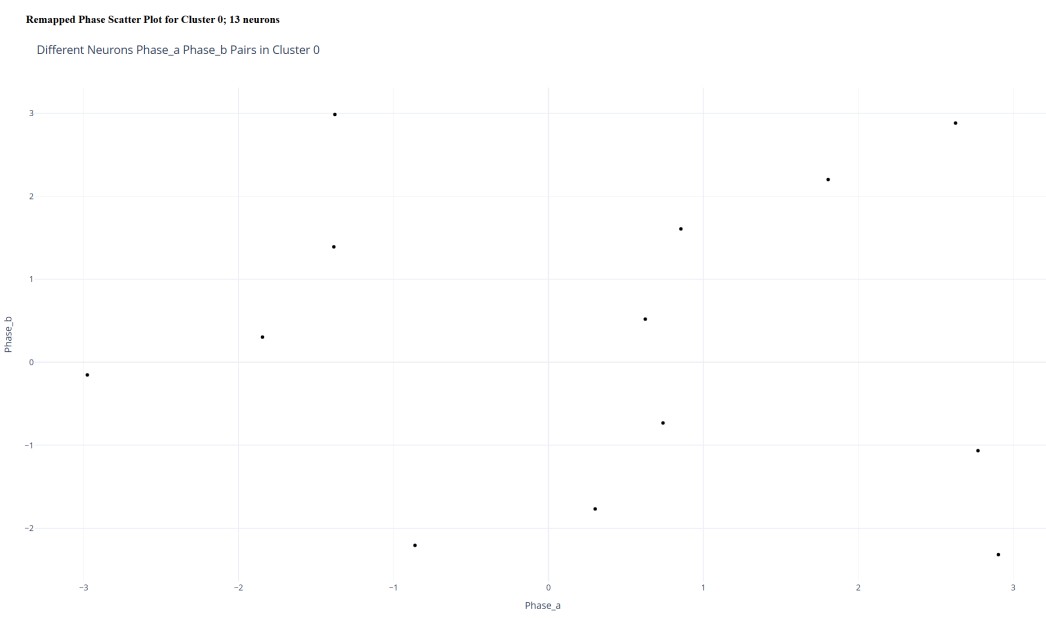

Figure 27: Here's an example where the phases aren't in a nice grid like they were in Figure 26.

## G.4 Studying fine-tuning neurons

Fine-tuning neurons are composed of linear combinations of group representations, in contrast with simple neurons which are composed by one group representation. Additionally, fine-tuning neurons are composed of group representations for what are called harmonic frequencies of the main frequency, for $f = 7$ these would be the multiples of 7, *e.g.* $\{14, 21, 28, 35, 42, ...\}$.

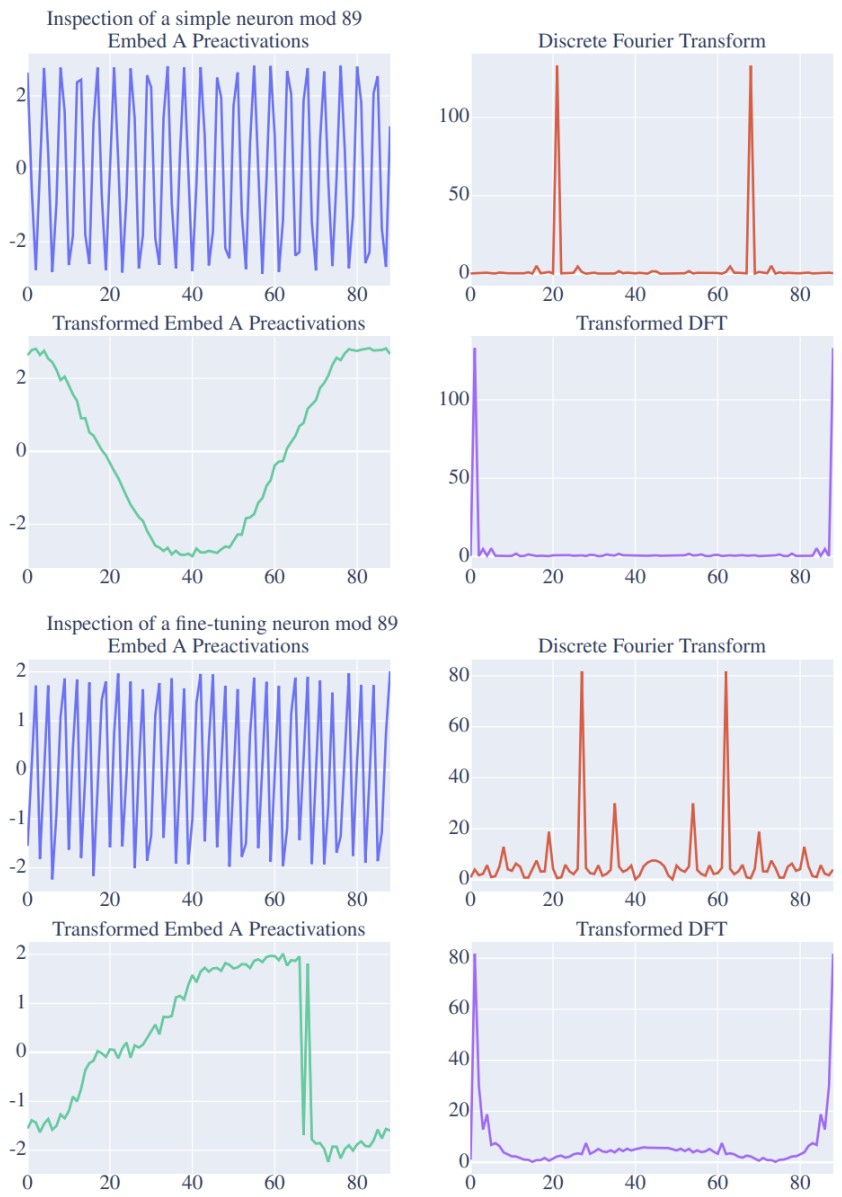

Figure 28: Comparing a simple neuron and fine-tuning neuron before and after transformation by a group isomorphism. The fine-tuning neuron has its DFT concentrating strongest on (27, 35, 19).

We train a neural network with random seed 133 and discover a cluster of fine-tuning neurons. The preactivations for two of these neurons are shown in Fig. 29 and the DFT's for these two neurons are shown in Fig. 30.

We show that these neurons can be generated by linear combinations of representations in Fig. 31.

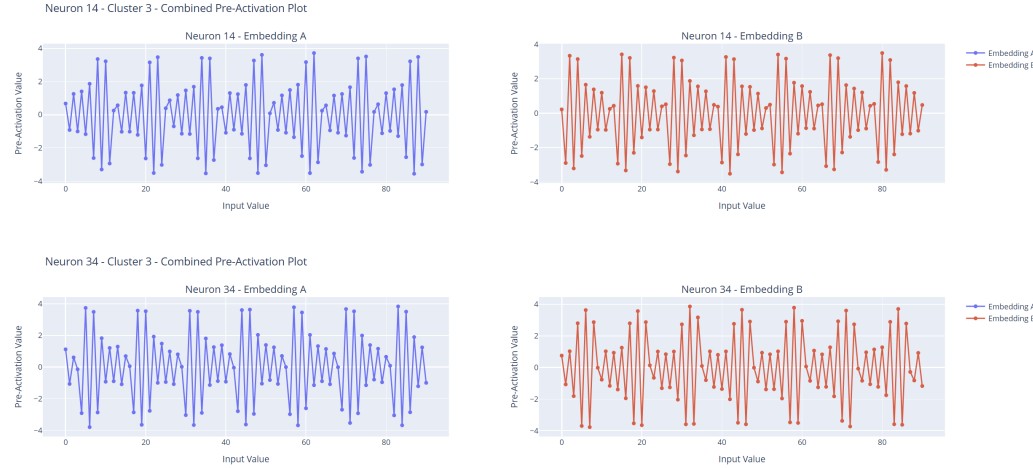

Figure 29: This shows a cluster of fine-tuning neurons and shows the preactivations of the first two neurons in the cluster. The x-axis is the input value into the network for $a$ on the left, and the input value for $b$ on the right.

Cluster 3

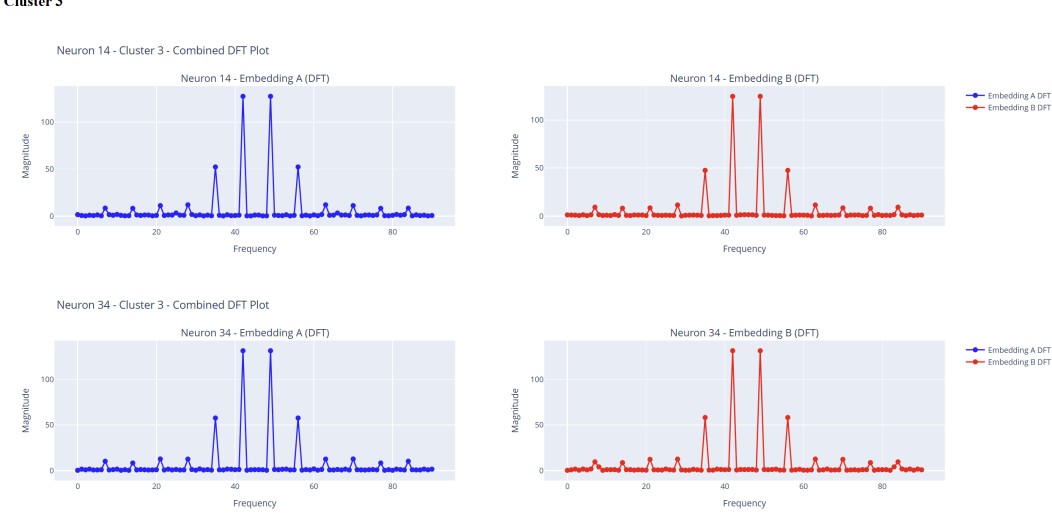

Figure 30: This shows the DFT's of the preactivations of the fine-tuning neurons seen in Fig. 29. The x-axis is the frequency (from 0-90 because this is $(a + b) \mod 91$. The y-axis shows that the representations contributing are $42, 35, 28, 21, 14, 7$ in descending order. Note the DFT is symmetric about its midpoint so values after 45 contain the same information as the values up to 45.

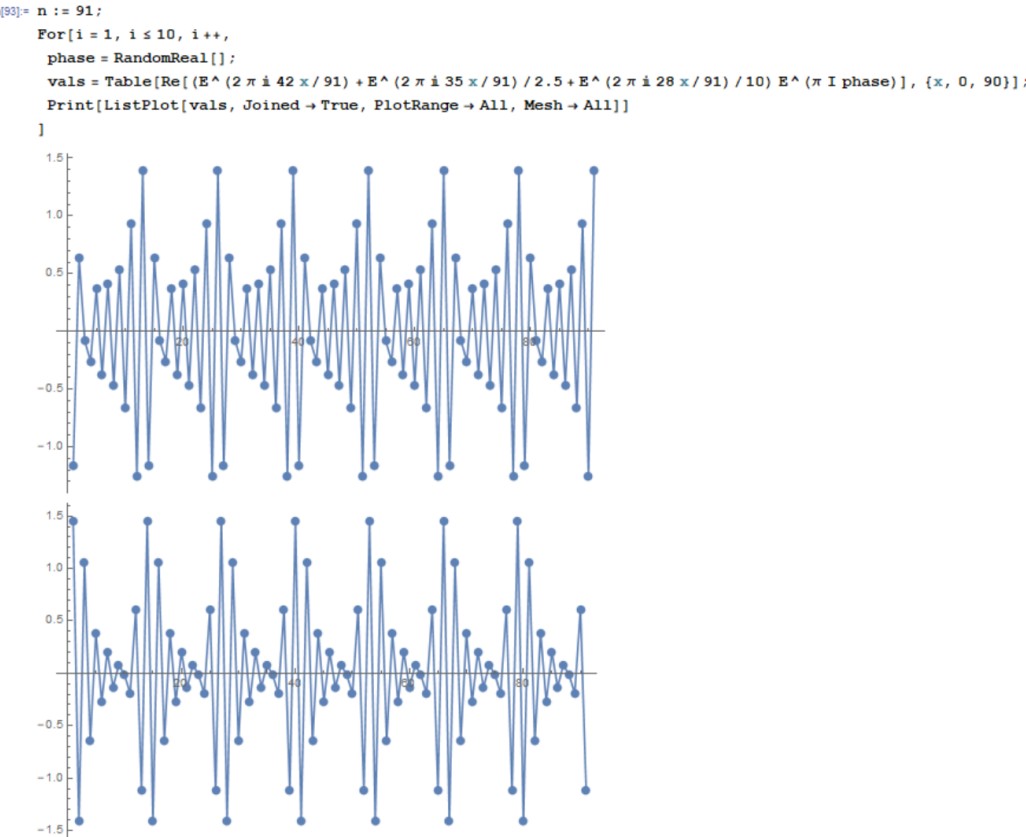

```
In[93]:= n := 91;
For[i = 1, i ≤ 10, i++,
  phase = RandomReal[];
  vals = Table[Re[(E ^ (2 π i 42 x / 91) + E ^ (2 π i 35 x / 91) / 2.5 + E ^ (2 π i 28 x / 91) / 10) E ^ (π I phase)], {x, 0, 90}];
  Print[ListPlot[vals, Joined → True, PlotRange → All, Mesh → All]]
]
```

Figure 31: Constructing a fine-tuning neuron. This diagram illustrates the step-by-step process of constructing a fine-tuning neuron, highlighting that it is a linear combination of representations.

### G.5 Histograms of frequency learned counts for simple and fine-tuning neurons

Note that the next two histograms are created by recording frequencies with weights in the DFT in the range of (7.5, 30). This is not a sufficient way to always detect fine tuning neurons, and sometimes it will include simple neurons in its counts, however this is much more rare. If you consider the ability for neurons with preactivations of specific frequencies to contaminate other neurons frequencies slightly (because they may modify values in the embedding matrix by a small amount), you will see where this counting method can go awry. It is however the case that usually, the contamination coming from a different cluster of simple neurons is below 7.5. Thus, these plots should not be considered "accurate" and just approximations.

These plots are still useful to show the relative frequency of simple neurons vs. fine tuning neurons. The histogram of Frequencies found Fig. 32(a) found a uniform distribution with each frequency showing up about 10k times. Removing the vast majority of contamination by filtering with 7.5 (usually the DFT magnitudes on other frequencies are 0 and if they aren't near 0 then they are less than 4 and there is a simple neuron making use of that frequency in a different cluster (*i.e.* a simple neuron has one big spike with magnitude over 60 on that frequency). This gives us about 2200 fine-tuning neurons found with each frequency, including overcounting because fine-tuning neurons make use of linear combinations of representations and thus their DFT usually has three or more values in the range (7.5, 30). Thus the histograms of frequencies associated with fine-tuning neurons are upper bounds on the number of clusters that are identified across 100k random seeds to be fine-tuning neurons. Around 25 percent of runs have fine-tuning neurons in them, but we aren't sure how hyperparameter settings affect this.

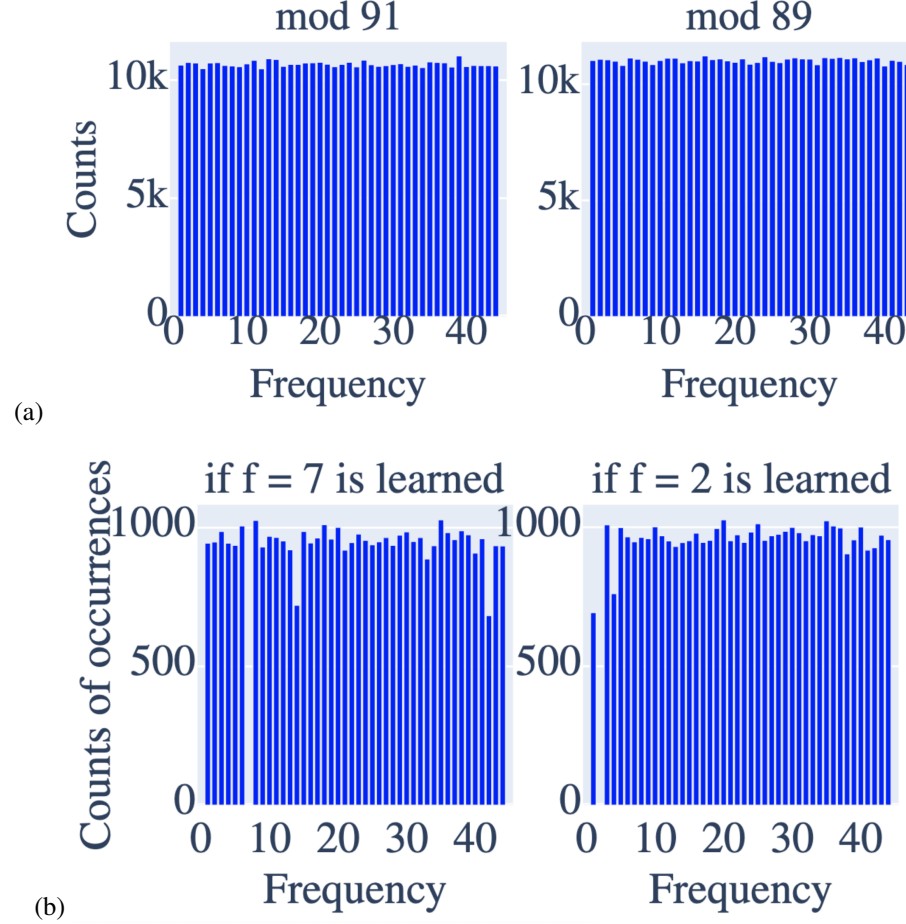

Figure 32: (a) Histograms of frequencies found across 100k random seeds in MLPs mod 91 (factors 7 and 13) and mod 89 (prime) are both uniform. The fact both prime and composite numbers give uniform distributions is strong evidence that networks are learning the same thing in both problem settings. In fact, this observation – that networks do not prefer prime factor frequencies (exact cosets) – is what led us to define approximate cosets and identify the abstract approximate CRT algorithm, which is learned with both prime and composite moduli. (b) Conditional histograms of frequencies over 100k seeds, both mod 91. Left: if frequency 7 is found then neurons with $f = 14$ or $f = 43$ are less likely (note that $2 \cdot 43 \equiv -7 \pmod{91}$). Right: if frequency 2 is present then frequencies 1 and 4 are less likely. The conditional histograms show that networks try to avoid learning frequencies with additive and subtractive relations. This is for a similar reason to why the CRT does not work unless all factors of $n$ are coprime – they would intersect and boost the value of incorrect logits, substantially increasing the loss.

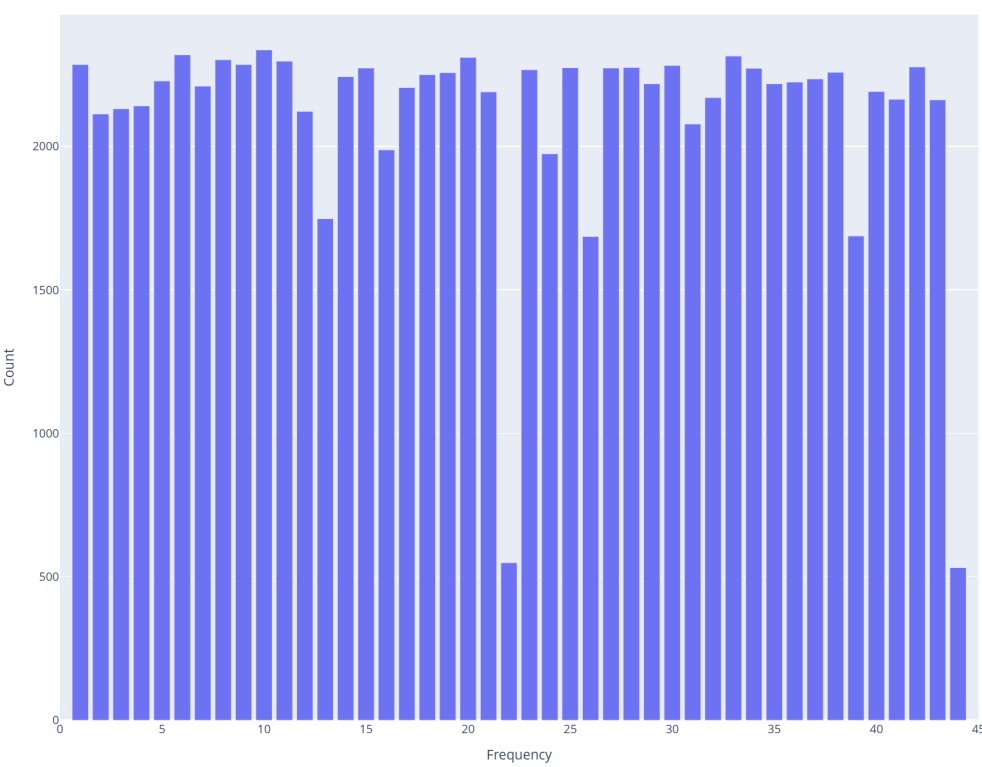

Figure 33: Histogram of frequencies (0–45) associated with fine-tuning neurons over 100k random seeds for modulus91. Note that frequencies 22 and 44 are least common, and 13, 26, 39 also appear less frequently, giving a case where the neural net is less likely to find some prime factors. It makes sense that neural networks won't always discover the prime factors or they'd be solving prime factorization.

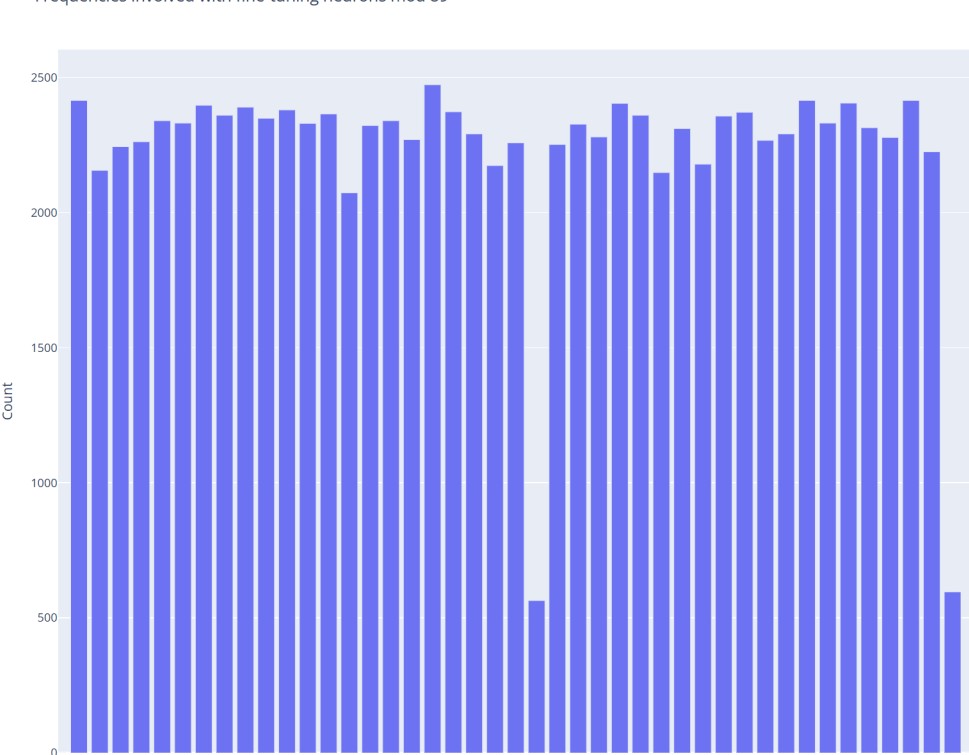

Figure 34: Histogram of frequencies (0–44) associated with fine-tuning neurons over 100k random seeds for modulus 89. Note that frequencies 23 and 43 are the least common.

## G.6 Fine-tuning neurons like additive and subtractive relations

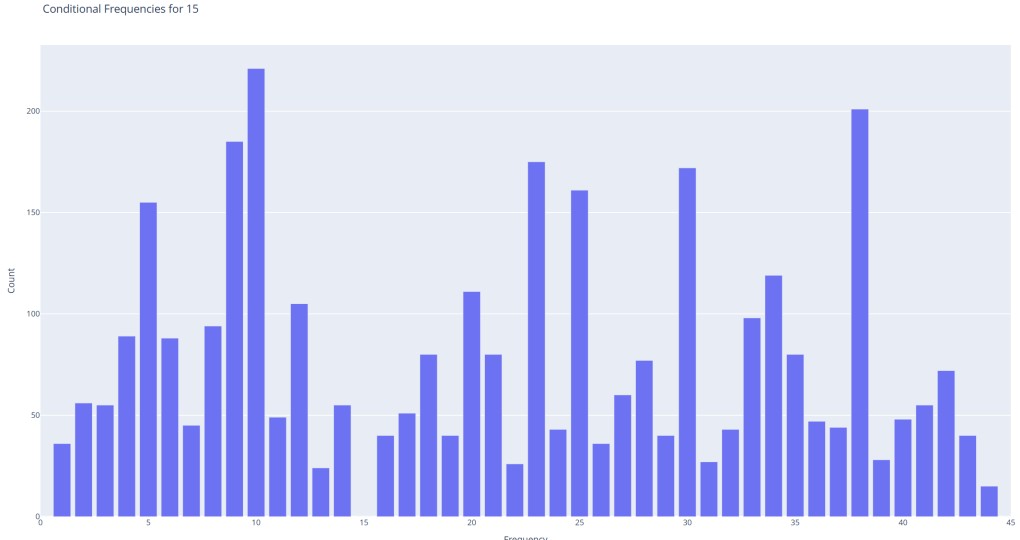

(a) Fine-tuning Neuron Additive Relations given 15 is a frequency; $(a + b) \bmod 91$.

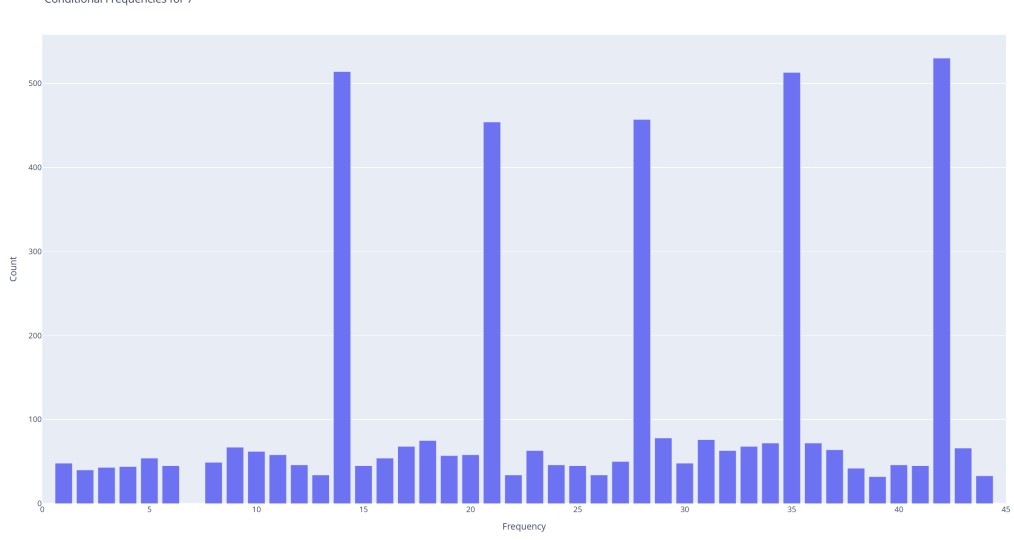

(b) Fine-tuning Neuron Additive Relations given 7 is a frequency; $(a + b) \bmod 91$.

Figure 35: Fine-tuning neuron additive relations for two different cases. If a neuron with frequency 15 is learned, frequencies that are multiples of 5 are more likely to be found. The same applies to 7, which is a prime factor of 91, the moduli.

## G.7 Histograms of counts of frequencies being learned in mod 59 and mod 66 across varying depths in the pizza and clock transformers as well as MLPs (with 1 embedding layer)

Here we show histograms counting the frequencies learned and the lengths of the average number of neurons involved in a frequency cluster given frequency $f$ was learned over 500 seeds for each architecture.

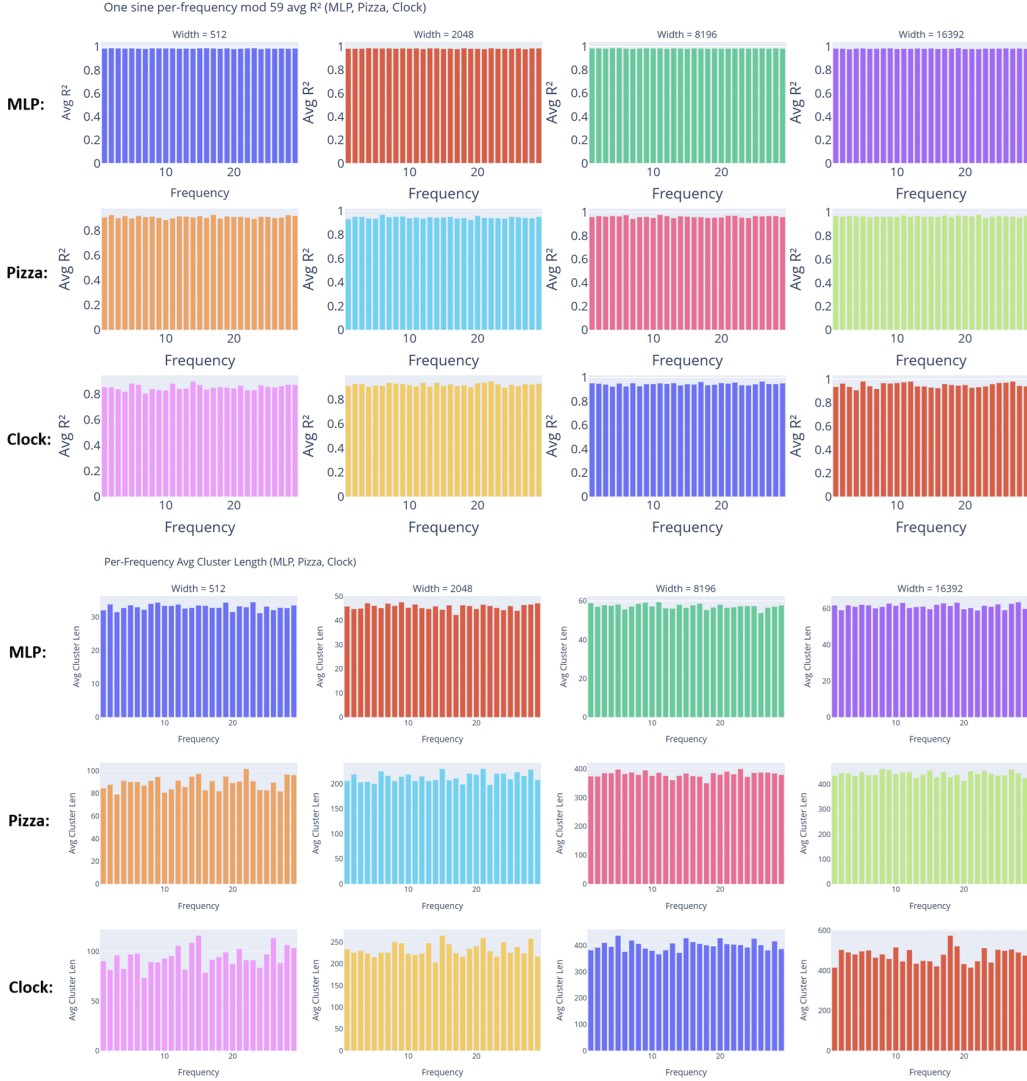

Figure 36: $(a + b \mod 59)$: The first three rows show the histograms of learned frequencies and the bottom three rows show histograms of the average length of a cluster, i.e. number of neurons in the cluster.

We check the goodness of fit as a function of frequency, and find that on mod 66 with prime factors 2, 3 and 11, the $R^2$ value is much closer to 1.0 for MLPs when frequencies 22 or 33 are learned $(2 \times 11)$ and $(3 \times 11)$. Indeed, if the network learns 22 or 33, we see that the length (the number of neurons with that frequency) is substantially lower than if it learns other frequencies. The $R^2$ is also higher for MLPs if it learns $(2 \times 6)$, however it's not as high in the previous cases, and doesn't have the change in the number of neurons that the other two cases have. Overall, this makes sense as in this situation, it can learn the exact cosets instead of needing to learn approximate cosets. Investigation into which frequencies require less neurons is an interesting subject for future study; why is it not all cosets?

Some cosets are more likely to be learned than others.

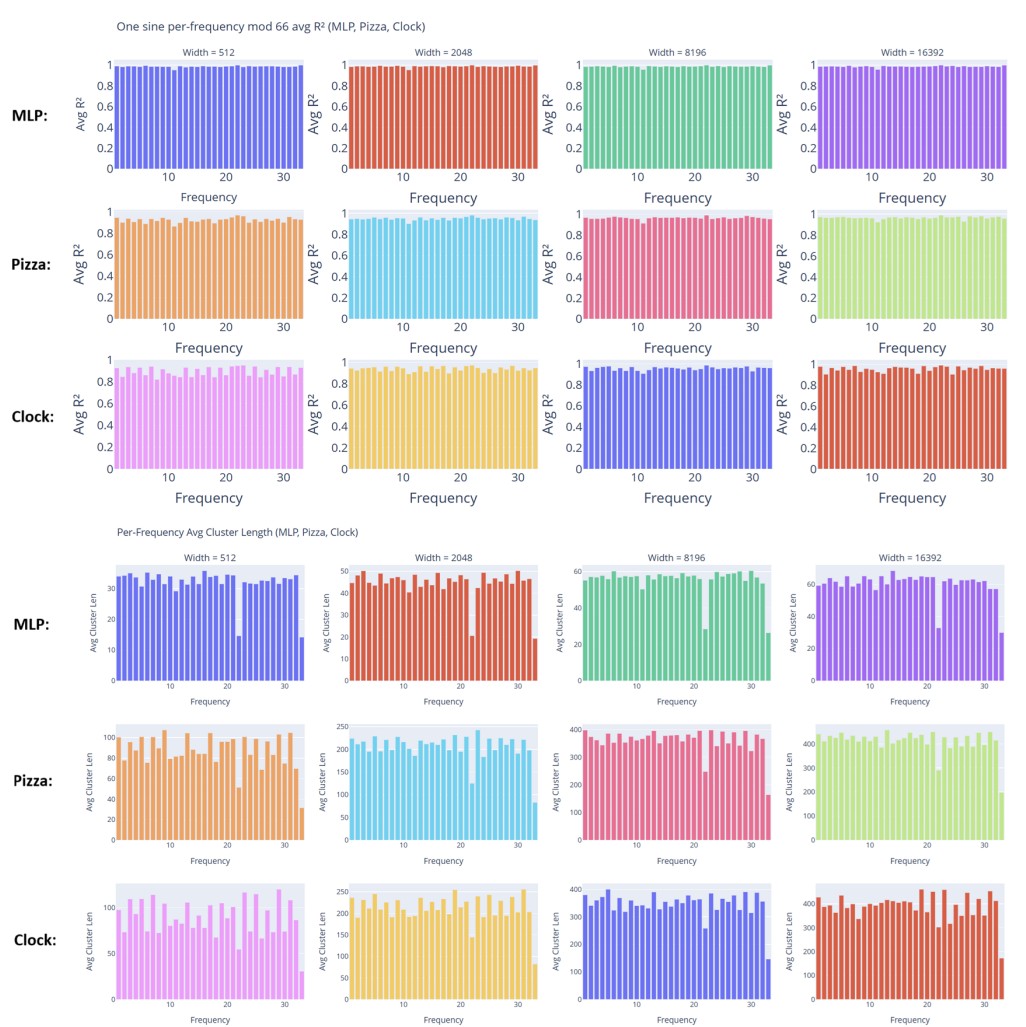

Figure 37: ($a + b \mod 66$): The first three rows show the histograms of learned frequencies and the bottom three rows show histograms of the average length of a cluster, i.e. number of neurons in the cluster; note that learning precise cosets results in less neurons being required.

## G.8 Noise and ablation studies

In this section, we take the clusters from random seed 133 and we randomly inject multiplicative scaling noise into every weight attached to neurons in the cluster. We do this by multiplying the weight by $e^s$, $s \sim \mathcal{N}(0, \sigma)$, for various $\sigma$ on the x-axis in Fig. 38(b)

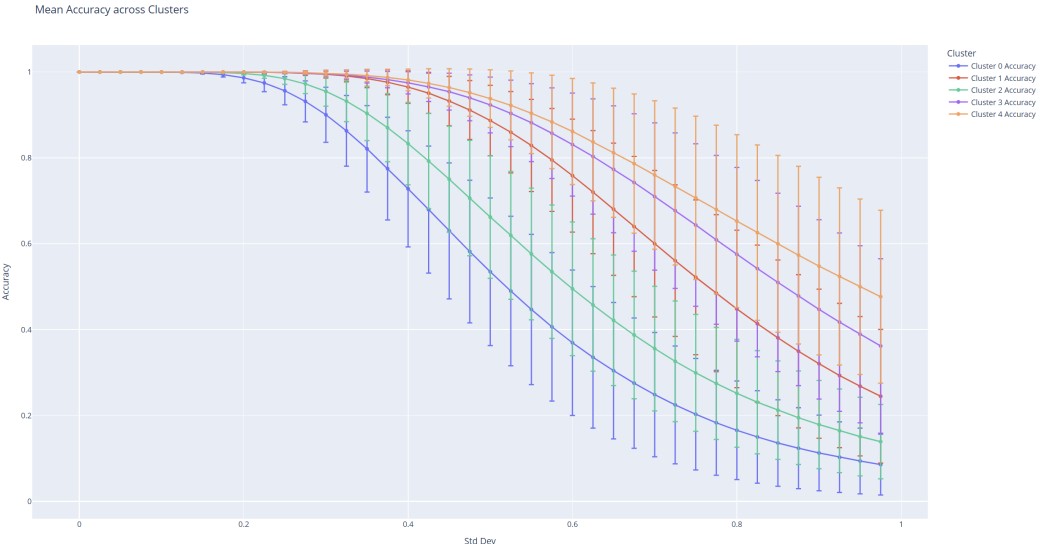

(a) Multiplicative Noise injected into every weight of every neuron in a cluster from a normal distribution with std dev $\sigma$.

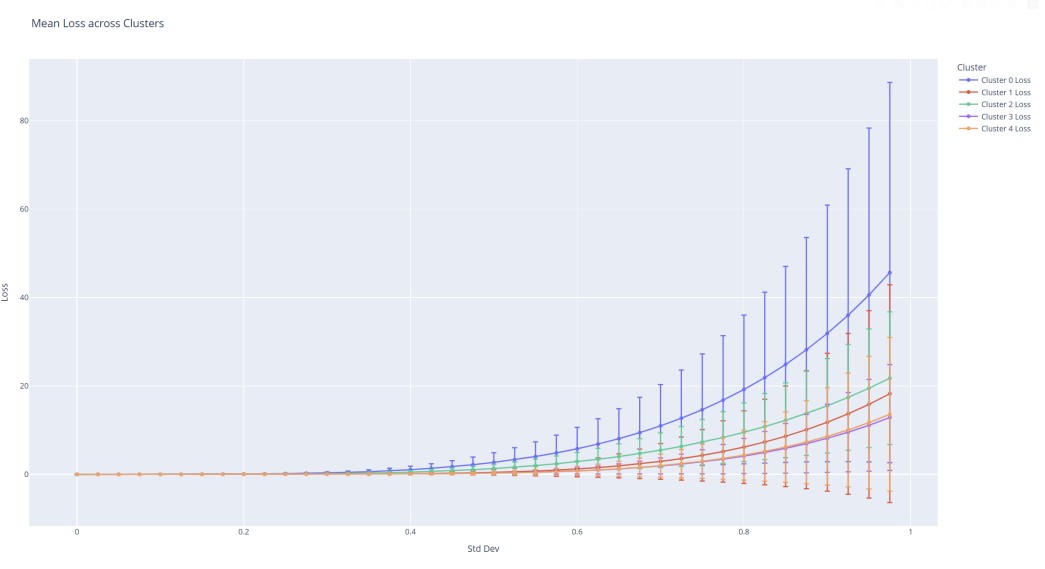

(b) Effect of Multiplicative Noise on the loss function.

Figure 38: Neural network robustness to injected multiplicative noise. The loss remains stable even with a std dev of 0.225. Note cluster 3 and 4 are composed of four fine tuning neurons each. Every other cluster is composed of simple neurons.

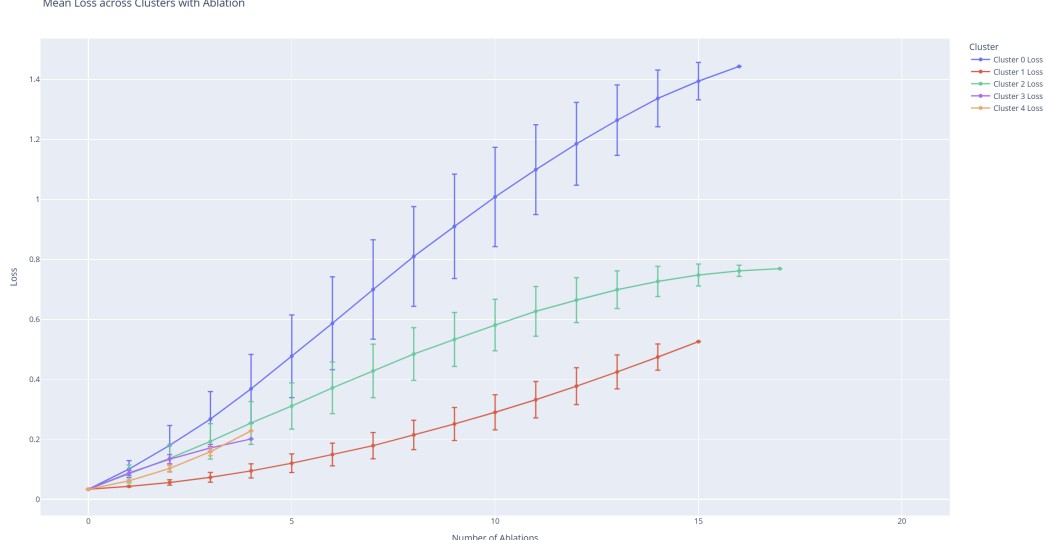

(a) Ablation study showing the impact on the loss function when removing neurons from specific clusters.

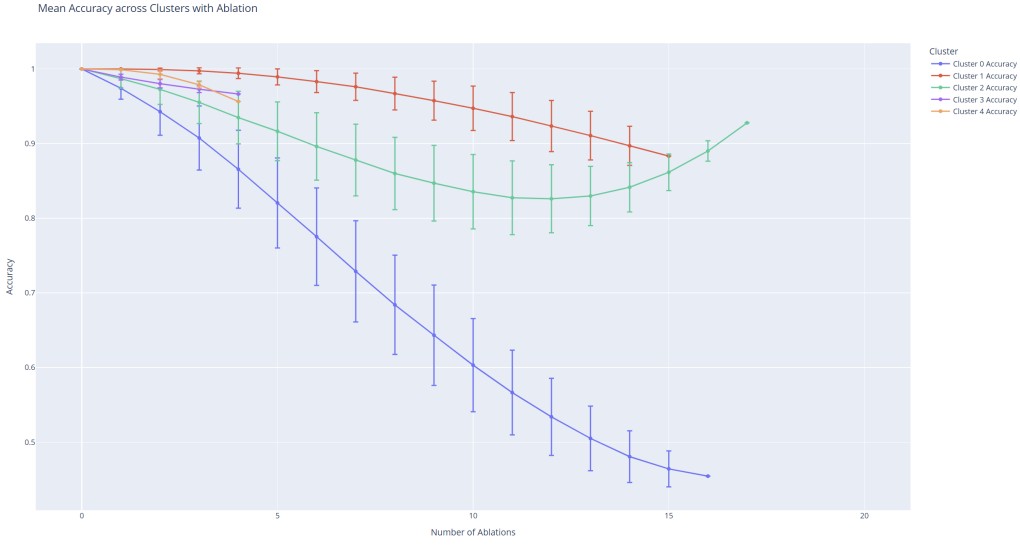

(b) Impact of neuron removal on accuracy.

Figure 39: Ablation study results. Loss and accuracy metrics highlight the impact of randomly removing neurons "number of ablations" number of neurons from a cluster. Note cluster 3 and 4 are composed of four fine tuning neurons each and deletion of every neuron in the cluster doesn't affect the accuracy by much. Every other cluster is composed of simple neurons.

### G.9 Number of frequencies

#### G.9.1 More overparameterized means less frequencies

Scaling the number of neurons in the layer achieves experimental results within $\mathcal{O}(\log(n))$.

Experiments in scaling the number of neurons show that the average number of frequencies found can be shifted based on hyperparameters, but this is something we don't fully understand at this point in time, but as our results in the main body show–it is still logarithmic. See Fig. 40.

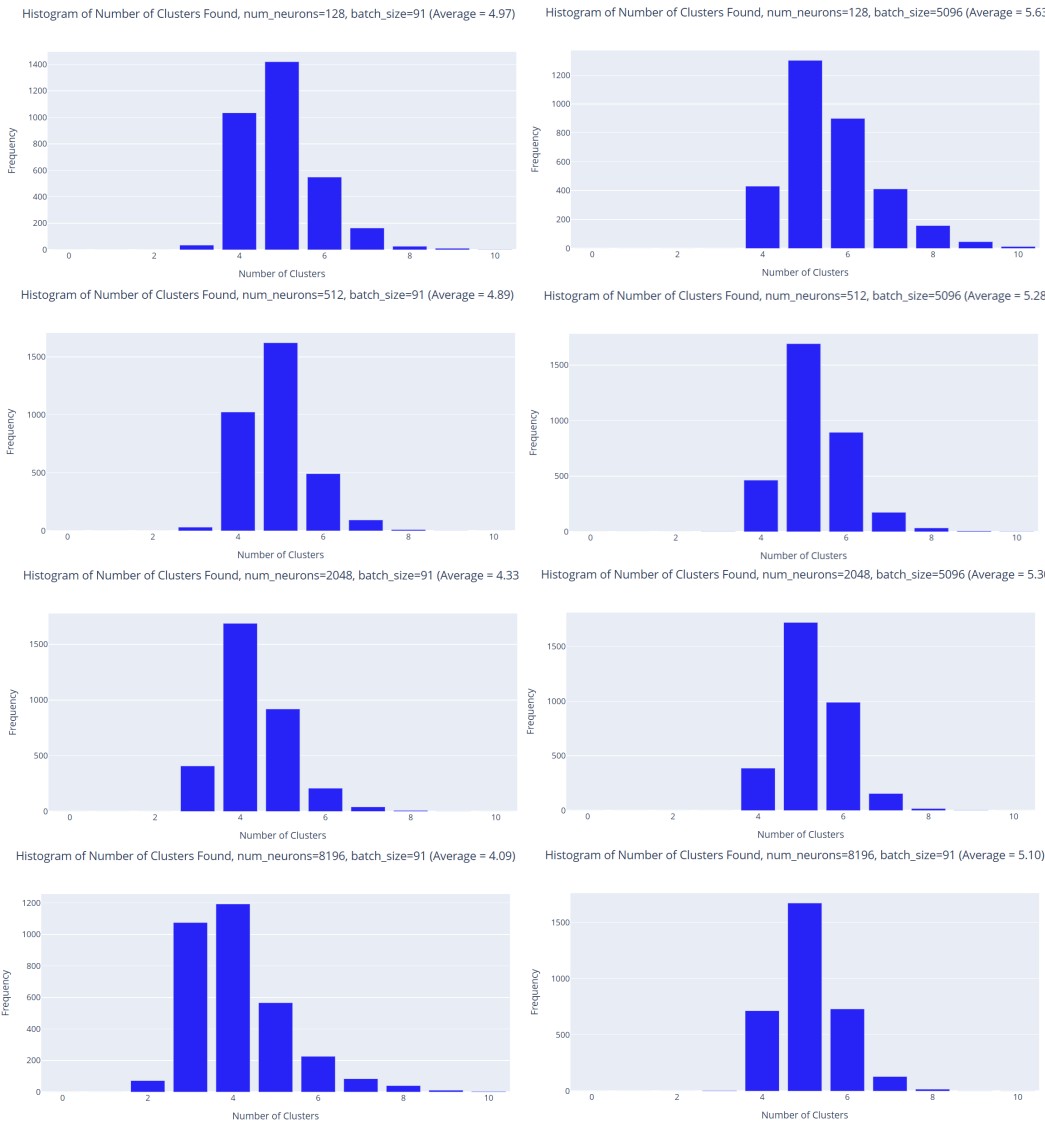

Figure 40: This figure shows that the scaling is always $\mathcal{O}(\log(n))$, even as the number of neurons is increased from 128, to 512, to 2048, to 8196. The first column is batch_size = 91, and the second column is batch_size = 5096, *i.e.*, the entire training set size. All results are upper bounded by $\mathcal{O}(\log(n))$.

#### G.9.2 Adding depth also means less frequencies are learned

In Figure 41 we see that adding layers results in fewer frequencies being learned.

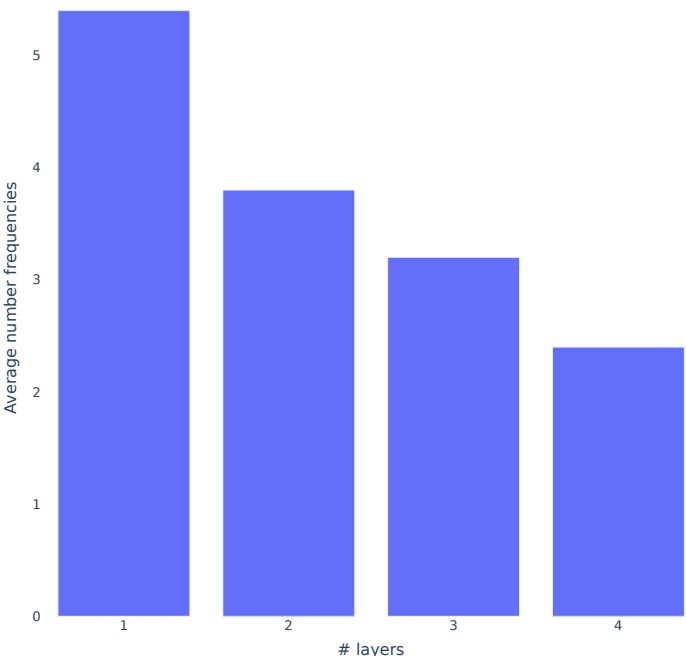

Figure 41: Adding depth causes the network to learn slightly less frequencies.

Indeed, instead of just looking at this for one hyperparameter combination, we can check it out for many like we did in figures in the main paper (see Figure 42).

### G.9.3    Deep networks learn error correcting codes: empirical results

**Deep networks learn error correcting codes.** This discussion was omitted from the main paper's Discussion due to space constraints, but we believe it's interesting.

Our result finding that in deep networks, layers after 1 keep around a % of first-order sinusoids (Figure 8) can be interpreted as the network constructing an error correcting code. We see in Figure 42 that deeper networks learn less frequencies. While this is true, they simultaneously achieve lower cross entropy loss (and better margins) with less frequencies than shallower networks 43. This is because the first order (simple) neurons in layer 2, compute the exact same coset computation as simple neurons performed in layer 1. Resultantly, the second order cosine neurons that store the correct answer as $\cos(\frac{2\pi \cdot f(a+b-c)}{n})$, which is maximized at the correct answer $(a+b) \bmod n = c$, receive an additional linear combination of Theorem 4.7. This boosts the height of the correct answer (linearly in the number of layers) as a function of number of distinct frequencies. Furthermore, it boosts the height of incorrect logits at most $\mathcal{O}(\log(n))$. After softmax, the exponential difference between the correct logit and incorrect logits is thus amplified, and thus the softmax (cross entropy loss) is much lower. With 1 hidden layer it was $\mathcal{O}(n^{-\Omega(1)})$. In general, with $L$ the number of layers, we conjecture that after softmax, using $\mathcal{O}(\log(n))$ distinct frequencies will give closer to $\mathcal{O}(n^{-\Omega(L)})$ as the height of incorrect logits. This results from the network redundantly doing the same computation to "error correct" and thus reduce errors.

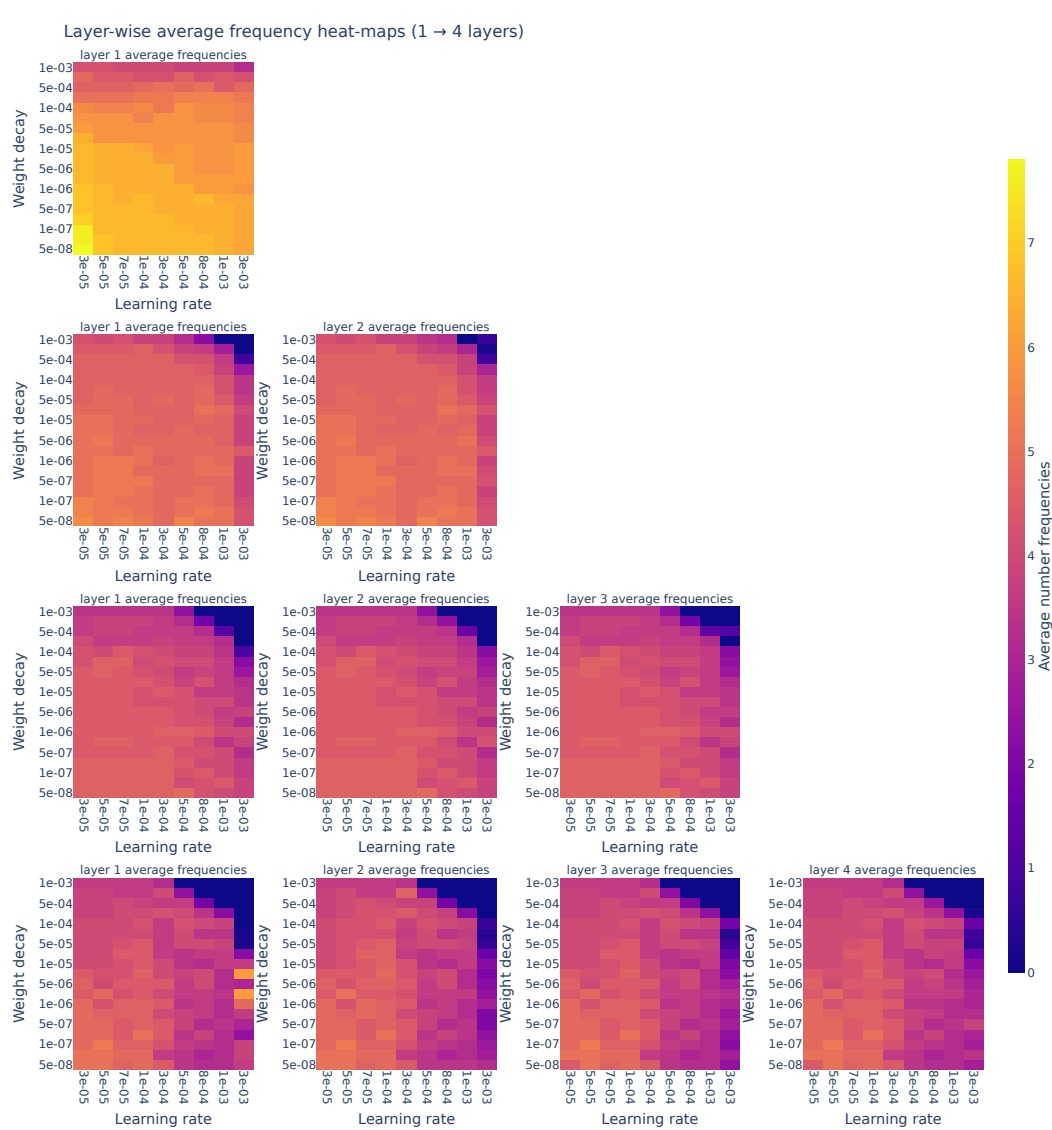

Figure 42: The number of frequencies found in the network decays as we add layers across almost all hyperparameter combinations.

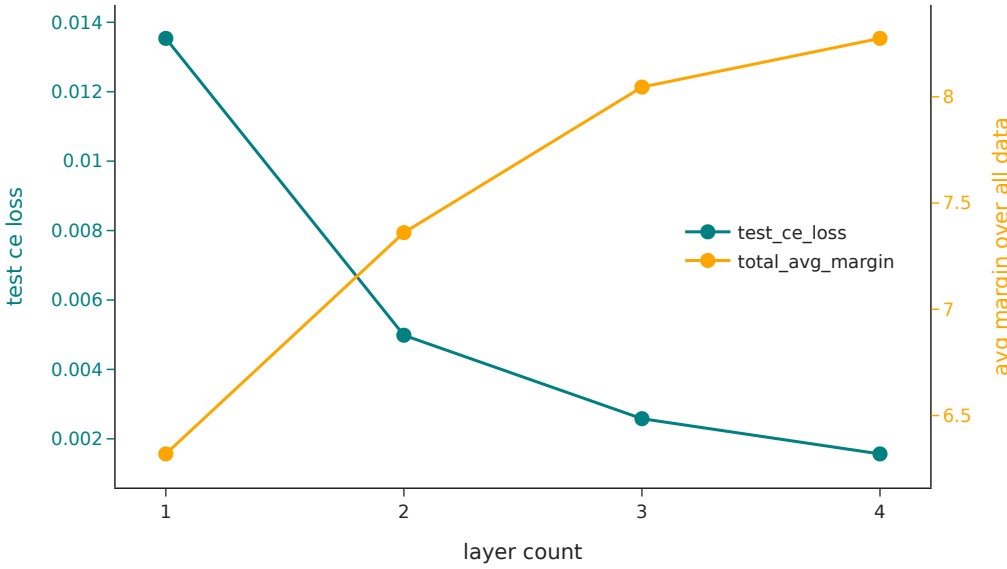

Figure 43: The loss and margins improve as layers are added, yet figure 42 shows that less frequencies are learned. This can be explained by the presence of first order sinusoidal neurons in layers after layer 1.

### G.10   Qualitative: equivariance of the cluster contributions to logits

Here you can see that the clusters of neurons are approximately equivariant to shifts in the inputs, *i.e.* the cosets shift with the inputs. We show that if you shift (a,b) both by 2, the clusters shift by 4.

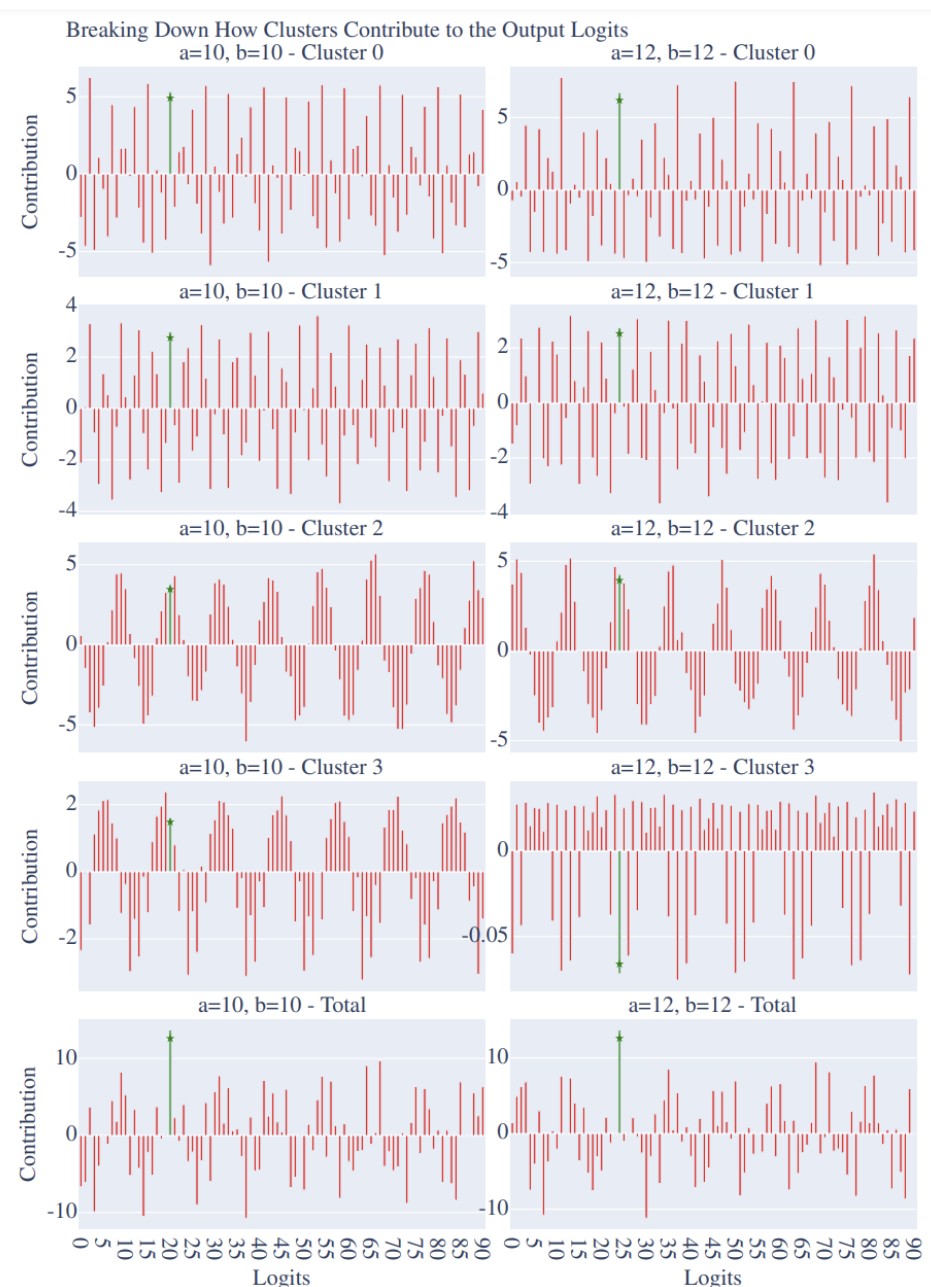

Figure 44: Clusters of neurons are approximately equivariant to shifts in the inputs, meaning coset clusters shift with the inputs. This suggests the network has learned cosets that it uses to intersect, via linear combinations, to perform the approximate CRT. This example demonstrates that the network did not learn a global minimum—e.g., Cluster 3 has only four neurons, limiting its expressivity and equivariance. Cluster frequencies: Cluster 0 (35, coset), Cluster 1 (25, approximate coset), Cluster 2 (8, approximate coset), and Cluster 3 (42, coset). This example uses the same random seed as the ablation study (Fig. 39(a)), where Cluster 0 is the most active. This data is from an MLP.

### G.10.1 Pizza model

We take model A, specifically `model_p99zdpze5l.pt`, from [5] and make Figure 45, which shows that pizzas also output on approximate cosets and perform an approximate CRT. Note for example, that the output logits for the cluster with max freq = 15: has maximum activation along an approximate coset $\frac{59}{15} = 3.93$, and if the neuron activates strongly at $a$ then it also activates strongly at $a \pm 4$.

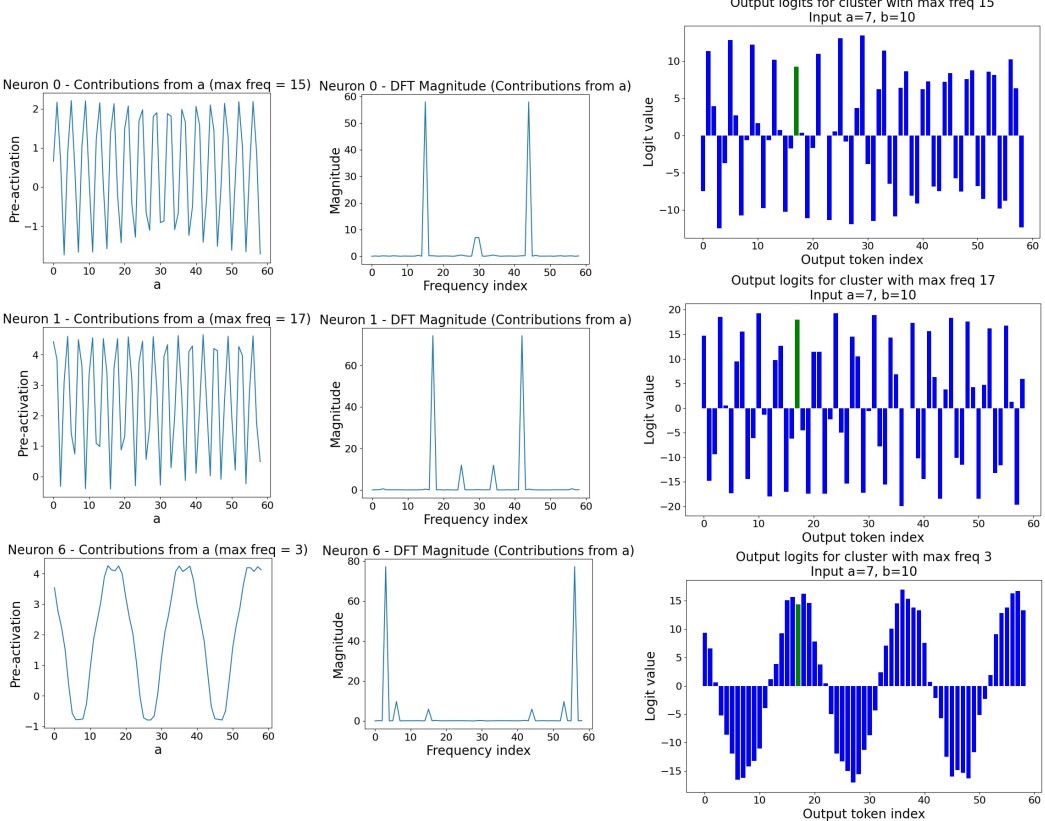

Figure 45: This figure shows three neurons and their DFT's, each from one of three clusters in model A (a pizza-transformer) from [5] for these experiments. Note that the pizza neurons (and clusters) are also implementing the abstract approximate Chinese Remainder Theorem algorithm, despite their low level differences with clocks.

Furthermore, consider that remapping the pizza neurons makes their behavior look almost identical to simple neurons when they are remapped, see Figure 46.

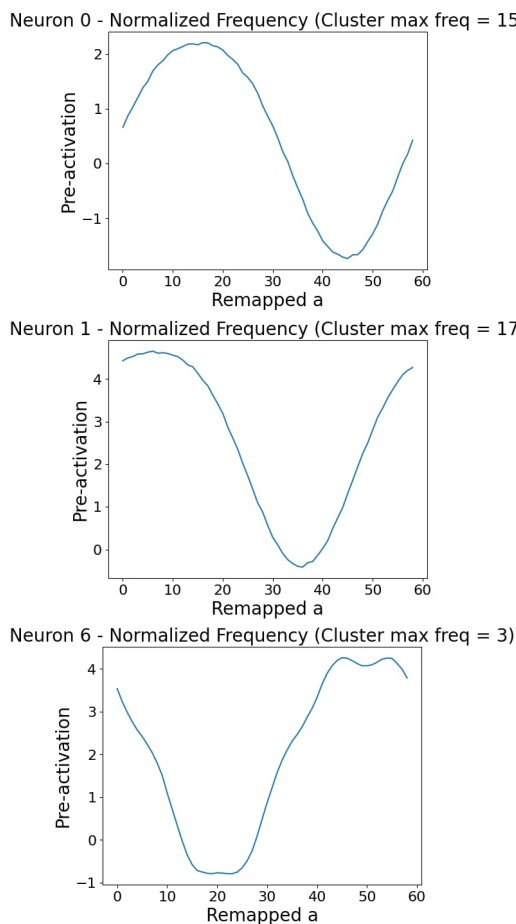

Figure 46: Remapping the pizza neurons shown in Figure 45 shows that they look identical to simple neurons.

### G.10.2 Clock model

Here we show that the approximate CRT in a clock model (Fig. 47) looks just like it does in a pizza model (Fig. 45).

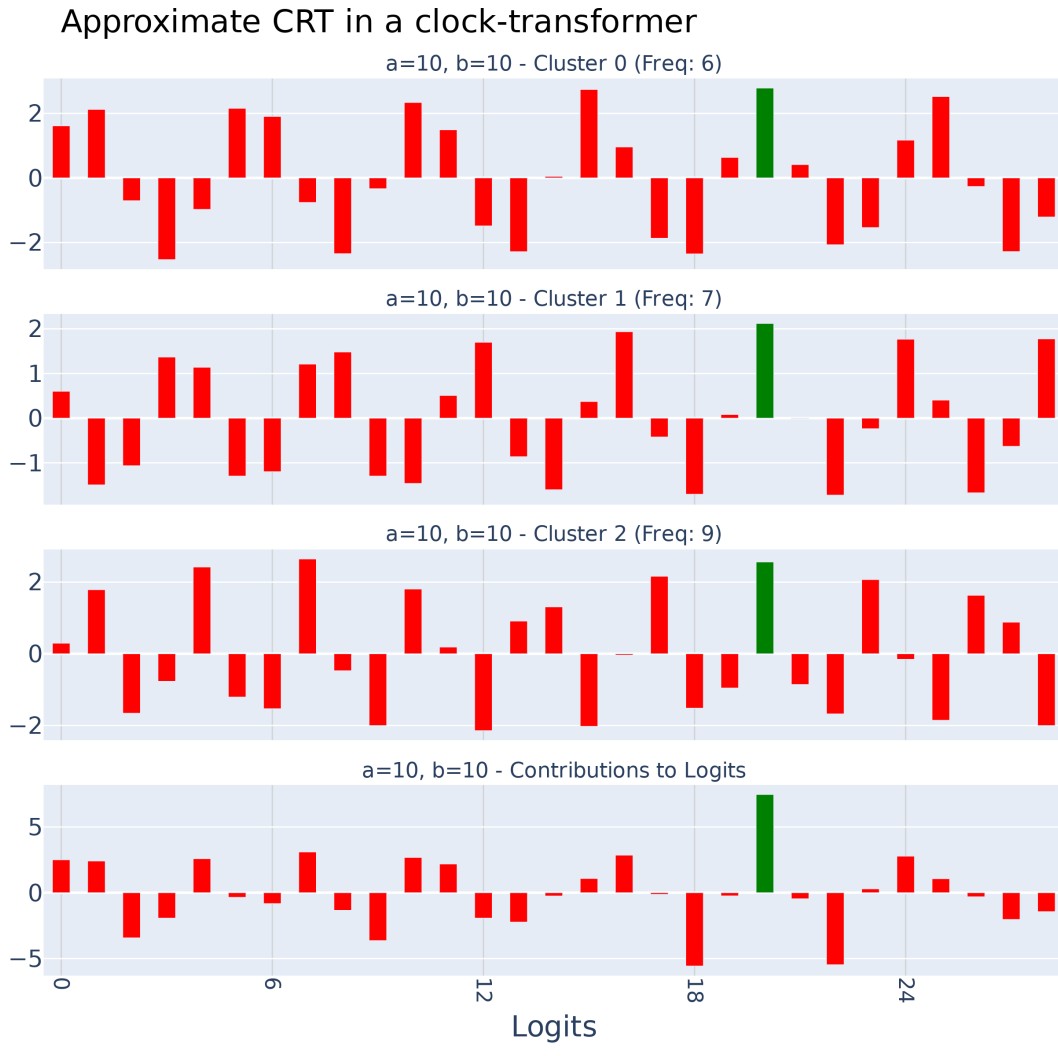

Figure 47: A view of the approximate CRT within a clock-transformer. We cluster all neurons together with the same frequencies, then inspect the cluster's contribution to each logit by summing the contributions of each neuron in the cluster to each logit. Note that the three clusters each contribute to the logits.

