# OpenReview forum: "Uncovering a Universal Abstract Algorithm for Modular Addition in Neural Networks"
_NeurIPS.cc/2025/Conference — NeurIPS 2025 poster_

### Official Review · Reviewer_hSU2 · 2025-06-24

**Clarity:** 2
**Significance:** 3
**Originality:** 4
**Rating:** 5
**Confidence:** 4

**Summary:**

The paper presents a universal algorithm -  through the notion of approximate cosets - to explain learned feature representations in models solving modular arithmetic.

**Questions:**

- Definition 4.2 doesn't clarify what $c, d$ are in the particular context. In particular, how does $d$ which is a function of $f, n$ relate to $c$?

- The authors show that the simple neuron model activates on approximate cosets with ReLU activation. How does this extend to smooth activations like tanh and quadratic?

- The assumption that training converges to simple neuron models, may not necessarily be true, as Ref. 6 in the paper (Fig 5) shows that the final trained model differs from the "simple neuron model". This is however not in conflict with the accuracy remaining unchanged in Fig. 6 of the paper, since the simple neuron model can give high accuracy but high MSE loss.

- In Section 4.4 "depth's effect on neural preactivations", what exactly do the authors mean by fitting cos(a+b), cos a + cos b ? What are the fitting parameters in each case?

**Ethical Concerns:**

["NO or VERY MINOR ethics concerns only"]

**Final Justification:**

The paper is significant and proposes a "top-down" approach to unify previous interpretability research in modular arithmetic.

**Limitations:**

- While the paper addresses applications to discrete group operations, insights on how to generalize this notion for arbitrary tasks, including continuous groups, is missing. For example, what is the equivalent of the "simple neuron model" for a continuous group like $U(1)$?

- Apart from that, I think the authors acknowledge other important limitations.

[1] Doshi, D., Das, A., He, T., \& Gromov, A. (2024). To grok or not to grok: Disentangling generalization and memorization on corrupted algorithmic datasets. In Proceedings of the International Conference on Learning Representations (ICLR). https://openreview.net/forum?id=UHjE5v5MB7

Overall, I feel the paper does a good job of tackling an important question, but could do with some improvements.

**Quality:**

3

**Strengths And Weaknesses:**

Strengths:
- The abstraction is useful since it unifies previous interpretations of learned features in modular arithmetic datasets, as well as provides a universal testable hypothesis.
- The authors perform extensive sets of experiments, across various architectures and range of moduli, which shows that their claims are universally true.


Weaknesses:
- It would be useful if Section 4.1, Approximate cosets, can be rewritten to distinguish between precise cosets and approximate cosets, maybe with an example.
- It wasn't immediately obvious why activation on approximate cosets can help the model solve modular arithmetic.
- Missing reference to previous work : the order-1 sinusoids in layer 1 was analyzed in [1], Section 2.2.

---

> ### Author Rebuttal · Authors · 2025-07-31
>
> Thank you for your thoughtful review. Your questions and comments reflected a clear understanding of our work’s originality, and we’re grateful for the opportunity to clarify several key points. We also appreciate your recognition of the value of our extensive experimentation and our unified theory reconciling competing interpretations in the literature.
>
> # Weaknesses
>
> Confident that we made a significant contribution, we included Appendix A to rigorously and intuitively explain the mathematics underpinning our framework. We believe Conjecture 4.9 is of substantial importance, and our goal is to make it accessible to a broad range of researchers. Notably, Appendix A directly addresses all three weaknesses raised in your review.
>
> > W1. It would be useful if Section 4.1, Approximate cosets, can be rewritten to distinguish between precise cosets and approximate cosets, maybe with an example.
>
> **We already provide worked examples in the main paper and in the appendix.**
> In the main paper:
> - Figure 2 provides 3 visual examples to distinguish neurons that learn precise cosets vs those that learn approximate cosets.
> In the appendix:
> - Appendix A.2 "Additional mathematical background" walks the reader through group theory and worked examples.
> - Appendix A.2.1 "Examples: cosets, Cayley graphs, step size d" builds on the intuitions developed in A.2 to ensure readers have the background to understand the main text.
>
> This should provide the relevant background to understand the visual examples in Figure 2. We will add a link to Appendix A.2.1 in the caption of Figure 2 to make this clearer to readers.
>
> **Elaboration to ensure your understanding.** Note in Figure 2, both coset and approximate coset neurons activate on half of the Cayley graph (panels 1, 4).
> The coset neuron (panels 1,2,3) activates strongest on the coset a mod 66 = 0, and second strongest on cosets a mod 66 = 1 and a mod 66 = 5 and 0 otherwise. Panels (4-6) show a neuron that learned an approximate coset centered on a = 0. The approximate coset neuron doesn’t have discrete level sets like the coset neuron, so it activates on the half of the Cayley graph centered on the group element it best understands. Thus, the approximate coset definition generalizes cosets to group elements that are close on the Cayley graph.
> Figure 2: This shows approximate cosets have no equivalence classes, i.e. the neuron doesn’t activate on nice level sets like in the coset case (panel 2 shows 4 level sets). We show coset neurons don’t have invertable remappings, so all the points collapse onto 6 equivalence classes (giving 4 level sets in panel 3), whereas approximate coset neurons have invertible remappings (bijective; panel 6). Also note, the 6 equivalence classes of the coset neuron are shown as circles overlaid on the circle graph in panel 1, and everything is color coded to make it clear where the points in each coset (or approximate coset) come from.
>
> > W2. It wasn't immediately obvious why activation on approximate cosets can help the model solve modular arithmetic.
>
> Please see Figure 10 in Appendix A.2.1. Now, suppose we have a 1 layer MLP with an embedding matrix. The preactivations of a neuron are the sum of two sinusoids, one for a and one for b. These two sinusoids always have the same frequency (as our R^2 fits support). Suppose that a neuron has the leftmost sinusoid for both a and b (on the x axis). Then the preactivations of this neuron receive cos(a) + cos(b). Since cos(0) = 1 and is the maximum, when a = b = 0, this neuron will output +2. Now, the weight connecting this neuron to the output logit at 0 will be large and positive. Thus this neuron used the coset information of a and b to activate strongly on the coset of the answer (0 + 0 mod 6 = 0).
>
> > W3. Missing reference to previous work : the order-1 sinusoids in layer 1 was analyzed in [1], Section 2.2.
>
> We actually do cite [1] in Appendix A.1. That said, we will add a citation to [1] in the main text where we introduce the simple neuron model.
>
> # Questions
>
> > Q1. Definition 4.2 doesn't clarify what c, d are in the particular context. In particular, how does d which is a function of f, n relate to c?
>
> Did you mean definition 4.3 since c is only introduced after definition 4.2? c can be thought of as the "center" or starting point, and where a neuron activates most strongly. d is the step size, a function of f, n as you point out and traces out (approximate) cosets that are close. These are independent of each other. d relates to distances and c just relates to a starting point. We can make this clearer in the manuscript.
>
>
> > Q2. The authors show that the simple neuron model activates on approximate cosets with ReLU activation. How does this extend to smooth activations like tanh and quadratic?
>
> If the neuron preactivations are still sinusoidal based periodic functions, our theory will transfer, e.g. the theoretical results of Gromov and Morwani et al., are covered by our theory.
>
> An example of our theory working a bit “out of distribution” can be seen in Figure 28 in Appendix G.4. Notably, G.4, G.5, G.6 study fine-tuning neurons that learn sawtooth functions. When neurons learn these functions, they break the interpretations of all prior work. We do not think their existence is practically relevant. This is because they only show up with bad hyperparameters: they would never be found in practice due to corresponding to poor local minima (hyperparameter tuning for good settings avoids them).
>
> > Q3. The assumption that training converges to simple neuron models, may not necessarily be true, as Ref. 6 in the paper (Fig 5) shows that the final trained model differs from the "simple neuron model". This is however not in conflict with the accuracy remaining unchanged in Fig. 6 of the paper, since the simple neuron model can give high accuracy but high MSE loss.
>
> Please see the answer to question Q2, particularly appendices G.4, G.5, G.6. The regions corresponding to low R^2 values are on the edge of where it’s possible to train a network. This is because this is the region that hyperparameters cause overfitting of the training set. Thus, neurons can learn sawtooth functions and become what we call “fine-tuning" neurons because they are “overfitting the training set”. Thus, when you replace them with a single sinusoid, the network still gets the correct answer because the extra “fine-tuning” they were doing was not critical to the network selecting the correct answer. We put this section in the appendix due to our comments above: we do not believe these are practically relevant since proper training setups will never find them. They can only be found by exhaustively searching with hyperparameters that are bad, but just good enough to learn.
>
> > Q4. In Section 4.4 "depth's effect on neural preactivations", what exactly do the authors mean by fitting cos(a+b), cos a + cos b ? What are the fitting parameters in each case?
>
> We are fitting the amplitudes (A_a, A_b) and phases (phi_a, phi_b) of two order-1 sinusoids, e.g. g(a,b) = A_a * cos((f * 2 * pi * a )/ n + phi_a) + A_b * cos((f * 2 * pi * b) / n + phi_b). We assume the frequency is the same for both sinusoids (which it is, or else our R^2 would be low and the prediction would not remain correct after replacing the neurons with the fit). We get the frequency f deterministically from the discrete fourier transform (DFT), i.e. it is not fit.
>
> Similarly, for order-2 sinusoids we fit the amplitude and phase, and take the frequency f from the DFT: g(a,b) = Acos(f*2*pi(a+b)/ n + phi).
>
>
>
> ## Limitations
>
> > While the paper addresses applications to discrete group operations, insights on how to generalize this notion for arbitrary tasks, including continuous groups, is missing. For example, what is the equivalent of the "simple neuron model" for a continuous group like U(1)?
>
> ​​We believe the generalization for a continuous group is projections of group representations. See Appendix A.2 for where we introduce group representations and Appendix D where we discuss projections of representations as a more general theory.
>
> # Final thoughts
>
> We once again thank you for your review. We hope in light of our clarifications you’ll consider increasing your score and we look forward to our continued discussion.

---

> ### Comment · Reviewer_hSU2 · 2025-08-05
>
> I thank the authors for their detailed explanation and for pointing out the appropriate sections to look at. However, I wanted to clarify that for
> > It wasn't immediately obvious why activation on approximate cosets can help the model solve modular arithmetic.
>
> my focus was on why *approximate* cosets can help solve modular arithmetic, as opposed to the precise coset example provided in the rebuttal.

---

> > ### Author Response · Authors · 2025-08-06
> >
> > Thanks for clarifying your question. We're happy to explain further and truly appreciate our continued discussion. Thanks a lot.
> >
> > The classical Chinese Remainder Theorem (CRT) solves modular arithmetic by utilizing O(log(n)) types of cosets: one type for each prime factor of the modulus. It’s the case that if every frequency learned by a neural network cleanly factorizes the modulus then **every neuron in the network has learned exact cosets**. In this case, the network has implemented the “sinusoidal Chinese Remainder Theorem” given as Remark 4.5 in Section 4.2. Furthermore, since there is a one-to-one correspondence with cosets and frequencies and the CRT has O(log(n)) types of cosets, the network must use O(log(n)) frequencies. Thus, such a network has implemented the classical CRT.
> >
> > This **does not happen across all random seeds** because neurons can learn a frequency that **does not divide the modulus**. In these situations, approximate cosets don’t just help: *they are necessary* to show that the network still performs operations analogous to the sinusoidal Chinese Remainder Theorem. Indeed, Theorem 4.7 gives that O(log(n)) random frequencies can solve modular arithmetic. This matches our scaling experiments on networks and moduli, and also matches the efficiency of the classical Chinese Remainder Theorem. Since approximate cosets are a generalization of cosets (all cosets are approximate cosets), they are *necessary* in order to derive algorithm 4.6.
> >
> > It follows by construction from Theorem 4.4 (*neurons activate only on approximate cosets*) and Theorem 4.7 (*O(log(n)) random frequencies are needed*) that **all networks** trained on modular addition are implementing Algorithm 4.6. This is empirically validated across a substantial breadth of experimental settings that prior work did not validate across. Resultantly, we discover counter-examples to prior work (fine-tuning neurons Fig. 28 Appendix G.4) that activate on approximate cosets, **but break the interpretations of all prior work**.
> >
> > Importantly, we emphasize that the notion of approximate cosets is not ad hoc, but is motivated by and grounded in real structural considerations. As Figure 2 shows, even when the frequency doesn’t divide the modulus, approximate cosets still encode algebraic structure. Indeed, this is why approximate cosets are so powerful and why they allow us to prove theorems of surprising strength. To our knowledge, this is the first rigorous result perfectly matching empirical results in *deep* networks on a problem that’s not linearly separable (we prove: O(log(n)) frequencies are needed and empirically verify it in scaling experiments with R^2 scores >=0.99 **in both MLPs and transformers**!).
> >
> > In light of these clarifications, we sincerely hope that you consider increasing your score to reflect that our framework unifies 5 preceding papers under one concise and all-encompassing theory while simultaneously providing fine-tuning neurons as a counter-example to the models of all prior work. Also, we provide the first theory on modular addition for deep networks; previously, the best theoretical bound addressed 1-layer MLPs (Morwani et al. and Gromov) and is O(n) frequencies.

---

> > > ### Comment · Reviewer_hSU2 · 2025-08-06
> > >
> > > Thanks for the detailed explanation. I have increased my score accordingly.

---

### Official Review · Reviewer_RppL · 2025-06-29

**Clarity:** 3
**Significance:** 2
**Originality:** 3
**Rating:** 5
**Confidence:** 4

**Summary:**

The paper introduces a unifying theory to show how the model learn to sum two numbers mod $n$.

They introduce the aCRT idea and show that NN model requires $O(\log n)$ features, which is the upper bound of the number of prime factors which divides $n$. aCRT theory in conjunction with the experimental section shows also why NN networks solve modular addition.

**Questions:**

Q1. Do you have an explanation of why 1 layer model cannot learn this? Is it related to the famous paper from Antrophic about induction head and non-learnability of the copy task from 1 layer model?

Q2. Going from $O(\log n)$ to $O(1)$ features is extremely important. Can we find some particular versions of NN with some form of induction heads to effectively use less features?

**Ethical Concerns:**

["NO or VERY MINOR ethics concerns only"]

**Final Justification:**

Authors justified and explained the minor weakness and the questions I had.

**Limitations:**

yes

**Quality:**

3

**Strengths And Weaknesses:**

S1. The most important contribution is how authors unified the theory behind modular arithmetic disseminated across multiple works into a single theory. Also, although I didn't check all the details of the theorems, it sounds pretty right.

S2. Authors opened a new route on the mechanical interpretability, without looking only at the model weights only or PCA style analysis, but more holistically at the entire model itself.

S3. The experimental section is just enough to show results that are coming from the theory.

W1. I suggest authors to change the "spectacular" wording used in the abstract: I am not sure this understanding cannot be easily extended to other modular tasks, especially when other operations are introduced.

Observation. I probably need more explanations on how model goes from coset to the final output representation, it's not super clear from the writing. It's written in Remark 4.5 that "the argmax of ... selects the correct answer", but I am interested to clearly understand the mechanism to select the correct answer.

---

> ### Author Rebuttal · Authors · 2025-07-31
>
> Thank you for your review. We're especially glad you identified our unification of prior work across theory, interpretation, and architectures as the central contribution. That was precisely our goal. Prior interpretations were often cited as evidence *against* universality, presenting a fragmented picture; we show that they are simply different implementations by different architectures of the same algorithmic strategy, what we call an **abstract algorithm**. We also appreciate your recognition of our holistic, network-level analysis, and that our theoretical predictions were validated empirically.
>
> # Weaknesses
>
> > W1. I suggest authors to change the "spectacular" wording used in the abstract: I am not sure this understanding cannot be easily extended to other modular tasks, especially when other operations are introduced.
>
> We thank the reviewer for this feedback. While the phrasing is somewhat ambiguous, we interpret it as questioning the generalizability of our findings to other modular or non-group tasks. If there are specific phrases in the abstract that appear overstated or unclear, we would welcome concrete suggestions. That said, we’d like to clarify that the abstract is intended to reflect both the core contributions of our work and the scientific philosophy behind them.
>
> You identified the unification of prior work as the most important contribution, and we agree. Unifying previously incompatible interpretations was not only novel but necessary: without understanding the algorithmic structure in modular addition, a task once seen as fragmented, there is little hope of generalizing to more complex settings.
>
> This resolution reframes modular addition: what were thought to be incompatible solutions now reveal a single underlying algorithm implemented differently across architectures. This provides a solid footing for future work on algorithmic understanding.
>
> The abstract was designed to communicate how this unification was achieved through hypothesis-driven research. We began with a testable hypothesis: that neural networks learn an approximate Chinese Remainder Theorem (aCRT) on modular addition (lines 95–96). That hypothesis drove the rest of our investigation, which delivered:
> - A unifying algorithmic structure (aCRT) across three architectures and all hyperparameter settings, including depth
> - Multi-level analysis (neurons, clusters, networks) that verified this structure
> - A theoretical prediction (an O(log n) feature bound) validated empirically
> - A testable universality hypothesis for group tasks, supported by our insights and theory derived from this process
> This work demonstrates that interpretability can follow the scientific method: make hypotheses, derive consequences, and validate them. That process enables interpretability to yield generalizable, falsifiable insights. On this foundation, we can hopefully move towards more complex tasks. But before this unification, it was certainly not obvious.
>
> We do *not* claim that our findings trivially extend to non-group tasks. But having a fitting interpretation of how networks solve modular addition is an important stepping stone in this search.
>
> Finally, we note that **reviewer EPiJ, acknowledged the potential of the work’s broader impact**:
> "the work stands to be highly significant"
> and "this work will likely inform a lot of subsequent work and provide a useful perspective on the emergent behaviour of neural networks on algorithmic tasks."
>
> We hope this clarifies the tone of the abstract. We are happy to revise its wording for clarity, but we stand by its core message as a faithful reflection of our contributions and approach.
>
> ## A recent position paper gives context: our work is timely
>
> The ICML 2025 position paper, “We Need an Algorithmic Understanding of Generative AI” [1], argues that the ML community must move beyond fragmented, bottom-up interpretability, ​​which often focuses on individual neurons, components or circuits without a guiding hypothesis. Instead, they propose the prioritization of top-down hypothesis/theory-driven research that explains how models implement full algorithms. The authors write in reference to prior approaches: "current findings are still largely fragmented, and we lack a solid theoretical foundation for understanding how these various components come together to implement algorithms." Furthermore, they argue that such an approach would close the theory-interpretability gap, since both communities are operating rather disjointly. And this would lead to better understood, safer, and more efficient models.
>
> Our work is timely and directly answers this call by providing such a theoretical foundation in the well-studied setting of modular addition, which up to this point had fragmented findings. We offer a clear instantiation of several core elements that [1] proposes as future goals:
>
> | [1] proposes    | Our work |
> | -------- | ------- |
> | Identify algorithmic primitives  | approximate cosets    |
> | Compose primitives into algorithms | approximate CRT    |
> | Testable computational hypotheses grounded in theory    | 1. Theorem 4.7: predicts O(log(n)) features are required by deep networks on modular addition 2. Conjecture 4.9: containing many diverse tasks of interest, e.g. sorting lists   |
>
> # Questions
>
> > I probably need more explanations on how model goes from coset to the final output representation, it's not super clear from the writing. It's written in Remark 4.5 that "the argmax of ... selects the correct answer", but I am interested to clearly understand the mechanism to select the correct answer.
>
> The network implements a divide and conquer algorithm that is distributed across the neurons. Figure 2 in the main paper demonstrates that each neuron that learned a coset or approximate coset only activates on half of the Cayley graph (first and fourth panels). This means that each neuron acts as a question “Which half of the Cayley graph is the answer in?”. Since different cosets and approximate cosets have different frequencies, they correspond to different graphs. Thus, Theorem 4.7 presents that only log(n) graphs are needed in order to divide and conquer in this way to get the correct answer while minimizing the loss sufficiently by Corollary 4.8, which gives that each additional frequency pushes the value on incorrect logits down exponentially. Therefore, the algorithm the network implements requires log(n) different graphs (different approximate cosets) and acquires the answer by asking log(n) questions: which half of the graph is the correct answer in.
>
> Essentially, we’ve so far explained the post-activations, as the ReLU ensures the negative values are 0 on all vertices in the Cayley graph that are farthest away (in graph distance) from the element at the center of the approximate coset. After this non-linearity, the problem has become linearly separable and achieves the most efficient solution possible due to the divide and conquer strategy. Each neuron can now have its activation projected to the half of the logits that are closest to the coset element the neuron best understands.
>
> > Q1. Do you have an explanation of why 1 layer model cannot learn this? Is it related to the famous paper from Antrophic about induction head and non-learnability of the copy task from 1 layer model?
>
> Yes. The reason 1-layer models learn O(n) features instead of O(log(n)) features is because 1 layer models do not implement the divide and conquer algorithm explained in the previous answer. Instead they learn something much more in the flavour of a dynamic program: they learn every frequency and use the frequencies as a look up table in order to get the right answer. Yes, this directly relates to Anthropic's work on the induction head and builds on it by presenting a novel idea: deep transformers learn implementations of divide and conquer algorithms (we showed this in MLPs also).
>
> > Q2. Going from O(logn) to O(1) features is extremely important. Can we find some particular versions of NN with some form of induction heads to effectively use less features?
>
> Can networks be more efficient, i.e. can they do better than O(log(n)) features and learn O(1) features? Theorem 4.7 tells us that it’s possible for a network to learn this task with O(1) features, but the network will have very small margins (e.g. 1e-3) between the correct logit and second largest, but incorrect logit. This means the cross entropy loss will not be minimized, and thus the network will try to learn more features. This is why we performed the scaling experiment in Figure 3, which shows that for all moduli from 3 to 4999, neural networks of all architectures learned on average O(log(n)) features.
>
> Indeed, a satisfying aspect of our work shows how feature efficiency is exponentially increased from O(n) to O(log(n)) by depth.
>
> # Final thoughts
>
> We thank you again for your review. We hope this addresses your concerns clearly. If so, we would greatly appreciate a reconsideration of your score.
>
> ## References
> [1] Eberle et al., "Position: We Need An Algorithmic Understanding of Generative AI", ICML 2025.

---

### Official Review · Reviewer_2YER · 2025-06-30

**Clarity:** 3
**Significance:** 2
**Originality:** 3
**Rating:** 4
**Confidence:** 3

**Summary:**

This paper examines the mechanism by which neural networks solve the modular addition task and advances a testable universality hypothesis. While prior studies have attributed diverse internal representations—such as the block and pizza algorithms—to fundamentally different learned solutions, the authors argue that these can be unified under a single abstract algorithm, namely the approximate Chinese Remainder Theorem (aCRT). A key theoretical contribution is the derivation that deep neural networks (DNNs) require only $O(\log n)$ features to solve modular addition, matching the empirical results.


**Overall Recommendation**
The paper presents an interesting attempt to unify multiple mechanisms by which DNNs solve modular addition tasks. However, it lacks a clear articulation of how the proposed aCRT framework differs from and subsumes prior interpretations. Providing a clearer intuition explanation  would substantially enhance the contribution and may warrant a higher score.

**Questions:**

- Could the proposed explanation be extended to modular addition tasks with longer length, i.e., $a_1 + \cdots+a_k (\text{mod}   p)$?
-  Could the authors clarify, perhaps through a simple example, how the approximate Chinese Remainder Theorem is applied to modular addition? Traditionally, the Chinese Remainder Theorem addresses systems of congruences rather than modular addition itself.

**Ethical Concerns:**

["NO or VERY MINOR ethics concerns only"]

**Final Justification:**

The authors have addressed some of my concerns in their revisions. As a result, I am raising my score to 4.

**Limitations:**

yes

**Quality:**

3

**Strengths And Weaknesses:**

**Strengths**

- Clear and straightforward setup

- The paper offers a conceptually compelling attempt to unify disparate neural network mechanisms for modular addition within a single theoretical framework, called the approximate Chinese Remainder Theorem. The motivation and ides are interesting.

- In illustrating how many frequencies are needed to instantiate the aCRT, Figure 3 provides strong empirical support for the theoretical result established in Theorem 4.7.

**Weakness**
-  The formal definitions (Definition 4.1, 4.2 and 4.3 ) are difficult to follow and lack sufficient intuitive grounding. To enhance clarity and accessibility, it is recommended that the authors illustrate key concepts—such as step size, frequency normalization, and approximate sets—through simplified, concrete examples. In addition, the manuscript would benefit from a clearer articulation of the motivation and underlying intuition behind these definitions.
- From the reader’s perspective, the proposed explanation based on the approximate Chinese Remainder Theorem offers an alternative perspective on how neural networks solve the modular addition task. However, it remains unclear whether this explanation genuinely subsumes prior interpretations—such as the pizza and clock algorithms—or merely stands as another competing account. The manuscript lacks a clear theoretical or intuitive justification for why these previously observed mechanisms can be interpreted as specific instances of the proposed abstract algorithm. To strengthen the unifying claim, the authors are encouraged to provide deeper theoretical insights or illustrative examples demonstrating how and why these distinct algorithmic patterns emerge as manifestations of their framework.
- The authors draw conclusions based on a narrow set of experimental conditions, while the phenomena they aim to unify were originally observed in diverse settings.  The paper would be more convincing if it provided experimental evidence from the original settings used by others and demonstrated that the coset explanation holds consistently across different settings for the same task.

---

> ### Author Rebuttal · Authors · 2025-07-31
>
> Thanks for reviewing our paper! We appreciate the time you took to engage with our ideas. We're glad you found our unification via the approximate Chinese Remainder Theorem (aCRT) conceptually compelling, and that you highlighted the clarity of our setup and the strength of the empirical support for our theoretical claims (e.g., Figure 3, Theorem 4.7). We're encouraged by your openness to increasing the score with further clarification.
>
> # Weaknesses
>
> > The authors draw conclusions based on a narrow set of experimental conditions, while the phenomena they aim to unify were originally observed in diverse settings. The paper would be more convincing if it provided experimental evidence from the original settings
>
> We respectfully disagree that our conclusions rely on narrow experimental conditions and believe there may be a misunderstanding. Our analyses both replicate prior setups and substantially expand their scopes through a far more rigorous evaluation framework. Specifically, we vary key hyperparameters: depth, width, learning rate, L2 regularization, batch size, and training seed, training over 1 million networks in total. To our knowledge, no prior work has conducted such large-scale empirical verification. These experiments show that our theory holds robustly across diverse training settings.
>
> To demonstrate the breadth of our experiments, and how our theory subsumes prior work, we highlight key empirical findings that go beyond all existing models:
> - Counterexamples to prior theories: We show that small batch sizes prevent the formation of representation matrices (Figure 13), producing clear counterexamples to the interpretation proposed by Chughtai et al. These are detailed in Appendix D.
> - Previously undocumented behavior (sawtooth functions): We identify a new class of representations, sawtooth functions, which arise at the boundary between successful and failed training. These functions are not captured by prior theories but are accounted for by ours, as they activate on approximate cosets (Appendices G.4-G.6).
> - Reanalysis of prior models: We directly evaluate publicly released checkpoints from Zhong et al. (“pizza” and “clock” transformers; see Appendix E.1). Using their methodology, we extend the analysis and resolve their open questions about circular embeddings. Specifically, we show that models do not learn non-circular embeddings, and that their proposed circularity metric is unnecessary (Appendix G.1).
> - In Appendix G we run 26 pages of additional results
>
> These results reveal where previous interpretations fail and demonstrate that our framework is robust. For additional comparisons to prior work, please refer to our response to reviewer 1.
>
> > The formal definitions (Definition 4.1, 4.2 and 4.3 ) are difficult to follow and lack sufficient intuitive grounding. To enhance clarity and accessibility, it is recommended that the authors illustrate key concepts—such as step size, frequency normalization, and approximate sets—through simplified, concrete examples. In addition, the manuscript would benefit from a clearer articulation of the motivation and underlying intuition behind these definitions.
>
> In the main text we provide examples of these key definitions; please see Section 3 and for illustrations of the key definitions please see Figure 2. For “simplified, concrete examples” please see Appendix A.2 titled “Additional Mathematical Background”. For more examples and “a clearer articulation of the motivation and underlying intuition behind these definitions.” see A.2.1 titled “Examples: cosets, Cayley graphs, step size d”.
>
> > It remains unclear whether the aCRT explanation genuinely subsumes prior interpretations (e.g., pizza and clock algorithms), or merely stands as another competing account.
>
> We believe the evidence provided demonstrates that the aCRT framework does not merely offer an alternative explanation; it subsumes all prior interpretations. Unlike previous work, which presents disjoint, architecture-specific theories (e.g., Nanda et al., Zhong et al., Morwani et al.), our theory offers a single, mathematically grounded explanation that holds across all tested architectures and training regimes.
>
> This work represents a substantial contribution because it unifies previously disjoint theories, and introduces a new line of thinking. Prior to our study, the field had begun to accept that different architectures learn fundamentally different circuits, in part due to conflicting interpretations in the literature. For example, Zhong et al. has received significant attention for highlighting such divergence, leading to broader concerns that mechanistic interpretability may not generalize across architectures.
>
> This concern is echoed in papers citing Zhong et al., including MoSSAIC: AI Safety After Mechanism (Farr et al.), which writes: "Researchers have successfully reverse-engineered these models and discovered that they learn a specific, intricate algorithm based on trigonometric identities and Fourier transforms, implemented via clock-like representations in the attention heads. The specific implementation is fundamentally tied to the architectural properties of the transformer substrate. A different architecture would almost certainly learn a different algorithm, rendering this detailed explanation obsolete."
>
> Our findings directly challenge this conclusion. We show that different architectures do not learn fundamentally different algorithms; rather, they learn distinct implementations of a shared abstract strategy, formalized by the aCRT. In this sense, our framework reveals a common algorithmic structure underlying previously conflicting interpretations.
>
> This insight also aligns with the argument in “Position: We Need an Algorithmic Understanding of Generative AI” (Eberle et al., ICML 2025), which emphasizes the need for theoretical foundations that explain how neural components work together to implement algorithms. We believe our paper is the first to rigorously demonstrate that neural networks can instantiate abstract algorithms in their learned weights. In doing so, we contribute two key insights:
> 1. While it is widely hypothesized that neural networks may learn algorithms, we offer the first formal demonstration that they instantiate an abstract algorithm.
> 2. Proving such behavior requires a new theoretical framework, one that characterizes the learned solution at an abstract level, as our notion of approximate cosets enables.
>
>
> > The manuscript lacks theoretical or intuitive justification for why these previously observed mechanisms can be interpreted as specific instances of the proposed abstract algorithm. To strengthen the unifying claim, the authors are encouraged to provide deeper theoretical insights or illustrative examples demonstrating how and why these distinct algorithmic patterns emerge as manifestations of their framework.
>
> In section 4.1 we introduce approximate cosets. These are more general than cosets, as every coset is an approximate coset, but approximate cosets also include modular subsystems that do not perfectly divide the modulus. Then we provide Fig. 2 as a visual example to help the reader geometrically understand the approximate coset definition. Then, we introduce Theorem 4.4, which states: a ReLU neuron only activates if it is activating on an approximate coset. Later, section 4.2 provides a step-by-step guide on how to build the abstract approximate Chinese Remainder Theorem (CRT) algorithm utilizing Theorem 4.4.
>
> This is done by first guiding the reader through a construction (Remark 4.5) that creates an abstract classical CRT algorithm that neural nets utilize only if they learn frequencies that are divisors of the modulus. The classical CRT (as you ask later in your review) only works for non-prime moduli because it utilizes cosets which only exist if divisors of the moduli exists, but the approximate CRT will work for frequencies that don’t divide the modulus. Thus, to build the more general approximate CRT from the classical CRT (Remark 4.5), we only need to relax one thing: we replace cosets by approximate cosets. This gives Abstract Algorithm 4.6: the approximate CRT.
>
> The theoretical justification: By Theorem 4.4, we know for a fact that all neurons activate only on approximate cosets. We know from number theory that a number can have at most O(log(n)) prime factors (e.g. consider 2^n, which has the factor 2 n times). So we must prove the aCRT can use O(log(n)) frequencies. Indeed, this is Theorem 4.7, and we empirically verify that networks have logarithmic growth rates with R^2 scores >= 0.989 in Figure 3 on moduli ranging from 3 to 4999.
>
> The empirical justification: As explained above, the results in the paper show beyond any reasonable doubt in Figures 3-9 that across very diverse training settings our theory always explains the weights networks converge to. No previous paper tested their interpretation under as broad of a variety of circumstances or reported this.
>
> # Questions
>
> > Could the proposed explanation be extended to modular addition of longer length i.e., a_1+a_2+...+a_k (mod p)?
>
> Yes. Simple neurons become a sum of k sinusoids, one for each a_i.
>
> > Perhaps through a simple example, how is the approximate Chinese Remainder Theorem is applied to modular addition? Traditionally, the Chinese Remainder Theorem addresses systems of congruences rather than modular addition itself.
>
> Please see Appendix A.2 dedicated to examples, particularly A.2.1, which visually illustrates how the Chinese Remainder Theorem (CRT) could be computed by a network that learned sine functions in its neurons. After seeing this, we trust it will be clear how approximate cosets can replace cosets, giving our core result: the approximate CRT.
>
> # Final thoughts
>
> We appreciate your suggestions to improve clarity and your openness to increasing the score. We hope this rebuttal addresses your concerns and are happy to provide further clarifications.

---

> ### Author Response · Authors · 2025-08-06
>
> Hi Reviewer,
>
> As the discussion period is wrapping up, we wanted to check in and see if you need any further clarifications from us. You had mentioned that you might consider increasing your score if your questions were addressed, and we’ve done our best to respond to your concerns. Please let us know if there's anything else we can help clarify.

---

### Official Review · Reviewer_EPiJ · 2025-07-02

**Clarity:** 3
**Significance:** 3
**Originality:** 3
**Rating:** 5
**Confidence:** 3

**Summary:**

This work presents a theory of how ReLU neural networks (MLPs and Transformers) learn to perform modular addition. The algorithm implemented by the network is termed the approximate Chinese Remainder Theorem (aCRT). This is achieved by forming structures called approximate cosets - a generalisation of cosets which define equivalence classes within the cyclic group defined by the modulo operator. The generalisation to approximate cosets then lets sets of "close" cosets be treated equivalently. Empirical experiments support that neurons in networks trained to perform modular addition can be replaced by "simple neurons" - neurons specialised to specific frequencies - without a significant drop in performance, support the notion that the networks are learning the aCRT.

**Questions:**

1. What qualitative differences should I be noting in Figure 1.
2. Why is an $R^2$ score of $0.3$ acceptable in Figure 7 as long as the green tick is there?
3. How does the presence of a green tick in Figure 6 and 7 corresponding to the actual behaviour or representations of the network.
4. Why is the maximum output logit over the entire dataset used in Theorem 4.7 and is there no tighter bound for this?

I would definitely be inclined to raise my score if some of the questions were answered and it turns out I have missed something important.

**Ethical Concerns:**

["NO or VERY MINOR ethics concerns only"]

**Final Justification:**

The authors provided a very detailed response to my review and answered my questions. With these points made clear I am happy to advocate for acceptance and thing the proposed changes will benefit the clarity of the work.

**Limitations:**

Overall the assumptions of the work are clearly stated and the Discussion section is fair and stated the necessary limitations.

**Quality:**

3

**Strengths And Weaknesses:**

# Strengths
## Quality
The work does a good job of covering the theoretical background on group theory. Particularly sections 1, 2 and 3 are particularly effective at contextualising the work and also introducing the technical details of modular arithmetic. The assumptions necessary for the theory are stated clearly and the design of the experiments is well suited to establishing the validity of the assumptions. Precise definitions are given for the necessary concepts. Overall, the goals of the work are clear and the theory presented is precise with the necessary empirical support to justify the framework.

## Clarity
I reiterate that I think the first few sections (up to Section 4) are very well done and support clarity immensely. Figures, particularly Figure 2, are helpful with sufficiently detailed captions to aid understanding. I also appreciated the helpful analogies which help broaden the applicable audience of the work by given more readers something to related to - for example the analogy to the breadth-first search on Lines 161 to 168. The notation used is intuitive and consistent which aids clarity.

## Significance
The claimed findings of this work unify a number of different perspectives on a topic which has received quite a bit of recent interest. Thus, the work stands to be highly significant. I do have some questions about the precision of the framework, but taking the claims on their own merit then I think this work will likely inform a lot of subsequent work and provide a useful perspective on the emergent behaviour of neural networks on algorithmic tasks. I also find the comparison between MLPs and Transformers here useful and the general lack of a very clear behavioural difference is interesting as I usually only associate these sorts of tasks (and mechanistic interpretability broadly) with Transformers.

## Originality
Similar to the point on significance, I think that the combination of recent ideas from the literature being combined is original and results in a potentially powerful framework for understanding the sub circuits responsible for modular addition. The approximation to the CRT is also new and so there is a decent amount of originality overall. The originality is also supported by the detailed background in Section 2 which helps make it clear what is new to this work.

# Weaknesses
## Clarity
Section 4 and the following sections do lose some of the clarity which was fantastic up to that point. For example, the equations on the bottom of page 3 are foundational to the rest of the work (introducing the very concept of a simple neutron) but introduced quickly in comparison to the previous concepts. I also just found the notation of $\omega(A,B)$ to be a bit unintuitive as it is the dot product of an embedding $U$ with a set of weights not depicted as parameters to the function. Secondly, Theorem 4.7 and Corollary 4.8 are the main theoretical results and yet they are stated quite bluntly, with little attempt to provide intuition or any sort of proof sketch. Some intuition on why is the key equation of interest here $m' -h(k) >\delta m'$ would be very helpful. As a minor point I also think writing this as $m'(1-\delta) > h(k)$ would be more intuitive and in line with the point of the theorem.

## Quality
Some of the results of the theory appear to be true by construction or at least the conclusions drawn seem to not be wholly justified. I am hoping these concerns are more a consequence of my own misunderstanding. Firstly, Figure 1 shows the preactivations of a neuron where the frequency has been normalised. It is claimed that this figure shows the qualitative equivalence of the neurons, but to me everything about it seems different except for the frequency. The range of values and the phase are different, so it is unclear what qualitative property I should be focusing on here. Secondly, the general approach of Section 4.4 is a bit of a concern, but also the statement on Lines 188 and 189 which says that all architectures trained to perform modular addition are abstracted by approximate cosets. I am not certain what the alternative would be - is this not true by virtue of the network being trained to perform modular addition itself? This alone does not seem to be a tight enough correspondence to draw conclusions about the actual circuitry of the network itself. Relatedly, in Figure 7 for example there are points where the $R^2$ value is as low as $0.2$ to $0.3$ between the network and simple neutron replacement but then there is still a green tick (this occurs in Clock: $1^{st}/4)$ for example. The green ticks presenting no loss in accuracy is used as the key point demonstrating that simple neurons are a good abstraction but this just makes me thing the green arrows are not a precise enough measure of success when it can have such as high range of $R^2$ values. Finally, in Theorem 4.7, it appears strange to me that the bound is given in terms of the maximum output logic across the dataset ($m'$) when it is compared to individual output logics $(h(k))$. I appreciate that the point is to show that sufficiently large bounds exists between the maximum logit and all possible incorrect logins, but the fact that this is not done per data point seems to make this quite imprecise. I suspect this question has more to do with my difficulty with the clarity of this portion of the paper.

---

> ### Author Rebuttal · Authors · 2025-07-31
>
> Thank you for your thoughtful and detailed review. We're excited that you recognized the originality and significance of our framework. We appreciate your compliments on the clarity and precision of the early sections. Your summary reflects a strong understanding of our goals and we’re glad that you thought our work stands to be highly significant. We think your concerns stem from areas we can readily clarify and are encouraged that you indicated you could raise your score if we succeed.
>
> # Weaknesses
>
> Thanks again for praising the clarity in sections 1, 2, 3 of our paper. We think that by addressing your points and incorporating your suggestions here, we can clarify the difficult sections of the paper.
>
> ## W1. Clarity and Presentation
>
> **Drop in clarity in section 4.** Due to lack of space, we agree that some things were introduced tersely.
> - We will revise the simple neuron equations to be more prominently presented and clarified, and revise the notation to reflect their dependence on network weights.
> - We will add intuition for Theorem 4.7 and Corollary 4.8, though we may relegate the proof sketch to the appendix as we don't think it's strictly necessary in the main text.
> - We agree with you to state Theorem 4.7 as $m’(1-\delta) > h(k)$. Thank you for the suggestion.
>
> ## W2. Quality
>
> The concerns and questions you raise here appear to be in direct correspondence with Q1, Q2, Q3, and Q4, so we address them in the Questions section.
>
> # Questions
>
> > Q1. What qualitative differences should I be noting in Figure 1
>
> - **Fig. 1 should qualitatively show the effect of remapping.** Neurons with different frequencies in MLPs, clocks and pizza transformers look like sine functions, but appear to have slight qualitative differences, e.g. MLP looks less structured compared to pizza, but after remapping, they’re qualitatively both sine functions with frequency 1.
> - Due to our exhaustive search over training settings, we found neurons missed by prior work that qualitatively looked like sine functions, yet $R^2$ scores implied they weren’t quantitatively.
> - After remapping, we saw these neurons had learned sawtooth functions, not sine functions.
> - Sawtooth functions are a sum of sine functions with different frequencies. Note: the first sine function in the sum explains some of the variance, so a sine function of best fit captures the key quantitative properties.
> - These neurons emerge at the boundary of generalization: when hyperparameters are set before the network fails to generalize to 100% test accuracy in training.
> - Replacing all neurons (including sawtooth neurons) with sine functions of best fit doesn’t change the networks test accuracy because the extra sine functions they learned were helping them overfit the training set—they were not relevant for generalization.
> - We study various properties of these neurons in Appendices G.4, G.5 and G.6.
> - Of note is Figure 28 in G.4 that shows how remapping can qualitatively show if a different function is learned. Figure 30 shows that the sawtooth shown in Figure 29 is a sum of sine functions with frequencies {42, 35, 28, 21, 14, 7}.
> - We aren’t sure if these are of practical relevance—they would never be found by a deep learning practitioner hypertuning for good parameters. Indeed, this is why prior works missed them.
>
> Thus, we will make the caption of Fig. 1 point to appendix G.4, Figure 28, so that interested readers can see how remapping can qualitatively discern different functions.
>
> > Q2. Why is an $R^2$ score of 0.3 acceptable in Figure 7 as long as the green tick is there?
>
> Because the green tick indicates 100% test accuracy was preserved after replacing all neurons with their sine fits. Even if the $R^2$ of individual neurons is low (e.g., for sawtooths), the network’s behavior remains unchanged, showing sawtooths aren’t critical to generalization out of the training data.
>
> > Q3. How does the presence of a green tick in Figure 6 and 7 corresponding to the actual behaviour or representations of the network.
>
> The green tick shows whether the network gets the correct answer after we replace all neurons with their best fits. It serves to elucidate the boundary cases, where hyperparameters are bad, but not so bad as to prevent generalization.
>
> Elaboration on weakness. The paper states: all architectures trained to perform modular addition are abstracted well by approximate cosets. You ask: I am not certain what the alternative would be - is this not true by virtue of the network being trained to perform modular addition itself? This alone does not seem to be a tight enough correspondence to draw conclusions about the actual circuitry of the network itself.
>
> There are two questions here and the first one is truly profound. The answer is no: it’s not true simply by virtue of the task. The second question: “This alone does not seem to be a tight enough correspondence to draw conclusions about the actual circuitry of the network itself.” is actually our goal. We are not making claims about the circuitry. We are abstracting the specific details of the circuitry away with a data structure—paths on Cayley graphs called approximate cosets. We use this abstraction to rigorously prove that disparities in prior interpretations occurred due to weights of different architectures converging to different implementations of one divide and conquer algorithm that we give: abstract algorithm 4.6.
>
> Our idea to abstract away circuits is very timely. A recent ICML 2025 position paper, “We Need an Algorithmic Understanding of Generative AI” [1], argues that the ML community must move beyond fragmented, bottom-up interpretability, ​​which often focuses on individual neurons, components or circuits—without a guiding hypothesis. Their position is to prioritize top-down hypothesis/theory-driven research that explains how models implement full algorithms. The authors write in reference to prior approaches and even cite the works we unify: "current findings are still largely fragmented, and we lack a solid theoretical foundation for understanding how these various components come together to implement algorithms." Furthermore, they argue that such an approach would close the theory-interpretability gap, since both communities are operating rather disjointly.
>
> Our idea to abstract away circuit level details gave two contributions that we believe are substantial. 1) we address the call to close the theory-interpretability gap and provide a theoretical framework and show how to utilize abstraction to form an algorithm out of learned circuits. 2) To our knowledge, we are the first to prove that the solutions learned by stochastic gradient methods can converge to implementations of divide and conquer algorithms by providing an example of it happening. Our Conjecture 4.9 gives a route to proving that neural networks learn implementations of divide and conquer algorithms on many tasks.
>
> Back to the profound question: the community has no theoretical reason to believe the network should prefer to learn cosets or approximate cosets. Recall our approximate coset definition is a path on a Cayley graph. Since neurons only activate on approximate cosets, each neuron activates if the answer to the question: “is the answer in my part of the Cayley graph?” is yes. But why learn approximate cosets? Ultimately, approximate cosets are functions on the Cayley graph. Any function can pick elements on the Cayley graph to serve as the foundation for these questions. Indeed, approximate cosets tend to divide a Cayley graph in half—this is in fact why we conjectured a O(log(n)) bound and later proved it. But many functions could be learned that divide the Cayley graph in half!
>
> The goal of Morwani et al., was to propose a theory to explain why networks learn sinusoids. For their theory, they rigorously proved it was because learning at least 4 sine functions of each frequency was the max margin solution. They tested this in 1-layer MLPs and found that empirical reality matched. Our tests on multi-layer networks tell us this isn’t the full story: >=2-layer networks no longer learn the max margin solution.
>
> Indeed, we prove deep networks utilize O(log(n)) types of approximate cosets (features) and not the brute force O(n) max-margin, thus achieving the algorithmic efficiency of a divide and conquer strategy. We form a connection to the famous Chinese Remainder Theorem divide and conquer algorithm, showing it utilizes O(log(n)) cosets, which we point out are modular subsystems of the original problem (subproblems).
>
> Thus, neither the community or we have an answer. Lines 312-313 in future work propose further investigation into this.
>
> > Q4. Why is the maximum output logit over the entire dataset used in Theorem 4.7 and is there no tighter bound for this?
>
> The maximum logit for each piece of data is used. For any individual datum (a,b) input to the network, the maximum output logit is the correct logit. Thus, the bound is tight for all data. This will be clarified in the text.
>
> We think it’s unlikely a better bound exists because the bound is tight to experimental results. Recall: it predicts O(log(n)) features will be learned on average and the >= 0.989 $R^2$ values for logarithmic fits in Figure 3 support this. It also predicts the margin will grow logarithmically (before applying the softmax exponential), which is verified experimentally in Figure 12 in appendix C with $R^2 \geq 0.988$.
>
> # Final thoughts
>
> We hope that in light of these clarifications, and our presentation of a new position paper you see added value in our contributions at this time. We look forward to our continued discussion.
>
> ## References
> [1] Eberle et al., "Position: We Need An Algorithmic Understanding of Generative AI", ICML 2025.

---

> > ### Author Response · Authors · 2025-08-06
> >
> > Hi Reviewer,
> >
> > As the discussion period is wrapping up, we wanted to check in and see if you need any further clarifications from us. You mentioned
> > > I would definitely be inclined to raise my score if some of the questions were answered and it turns out I have missed something important.
> >
> > and we’ve done our best to respond to your concerns. Please let us know if there's anything else we can help clarify.

---

> > ### Comment · Reviewer_EPiJ · 2025-08-08
> > **Response to Authors**
> >
> > I thank the authors for their detailed response.
> >
> > My questions have been answered sufficiently and I have a better understanding now of some of the key aspects of this work. I recommend the authors make the promised changes and additions to detail and it will avoid future readers from similarly missing some of the subtle points.
> >
> > I will raise my score to advocate for acceptance in light of the rebuttal

---

### Official Review · Reviewer_PXn7 · 2025-07-04

**Clarity:** 3
**Significance:** 3
**Originality:** 4
**Rating:** 4
**Confidence:** 4

**Summary:**

The paper presents a novel theoretical framework for understanding how neural networks solve modular addition tasks through a method called aCRT. The authors propose that various neural network architectures utilize the same underlying algorithm to tackle these tasks, relying on a mechanism based on approximate cosets.  The paper also extends the framework to group multiplication tasks, offering empirical results to support the validity of the proposed theory.

**Questions:**

How does the proposed framework extend beyond modular addition? Can you provide experiments showing its applicability to more complex tasks? How does the framework apply to tasks beyond modular arithmetic or group multiplication? Are there limitations in generalizing the aCRT framework to other operations? I would appreciate some clarification on these points, particularly in the context of LLM applications. How could aCRT be applied to LLMs? Can it improve efficiency or interpretability, especially for tasks like reasoning or generation?

**Ethical Concerns:**

["NO or VERY MINOR ethics concerns only"]

**Final Justification:**

This paper explores the idea that neural networks solving modular addition may share a common underlying algorithmic structure. I initially raised questions about its generalizability beyond modular addition and the lack of comparison with similar methods. The rebuttal clarified these points and better positioned the contribution. While experiments remain focused on modular addition, the theoretical insight is strong and may inspire future work.

**Limitations:**

Yes

**Paper Formatting Concerns:**

No major formatting issues were found.

**Quality:**

3

**Strengths And Weaknesses:**

Strengths:
- This paper presents a novel approach by unifying previous interpretations of modular addition and extending this to group multiplication, which could have broader implications for understanding neural network mechanis
- The concept of aCRT and the use of approximate cosets to explain modular addition provides a new perspective on neural network behavior and offers potential computational benefits, improving efficiency.
- The theoretical framework is backed by solid empirical results that demonstrate its applicability to neural network architectures.

Weaknesses:
- The experiments in the paper primarily focus on modular addition tasks, with limited validation in more complex or real-world tasks. While the theoretical framework is strong, there is less discussion on its practical applications. Could you provide more experiments on the framework's application to more complex tasks, especially those beyond modular addition and group multiplication? This would help better understand the framework's generalizability and practical feasibility.
- The paper introduces the aCRT framework but does not directly compare it with existing similar methods. A comparison with existing approaches would offer clearer insights into the advantages and limitations of the framework in practical applications.

---

> ### Author Rebuttal · Authors · 2025-07-31
>
> Thank you for your review. We’re glad that you highlighted the unification of prior interpretations of modular addition (and its extension to group multiplication) as a key strength, as this was a central goal. We also appreciate your recognition of the novelty of the approximate Chinese Remainder (aCRT) framework, its potential to shed light on broader neural network mechanisms, and the potential computational benefits suggested by this perspective. Finally, we're glad you found our theoretical contributions well-supported by broad empirical results across architectures, which we see as essential to making robust and testable interpretive claims.
>
> The concerns mostly relate to the framework's applicability to more complex tasks. We address all concerns below.
>
> # Weaknesses
>
> > W1. The paper introduces the aCRT framework but does not directly compare it with existing similar methods. A comparison with existing approaches would offer clearer insights into the advantages and limitations of the framework in practical applications.
>
> We utilize the methods of prior work and in fact build on them.
> - **Nanda et al.**: Fit second order (degree 2) sines and cosines through neuron activations, then replaced the neurons by the fit to ensure accuracy remained 100%. We augment this with a more rigorous evaluation framework, testing order 1 (degree 1) sine and cosine fits as well as many more hyperparameters. This gave two novelties:
>   - Figure 8 shows that order 2 sinusoids are more expressive than necessary. Note: we will update the caption to state clearly that the purpose of this figure is to show that Nanda et al.’s second order sinusoid framework is too expressive for layer 1.
>   - The discovery of fine-tuning neurons that learn sawtooth functions, studied in Appendices G.4, G.5 and G.6. Notably, sawtooth functions violate the interpretations of all prior work, but they still fire on approximate cosets as remapping shows in Figure 28 in the appendix. Remark: they are only found with bad hyperparameters that cause overfitting of the training set so they may not be of relevance to deep learning practitioners, but serve to show how exhaustively we pushed prior methods.
> - **Zhong et al.** Primarily utilize Principal Component Analyses (PCA) to conclude two algorithms exist: pizza and clocks.
>   - Please see Appendix G.1 where we answer the open questions presented by Zhong et al., about their principal component analyses (PCA). By augmenting their PCA plots with the 2d-Discrete Fourier Transform (DFT), we show their circularity metric isn’t needed and that non-circular embeddings (Lissajous) are simply an artifact of PCA, i.e. Lissajous embeddings do not correspond to networks learning differences. We show a direct comparison between Zhong et al.’s PCA plots and our DFT augmentation in Figures 14-23.
>   - See Appendix E.1. where we state we download checkpoints for networks on their Github (clock and pizza).
> - **Morwani et al.** Showed empirically that 1 layer networks learn O(n) features (frequencies) and rigorously proved it. Morwani et al. only study the margin in one layer shallow networks.
>   - We generalize to depth (see Fig. 4 showing 1-layer networks learn O(n) whereas deep networks learn O(log(n))); the caption will be updated to indicate Morwani et al.’s case is in blue learning O(n) features.
>   - See Appendix C “Proofs and details for Theorem 4.7 and Corollary 4.8” where Figure 12 studies the empirical average margin over 500 networks. This supports the predicted margin of Corollary 4.8 for deeper networks.
>   - Appendix G.9.3, Fig. 43 shows how margin and cross entropy loss correlate as the number of layers are increased.
>
> > W2. The experiments in the paper primarily focus on modular addition tasks, with limited validation in more complex or real-world tasks. While the theoretical framework is strong, there is less discussion on its practical applications. Could you provide more experiments on the framework's application to more complex tasks, especially those beyond modular addition and group multiplication? This would help better understand the framework's generalizability and practical feasibility.
>
> We believe this is outside the scope of the paper. We worry such additions could distract from our core message and the importance of confirming Conjecture 4.9. We think our result that different architectures do not learn disparate algorithms like prior work suggested, and simply learn different implementations of one abstract algorithm is fundamental. **We’d like to note that reviewer EPiJ stated under significance** “Thus, the work stands to be highly significant." and later “I think this work will likely inform a lot of subsequent work and provide a useful perspective on the emergent behaviour of neural networks on algorithmic tasks.”.
>
> That said, you may find interesting that training language models to output a sorted list when given an unsorted list as input can be rewritten formally as a group multiplication. Thus, Conjecture 4.9 predicts that such models will utilize approximate cosets so we give a direct path for future work to immediately study more complex questions with language models using our framework.
>
> # Questions
>
> > Q1. How does the proposed framework extend beyond modular addition?
>
> Recall: approximate cosets are literally subproblems of the original problem—they divide the Cayley graph into smaller sets. Thus, the confirmation of Conjecture 4.8, that approximate cosets will be learned for all group multiplication tasks, would provide the foundation for a falsifiable theory of deep learning: the weights of neural networks converge to implementations of abstract divide and conquer algorithms.
>
> > Q2. Can you provide experiments showing its applicability to more complex tasks?
>
> See weakness 1 above Questions.
>
> > Q3.  How does the framework apply to tasks beyond modular arithmetic or group multiplication?
>
> See Weakness 2).
>
> > Q4. Are there limitations in generalizing the aCRT framework to other operations?
>
> There are no limitations in generalizing it to other groups, though the work to do so is constructive and will build on our framework. To generalize the aCRT to all tasks, first, conjecture 4.9 should be resolved. It’s formal resolution will yield hints for how to generalize the framework toward proving neural networks may implement abstract divide and conquer algorithms on all datasets. We could make this conjecture now because it does make sense—it explains the unreasonable efficiency of deep learning—but we believe it’s too early because it’s not yet resolved on group multiplications.
>
> > Q5. I would appreciate some clarification on these points, particularly in the context of LLM applications. How could aCRT be applied to LLMs? Can it improve efficiency or interpretability, especially for tasks like reasoning or generation?
>
> Primarily, this work serves to help interpretability by resolving the disparate interpretations of prior work and re-opening the universality hypothesis as a now testable conjecture. Secondarily, the ideas in this work have the potential to scale to address what LLMs learn on non-group tasks. Thus, they could, if they workout, provide the groundwork for a falsifiable theory of deep learning.
>
> ## A recent position paper gives context: our work is timely
>
> The ICML 2025 position paper, “We Need an Algorithmic Understanding of Generative AI” [1], argues that the ML community must move beyond fragmented, bottom-up interpretability, ​​which often focuses on individual neurons, components or circuits without a guiding hypothesis. Instead, they propose the prioritization of top-down hypothesis/theory-driven research that explains how models implement full algorithms. The authors write in reference to prior approaches: "current findings are still largely fragmented, and we lack a solid theoretical foundation for understanding how these various components come together to implement algorithms." Furthermore, they argue that such an approach would close the theory-interpretability gap, since both communities are operating rather disjointly. And this would lead to better understood, safer, and more efficient models.
>
> Our core message is in service of the research agenda outlined in [1], and we think this will help our work be extended to more complex tasks and relates to questions about LLMs.
>
> # Final thoughts
>
> We hope that we’ve successfully clarified our contributions and how our research fits into the literature and provides ample opportunities for future work. If this addresses your concerns clearly, we would greatly appreciate if you could consider increasing your score.
>
> ## References
>
> [1] Eberle et al., "Position: We Need An Algorithmic Understanding of Generative AI", ICML 2025.

---

> > ### Author Response · Authors · 2025-08-06
> >
> > Hi Reviewer,
> >
> > As the discussion period is wrapping up, we wanted to check in and see if you need any further clarifications from us.
> >
> > As your primary concern related to future generality, we'd like to mention that you may be interested in the discussion with reviewer RppL, particularly this part of our rebuttal:
> >
> > > We acknowledge that our methodology is focused on modular addition, and explicitly address this with Conjecture 4.9, to encourage future extensions, but we do not view this as a weakness because:
> >
> > > - Our paper unifies five prior works (Nanda et al., Chughtai et al., Zhong et al., Gromov, Morwani et al.) on modular addition, resolving previously conflicting interpretations and other open questions
> >
> > >- Additional tasks would harm the clarity of this contribution. Since the goal of this work is to reconcile and unify prior interpretations on modular addition, expanding to new tasks would distract readers, especially because prior work was scoped to modular addition
> >
> > >- Even if our methodology doesn’t transfer to other tasks, our work provides a complete theory of how deep networks implement modular addition, something no prior work has achieved
> >
> > >- We’re first to present a theoretical bound for deep networks on this task. Prior work addressed 1 layer MLPs
> >
> > > We recognize generalization is a natural next step, which is why we framed Conjecture 4.9 as a key forward-looking result. But within its scope, this work offers a definitive resolution to longstanding contradictions in the literature, especially those that challenged the universality hypothesis, and we believe this stands firmly on its own merit.
> >
> > Reviewer Rppl also asked a question that we address that should be of interest to you concerning LLMs:
> >
> > > Q1. Do you have an explanation of why 1 layer model cannot learn this? Is it related to the famous paper from Antrophic about induction head and non-learnability of the copy task from 1 layer model?
> >
> > Please let us know if there's anything else we can help clarify.

---

> > > ### Comment · Reviewer_PXn7 · 2025-08-07
> > >
> > > The authors’ rebuttal addressed many of my earlier concerns, particularly regarding the scope and positioning of the aCRT framework and its connection to prior work. Their clarifications around Conjecture 4.9, Theorem 4.7, and comparisons with existing methods helped strengthen the theoretical foundation.
> > >
> > > After reviewing the discussion, I find the paper presents a meaningful conceptual contribution to understanding modular addition and group-related tasks in neural networks. While the experiments remain centered on modular addition, the framework shows promise for broader applications.
> > >
> > > I maintain my original rating of Borderline Accept, as the paper offers solid theoretical value, though the experiments are still mostly limited to modular addition.

---

> ### Author Response · Authors · 2025-08-08
>
> Hi Reviewer PXn7,
>
> Thank you again for your thoughtful engagement throughout the review and discussion period.
>
> We understand your remaining concern: that the paper is borderline due to its experimental focus on modular addition. We’d like to respectfully offer context. Several influential papers that we unify, including Nanda et al. and Zhong et al., were scoped only to modular addition. Our work builds on theirs, but goes significantly further: we reconcile their fragmented findings, extend the analyses across architectures and depths, and provide the first unifying theoretical framework. *Our framework is first to extend to depth: yielding quantitatively accurate predictions of feature learning in multilayer networks (R² ≥ 0.99) in scaling experiments* (Fig. 3).
>
> We especially appreciate your recognition that the paper “presents a meaningful conceptual contribution” and that our clarifications “strengthened the theoretical foundation", but we resolve a major open concern in the interpretability community: Zhong et al.’s “Clock and Pizza” paper suggested that models can learn fundamentally different circuits to solve the same task, raising concerns about the transferability of mechanistic insights. Our analysis generalizes these seemingly distinct circuits to be different implementations of algorithm 4.6, thereby relieving this worry.
>
> This has broader implications for interpretability. Zhong et al.’s results were especially pressing because *if models diverge on modular addition, they may do so even more drastically on large-scale tasks like language modeling with LLMs*. Our results suggest a more optimistic view: even when circuits appear different, they may reflect different implementations of one algorithmic idea.
>
> We transparently highlighted future work with Conjecture 4.9, but we believe that presenting a state-of-the-art interpretation for modular addition, with theoretical and empirical rigor, constitutes a standalone contribution.
>
> We’re grateful for your time and thoughtful feedback, and hope this response clarifies the scope and significance of our contribution in light of past literature scoped only to modular addition. We hope this addresses your final concern.
>
> (added via edit): This worry is echoed in papers citing Zhong et al., including MoSSAIC: AI Safety After Mechanism (Farr et al.), which writes: "Researchers have successfully reverse-engineered these models and discovered that they learn a specific, intricate algorithm based on trigonometric identities and Fourier transforms, implemented via clock-like representations in the attention heads. The specific implementation is fundamentally tied to the architectural properties of the transformer substrate. A different architecture would almost certainly learn a different algorithm, rendering this detailed explanation obsolete."

---

### Decision · Program_Chairs · 2025-09-17

**Decision:**

Accept (poster)

**Comment:**

**Scientific claims and findings**:
This paper proposes a unifying theoretical framework, the approximate Chinese Remainder Theorem (aCRT), to explain how neural networks solve modular addition. The authors claim that seemingly different solutions learned by various architectures are actually implementations of this single, universal abstract algorithm. A key finding is the theoretical and empirically verified result that deep networks require only $O(\log n)$ features for this task, a significant efficiency gain over shallow networks.

**Strengths**:
* **Unifying Theory (aCRT):** The primary strength is its success in unifying several disparate and conflicting prior interpretations of how neural networks perform modular addition under a single, coherent framework, the approximate Chinese Remainder Theorem (aCRT).
* **Novel Concepts:** It introduces novel concepts like "approximate cosets" to provide a more general explanation that holds across different training conditions and architectures (MLPs, Transformers).
* **Strong Empirical Support:** The theoretical claims, particularly the $O(\log n)$ feature scaling for deep networks, are backed by extensive experiments across over a million trained models, demonstrating the robustness of the findings.

**Weaknesses**:
* **Clarity in Technical Sections:** Some reviewers found the initial presentation of the core theoretical concepts in Section 4 to be dense and lacking intuition, though the authors committed to revisions to improve clarity based on feedback.

**Reason for Accept**:
The paper's main contribution is conceptual and empirical: it resolves a key puzzle in the interpretability community by showing that different learned "circuits" can be manifestations of the same abstract algorithm. While the paper presents some theoretical analysis regarding margins, it does not culminate in a complete, end-to-end argument. In particular, a more relevant margin notion should be normalized margin, i.e., margin divided product of norm of parameters from each layer. (Otherwise large margin could be trivial to attain by just scaling up the last layer.) Nevertheless, this paper provides interesting and insightful new perspective on an important problem to the community, which will likely to inspire follow-up works.